# Dynamic expression of candidalysin facilitates oral colonization of *Candida albicans* in mice

Ricardo Fróis-Martins [1,2], Julia Lagler[1,2], Tim B. Schille [3,4], Osama Elshafee[3], Kontxi Martinez de San Vicente [1,2], Sarah Mertens[1,2], Michelle Stokmaier[1,2], Iman Kilb[1,2], Natacha Sertour[5], Sophie Bachellier-Bassi [5], Selene Mogavero [3], Dominique Sanglard [6], Christophe d'Enfert [5], Bernhard Hube [3,4,7] ✉ & Salomé LeibundGut-Landmann [1,2,8] ✉

*Candida albicans* is a common fungal member of the human microbiota but can also cause infections via expression of virulence factors associated with the yeast-to-hyphae transition. The evolutionary selection pressure to retain these pathogenic traits for a commensal microorganism remains unclear. Here we show that filamentation and hyphae-associated factors, including the toxin candidalysin, are crucial for colonization of the oral cavity, a major reservoir of *C. albicans*. Low-virulent strains of *C. albicans* expressed the candidalysin-encoding gene *ECE1* transiently upon exposure to keratinocytes in vitro. In mice, *ECE1* mutants were defective at accessing terminally differentiated oral epithelial layers where the fungus is protected from IL-17-mediated immune defence. Tight regulation of *ECE1* expression prevented detrimental effects of candidalysin on the host. Our results suggest that hyphae-associated factors such as candidalysin govern not only pathogenicity, but also mucosal colonization through direct host interactions enabling *C. albicans* to create and maintain its niche in the oral mucosa.

Microorganisms colonizing human epithelial tissues are crucial for maintaining and modulating host barrier homeostasis and physiology. While microbiome research has primarily focused on bacteria, recent studies underscore the beneficial effects of commensal fungi for the host. *Candida albicans* can enhance resistance to infections via the induction of type 17 immunity[1–3]. Yet, *C. albicans* is also a pathobiont causing mucosal infections[4,5]. Furthermore, *C. albicans* can cause life-threatening systemic disease[6] and is implicated in non-infectious chronic inflammatory disorders, including inflammatory bowel disease,

hepatitis and airway allergies[2,7–9]. These opposing roles emphasize the importance of understanding how *C. albicans* maintains a balanced homeostatic relationship with its host.

Several studies suggest that the yeast morphology and repression of hyphae-associated genes are required for *C. albicans* gut colonization, while hyphal growth and associated factors are linked to pathogenicity[3,10]. Such a scenario would favour avirulent strains[3]. However, most clinical isolates of *C. albicans* retain considerable virulence potential[11], raising the question why such traits

[1]Section of Immunology, Vetsuisse Faculty, University of Zürich, Zürich, Switzerland. [2]Institute of Experimental Immunology, University of Zürich, Zürich, Switzerland. [3]Department of Microbial Pathogenicity Mechanisms, Leibniz Institute for Natural Product Research and Infection Biology–Hans Knoell Institute (HKI), Jena, Germany. [4]Cluster of Excellence Balance of the Microverse, Friedrich Schiller University Jena, Jena, Germany. [5]Institut Pasteur, Université Paris Cité, INRAE USC2019, Unité Biologie et Pathogénicité Fongiques, Paris, France. [6]Institute of Microbiology, University of Lausanne and University Hospital Center, Lausanne, Switzerland. [7]Institute of Microbiology, Friedrich Schiller University Jena, Jena, Germany. [8]Medical Research Council Centre for Medical Mycology at the University of Exeter, Department of Biosciences, Faculty of Health and Life Sciences, Exeter, UK. ✉e-mail: bernhard.hube@leibniz-hki.de; salome.leibundgut-landmann@uzh.ch

have been conserved. One explanation is that hyphae may also play non-pathogenic roles during colonization[12,13]. In fact, hyphal formation and candidalysin—a hypha-associated peptide toxin encoded by the *ECE1* gene and known to damage epithelial barriers and trigger inflammation[14,15]—have been shown to support gut colonization by inhibiting competing bacteria[16].

Beyond the gut, *C. albicans* also inhabits the oral cavity, a potential primary reservoir for fungal colonization[11,17]. The oral mucosa, with its cornified stratified epithelium and continuous exposure to environmental stimuli, differs markedly from the gut and presents unique colonization challenges[18]. Unlike in the gut, where *C. albicans* is found in the lumen[10,16], in the oral mucosa it resides within the uppermost layer of the stratified epithelium[19,20]. However, the fungal factors supporting stable colonization in this niche are not well defined.

Here we show that *C. albicans* relies on filamentation and candidalysin to establish and maintain oral colonization. In this process, candidalysin can breach the stratum corneum supporting growth and immune evasion without host adverse effects. These findings suggest that classical virulence traits have evolved not primarily for pathogenicity but to promote stable mucosal colonization.

## Results

### Filamentation and hyphae-associated genes are required for *C. albicans* oral colonization

The requirement of filamentation for the initiation of *C. albicans* oral colonization was tested by comparing the filamentation-competent wild type with yeast-locked mutant strains, which are unable to filament owing to the deletion of essential gene(s) of the yeast–hyphae transition, in a model of oral colonization in immunocompetent and not antibiotically treated mice. Three distinct yeast-locked mutant strains, SC5314$^{efg1\Delta/\Delta}$ (ref. 21), SC5314$^{efg1/cph1\Delta/\Delta}$ (ref. 22) and SC5314$^{flo8\Delta/\Delta}$ (ref. 23) showed a drastically reduced fungal burden compared with the parental strain[24] by day 1 after fungal administration (Fig. 1a). We then tested the implication of hyphae-associated genes[25] in this process known to act at the *C. albicans*–epithelial cell interface (Extended Data Fig. 1a,b). *C. albicans* mutants lacking the adhesin Hwp1 (SC5314$^{hwp1\Delta/\Delta}$)[26], the secreted aspartyl proteases 4–6 (SC5314$^{sap456\Delta/\Delta}$)[27] or the gene *ECE1* encoding candidalysin (SC5314$^{ece1\Delta/\Delta}$)[14], but not strains lacking *HYR1* (SC5314$^{hyr1\Delta/\Delta}$)[28] or *SAP1–3* (SC5314$^{sap123\Delta/\Delta}$)[27], showed severely impaired host association (Fig. 1b–d and Extended Data Fig. 1c,d). The requirement for filamentation and hyphae-associated genes was evident as early as 8 h after fungal association (Fig. 1e–h and Extended Data Fig. 1e). Specific deletion of Ece1 peptide III-encoding sequences[14], the precursor of candidalysin, confirmed that candidalysin was responsible for the observed effect of Ece1 (Fig. 1f). By 2 h, yeast cells adhered topically to the tongue epithelium, and by 8 h, individual filaments breached through the epithelial barrier (Fig. 1i). Penetration of the stratum corneum appeared to present a bottleneck with only few fungal cells succeeding. This was also reflected by the drastically reduced number of fungal cells recovered from the tongue tissue by 8 h, while thereafter, the fungus proliferated and reached high fungal loads by 24 h (Fig. 1j). The initial focal interaction of *C. albicans* with the epithelium was independent of filamentation (Fig. 1k) and formation of microcolonies, which were described as a specialized biofilm structure regulated by Sfl1 or Sfl2 (ref. 29; Extended Data Fig. 1f,g).

To assess whether the process by which *C. albicans* associates with the tongue tissue was conserved among strains bearing a lower intrinsic capacity to filament and to express hyphae-associated genes than strain SC5314, we turned to low-virulent strain 101, originally isolated from a healthy child and bearing a low intrinsic capacity to cause epithelial cell damage[19,20,30]. The dynamics of filamentation, initial adhesion to and invasion of the stratum corneum and fungal load were remarkably similar between the two strains (Fig. 1i,j). In contrast to strain SC5314, which penetrated across the entire stratified epithelium of the murine tongue, strain 101 was confined to the stratum corneum, partially

reverted back to the yeast and pseudohyphal morphology by 24 h and did not elicit tissue inflammation (Fig. 1i)[19,20,31].

To further corroborate these findings and to test the implication of virulence determinants beyond filamentation in oral colonization, we deleted the hyphae-associated genes *ECE1*, *HWP1* or *SAP4-6* in strain 101 (101$^{ece1\Delta/\Delta}$, 101$^{hwp1\Delta/\Delta}$, 101$^{sap456\Delta/\Delta}$). Consistent with results obtained with mutants of the high-virulent strain SC5314 (Fig. 1b,f), candidalysin, but not Hwp1 or Sap4–6, was an important strain-independent factor for the establishment of oral colonization (Fig. 1l–n). The selective dependence on Ece1 may explain the slightly inferior capacity of strain 101 compared with SC5314 to settle colonization by 8 h (Fig. 1o).

### Low-virulent *C. albicans* acquires hallmarks of virulence for the establishment of oral colonization

The requirement of Ece1 and candidalysin for colonization by strain 101 appeared inconsistent with its low expression of *ECE1* (ref. 19). We therefore investigated the *ECE1* gene expression profile during the first 24 h of *C. albicans*–epithelial cell interaction in vitro, using human epidermal equivalents (HEEs) (Fig. 2a). The marked filamentation and epithelial invasion of both strains, SC5314 and 101, during the initial phase of colonization were largely comparable (Fig. 2a). *ECE1* expression was induced massively in a strain-independent manner within 2 h (Fig. 2b) and rapidly declined thereafter in the case of strain 101, but not SC5314 (Fig. 2b). The acute and transient induction of *ECE1* expression was recapitulated in TR146 oral keratinocytes and even in cell-free serum-containing medium, with even more transient expression kinetics for strain 101 despite persistent filamentation (Fig. 2c and Extended Data Fig. 2a,b). Similar gene expression profiles were also observed for the hyphae-associated genes *HWP1* and *SAP5* in strains SC5314 and 101 (Extended Data Fig. 2c,d), but not in the yeast-locked *C. albicans* SC5314$^{efg1/cph1\Delta/\Delta}$ mutant (Extended Data Fig. 2e). We also detected *ECE1* expression in the tongue tissue colonized for 24 h by both strains (Fig. 2d). In agreement with the in vitro data, *ECE1* mRNA levels were lower for strain 101 compared with SC5314, although differences were less dramatic than in vitro (Fig. 2d). Together, the highly dynamic and tight regulation of hyphae-associated genes, *ECE1* in particular, may enable the establishment of colonization within hours of *C. albicans*–host cell interaction.

### Candidalysin is required for sustained oral colonization

Continuous *ECE1* expression (Fig. 2e), which implies candidalysin's necessity beyond initial niche establishment, prompted us to perform a kinetics experiment in colonized mice to evaluate the importance of Ece1 over time. Similarly to what we observed at 8 h (Fig. 1l), the tissue fungal burden of 101$^{ece1\Delta/\Delta}$ was at least 5× reduced at 24 h compared with the parental 101$^{WT}$ strain (Fig. 2f and Extended Data Fig. 2f). The difference was even more pronounced by day 7 and 28 (Fig. 2g,h). A similar colonization defect was also observed with 101$^{ece1p3\Delta/\Delta}$ (Fig. 2g,h), confirming the relevance of candidalysin in the Ece1-mediated process. When lowering the 101$^{WT}$ inoculum 50× relative to 101$^{ece1\Delta/\Delta}$ to achieve even colonization levels of both strains at day 1 (101$^{WT}$ low infection dose (l.d.); Fig. 2f), the colonization defect of 101$^{ece1\Delta/\Delta}$ persisted, while the fungal burden of 101$^{WT}$ caught up over time (Fig. 2g,h). We then engineered strain 101 to achieve doxycycline-inducible repression of *ECE1* (101$^{tetOff\_ECE1}$; Extended Data Fig. 2g). Doxycycline administration from day 21, when colonization was established, resulted in almost complete clearance of the fungus by day 35 (Fig. 2i), while doxycycline did not affect colonization of 101$^{WT}$ (Extended Data Fig. 2h). To further address whether *ECE1* provides a fitness advantage, we performed a series of competition experiments with a fluorescent variant of 101$^{WT}$ (101$^{WT\_mCherry}$ (ref. 32), showing comparable colonization properties as 101$^{WT}$; Extended Data Fig. 2i) and non-fluorescent 101$^{ece1\Delta/\Delta}$. When co-colonizing mice with a 1:1 mixture, 101$^{WT}$ drastically outcompeted 101$^{ece1\Delta/\Delta}$ by day 3 (Fig. 2j). When using the two strains at a 1:50 ratio to eliminate the initial colonization difference, 101$^{WT}$ again outcompeted

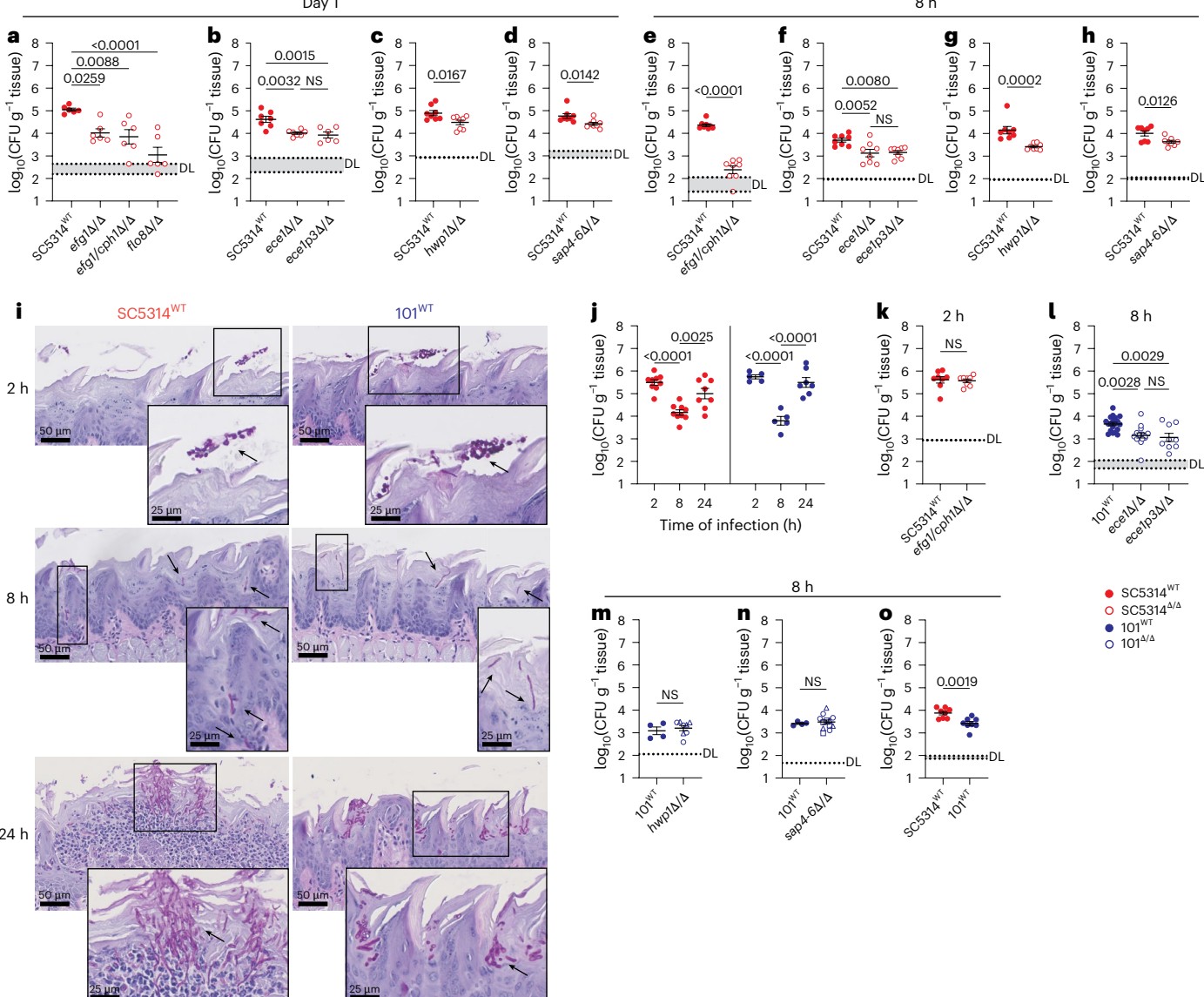

**Fig. 1 | Filamentation and hyphae-associated genes are required for *C. albicans* oral colonization.** C57BL/6 wild-type (WT) mice were associated with *C. albicans* strain SC5314 (red symbols) or 101 (blue symbols) via sublingual administration. Gene deletion mutants (open symbols) were compared with their corresponding parental strain. CFU per gram of tongue tissue are shown at the indicated time points. **a**–**h**, Tongue CFU of SC5314$^{efg1\Delta/\Delta}$, SC5314$^{efg1/cph1\Delta/\Delta}$ and SC5314$^{flo8\Delta/\Delta}$ (**a**, $n$ = 6 per group; **e**, $n$ = 8 per group; mean ± s.e.m.); SC5314$^{ece1\Delta/\Delta}$ and SC5314$^{ece1p3\Delta/\Delta}$ (**b**, $n$ = 6 or 7 per group; **f**, $n$ = 8 per group; mean ± s.e.m.); SC5314$^{hwp1\Delta/\Delta}$ (**c**, **g**, $n$ = 8 per group, mean ± s.e.m.); and SC5314$^{sap4-6\Delta/\Delta}$ (**d**, **h**, $n$ = 8 per group, mean ± s.e.m.) compared with the parental WT each at 1 day (**a**–**d**) or 8 h (**e**–**h**). **i**,**j**, PAS-stained tongue tissue sections (**i**) and tongue CFU at 2 h, 8 h and 24 h of colonization with SC5314$^{WT}$ or 101$^{WT}$ (**j**, $n$ = 5–9 per group and time point, mean ± s.e.m.). The black arrows indicate fungal elements. **k**, Tongue CFU of SC5314$^{efg1/cph1\Delta/\Delta}$ at 2 h; $n$ = 8 per group; mean ± s.e.m. **l**–**n**, Tongue CFU of 101$^{ece1\Delta/\Delta}$ and 101$^{ece1p3\Delta/\Delta}$ (**l**; $n$ = 9, 15 or 19 per group), 101$^{hwp1\Delta/\Delta}$ (**m**; $n$ = 4 or 8 per group, 2 different clones indicated by circles and triangles), 101$^{sap4-6\Delta/\Delta}$ (**n**; $n$ = 4 or 12 per group, 3 different clones indicated by circles, triangles and squares), all 101 background, at 8 h. Data presented as mean ± s.e.m. **o**, Tongue CFU of SC5314$^{WT}$ and 101$^{WT}$ at 8 h; $n$ = 8 per group; mean ± s.e.m. Each data point represents an individual mouse. Data are pooled from at least 2 independent experiments. DL, detection limit; NS, not significant. The statistical significance of differences between groups was determined by ordinary one-way ANOVA (**a**, **b**, **f**, **l**, **j**), two-sided Mann–Whitney test (**g**) or unpaired two-tailed *t*-test (**c**, **d**, **e**, **h**, **k**, **m**, **n**, **o**), respectively.

101$^{ece1\Delta/\Delta}$ by day 7 and fully eliminated 101$^{ece1\Delta/\Delta}$ by day 28 (Fig. 2k), confirming the fundamental role of *ECE1* not only to initiate but also to sustain fungal colonization. The predominant role of *ECE1* over other hyphae-associated genes was confirmed by the observation that neither 101$^{hwp1\Delta/\Delta}$ nor 101$^{sap4-6\Delta/\Delta}$ showed a defect in oral colonization by day 7 (Extended Data Fig. 2j).

**Candidalysin confers resistance to host antifungal responses**
Contrary to the gut, where *ECE1* provided a competitive advantage to *C. albicans* by impairing bacterial metabolic activity[16], the fitness advantage provided by *ECE1* in the murine oral mucosa appeared

to be independent of antibiotically induced dysbiosis (Fig. 3a and Extended Data Fig. 3a,b). The *ECE1*-dependent competitive advantage (Fig. 2k) coincided with the manifestation of IL-17 antifungal immunity rising from day 3 after fungal administration (Fig. 3b)[19,31], including IL-17 target genes as evidenced by RNA sequencing (RNAseq; Fig. 3c,d). The IL-17 pathway is critical for preventing *C. albicans* overgrowth, which manifests as mucocutaneous candidiasis in IL-17-deficient patients[33]. Therefore, we assessed whether *ECE1* deficiency affects fungal colonization of the oral mucosa in IL-17 receptor-deficient mice. Surprisingly, the fitness of strain 101$^{ece1\Delta/\Delta}$ compared with that of 101$^{WT}$ (50× reduced inoculum) was restored in *Il17rc*$^{-/-}$ mice (Fig. 3e and Extended Data Fig. 3c).

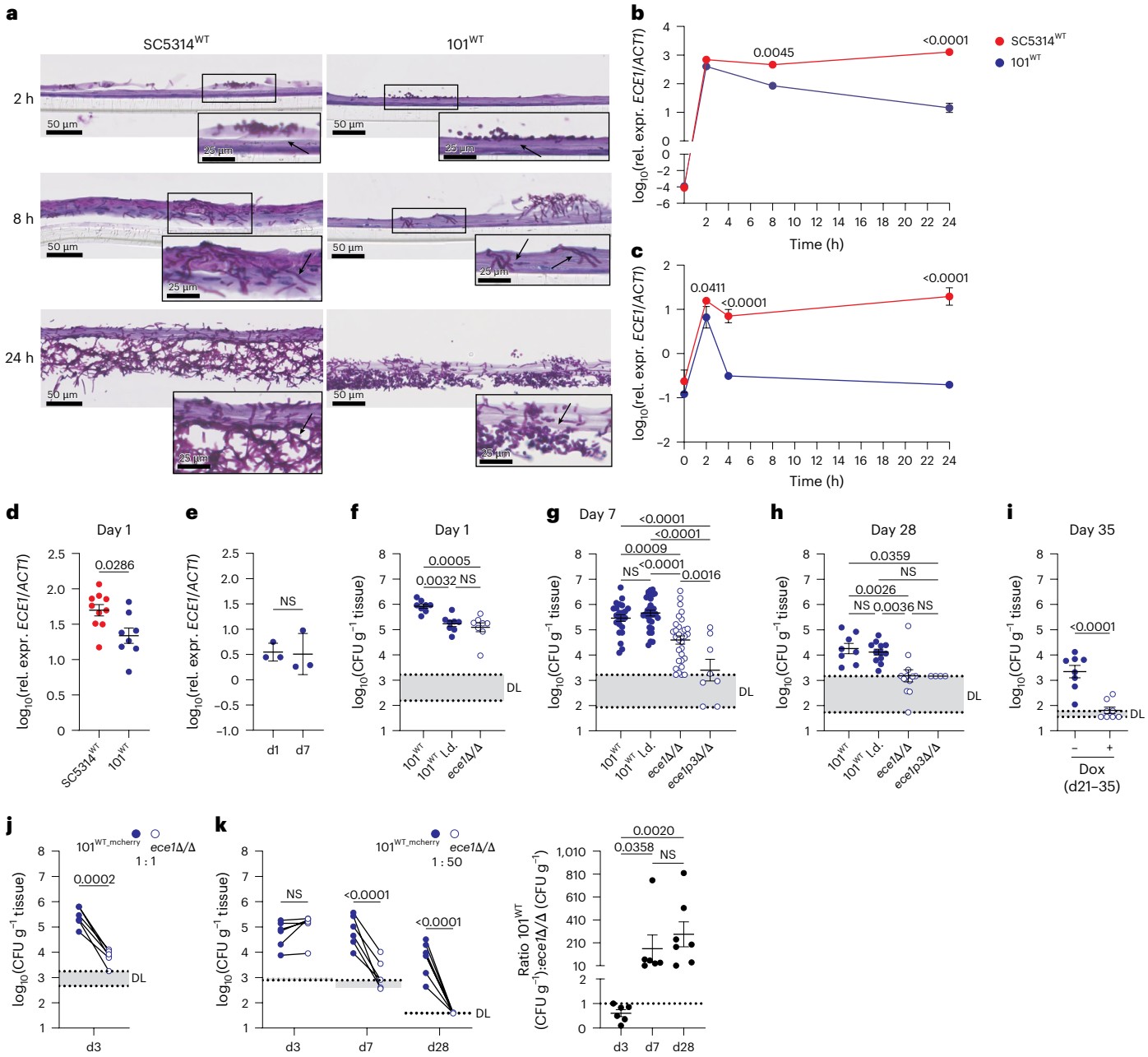

**Fig. 2 | Low-virulent *C. albicans* acquires hallmarks of virulence for the establishment and maintenance of oral colonization. a–c**, HEEs (**a,b**) and TR146 keratinocytes (**c**) were infected with *C. albicans* strain SC5314^WT or 101^WT for 2, 8 or 24 h, respectively, and analyzed by histology (**a**, PAS-stained HEE sections) and expression of *ECE1* transcripts (**b,c**); data are presented as mean per group ± s.d. The black arrows indicate fungal elements. **d,e**, C57BL/6 WT mice were associated with SC5314^WT or 101^WT, and *ECE1* transcript expression was quantified after 1 day (**d**, n = 8 or 10 per group, mean ± s.e.m.) or after 1 and 7 days (**e**, n = 3 per group, mean ± s.d.). **f–h**, C57BL/6 WT mice were associated with 101^WT, a 50× reduced inoculum of 101^WT (101^WT l.d.), 101^ece1Δ/Δ or 101^ece1p3Δ/Δ and tongue CFU were enumerated on day 1 (**f**), day 7 (**g**) or day 28 (**h**), n = 8–31 per group; mean ± s.e.m. **i**, C57BL/6 WT mice were associated with 101^tetOff-*ECE1* and treated with doxycycline (Dox) to induce *ECE1* deletion from day 21 to day

35, before tongue CFU were enumerated, n = 8 per group; mean ± s.e.m. **j,k**, Competitive colonization of C57BL/6 WT mice with 101^WT_mCherry and 101^ece1Δ/Δ at a 1:1 ratio for 3 days (**j**) or at a 1:50 ratio for 3, 7 and 28 days (**k**) before fluorescent and non-fluorescent tongue CFU were enumerated; n = 6 or 7 per group. Connected symbols are 101^WT and 101^ece1Δ/Δ CFU per individual animal (left) and fold differences between the two strains per animal (right). Data are presented as mean per group ± s.e.m. Data in **a–c** and **e** are from one representative of 3 independent experiments. Data in **d** and **f–k** are pooled from at least 2 independent experiments. The statistical significance of differences between groups was determined by two-way ANOVA (**k** (left)), repeated-measures two-way ANOVA (**b,c**), ordinary one-way ANOVA (**f,g,h**), Kruskal–Wallis test (**k** (right)), two-sided Mann–Whitney test (**e**), unpaired two-tailed *t*-test (**d,i**) or paired two-tailed *t*-test (**j**). rel. expr., relative expression.

A co-colonization experiment further confirmed that the competitive fitness advantage of 101^WT over 101^ece1Δ/Δ was lost in the absence of IL-17 signalling (Fig. 3f). Together, these results indicate that *C. albicans* also requires Ece1 to resist the antifungal response and to sustain oral colonization independently of the oral microbiota.

## Candidalysin promotes oral colonization by enabling *C. albicans* to overcome the barrier of the stratum corneum

To understand the mechanism underlying Ece1-mediated colonization, we found, in line with the literature[14], that *ECE1* did not affect *C. albicans* adhesion to epithelial cells in culture, filamentation nor

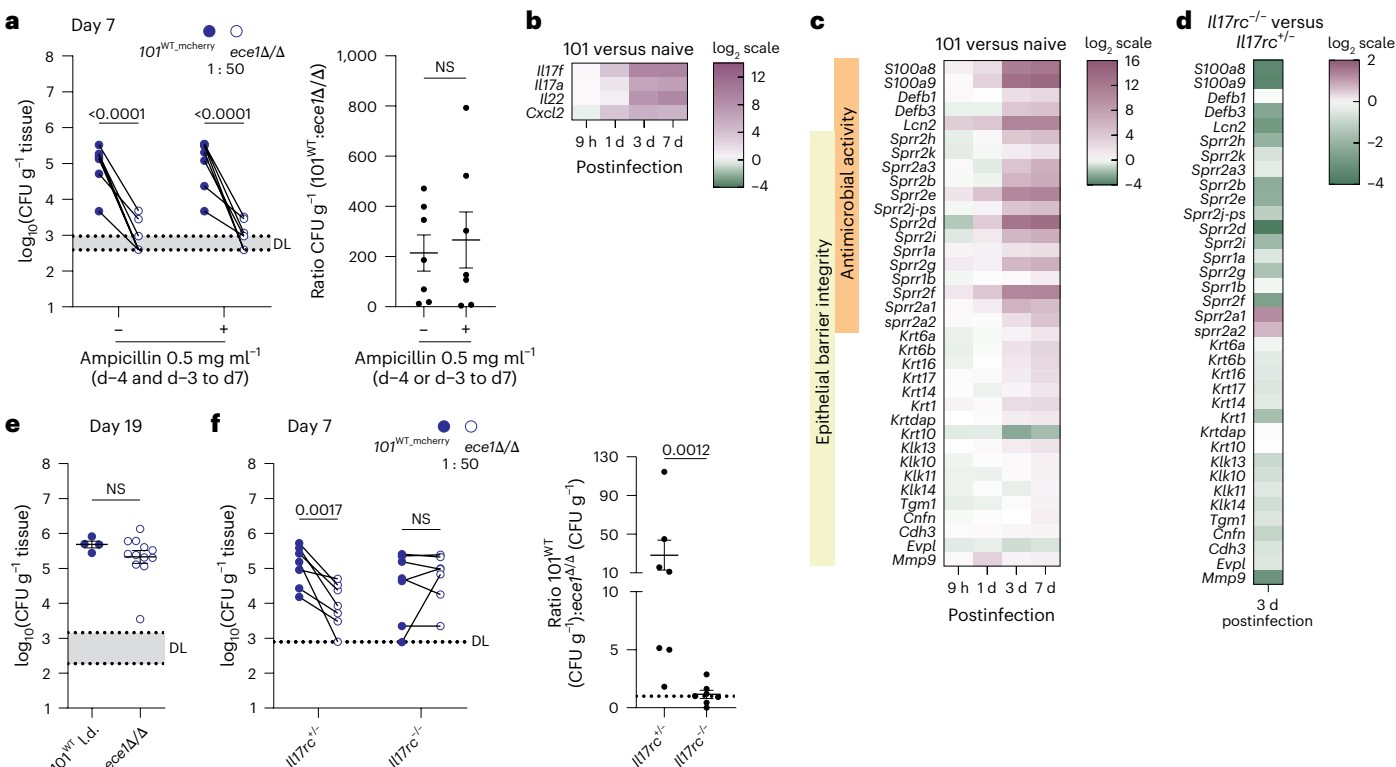

**Fig. 3 | Candidalysin confers resistance to host antifungal responses.**
**a**, Competitive colonization of C57BL/6 WT mice with 101$^{WT\_mCherry}$ and 101$^{ece1\Delta/\Delta}$ at a 1:50 ratio for 7 days before fluorescent and non-fluorescent tongue CFU were enumerated. One group of mice was treated with ampicillin in the drinking water from day −4 and −3 until the end of the experiment. Tongue CFU of either strain were determined based on the presence or absence of fluorescence, $n = 7$ per group. Connected symbols are 101$^{WT}$ and 101$^{ece1\Delta/\Delta}$ CFU per individual animal (left) and fold differences between the two strains per animal (right) in **a**. Data are presented as mean per group ± s.e.m. **b**,**c**, Differentially regulated genes (log$_2$(fold changes)) linked to antifungal immune response (**b**) and epidermal keratinization and antimicrobial activity (**c**) in the tongue of C57Bl/6 WT mice colonized with 101$^{WT}$ for the indicated period of time compared with naive mice. Data are extracted from a previously published RNAseq dataset[31]. **d**, Differential expression of the genes shown in **c** in *Il17rc*$^{−/−}$ compared with those in *Il17rc*$^{+/−}$ mice

colonized with 101$^{WT}$ for 3 days. Data are from an RNAseq dataset deposited in the NCBI GEO repository with accession number GSE280210. **e**, *Il17rc*$^{−/−}$ mice were associated with 101$^{WT}$ (1/50 inoculum) or 101$^{ece1\Delta/\Delta}$, and tongue CFU were enumerated on day 19; $n = 4$ or 12 per group; mean ± s.e.m. **f**, Competitive colonization of *Il17rc*$^{−/−}$ and *Il17rc*$^{+/−}$ mice with 101$^{WT\_mCherry}$ and 101$^{ece1\Delta/\Delta}$ at a 1:50 ratio for 7 days; $n = 7$ per group. Fluorescent and non-fluorescent tongue CFU were enumerated and fold differences between the two strains per animal are shown as in **a**; data presented as mean per group ± s.e.m. Data in **a**, **e** and **f** are pooled from at least two independent experiments. Paired datapoints in competitive colonization experiments (**a**,**f**) are connected with a line. The statistical significance of differences between groups was determined by two-way ANOVA (**a**,**f** (left)), unpaired two-tailed *t*-test (**a** (right), **e**), two-sided Mann–Whitney test (**f** (right panel)).

cellular invasion (Extended Data Fig. 4a–e). Candidalysin is known as a peptide toxin, which, at high concentrations, intercalates in host cell membranes, thereby inducing lytic cell death[14,34,35]. In contrast to strain SC5314 (ref. 34), strain 101 does not induce cellular damage in vitro (Fig. 4a and Extended Data Fig. 4f). At high fungal cell concentrations, strain 101 did, however, provoke host cell damage in spontaneously immortalized murine oral keratinocyte (IMOK) cells[36] showing an increased sensitivity towards cell death induction compared with the more commonly used TR146 cell line. The response by IMOK cells was independent of *ECE1*, *HWP1* or *SAP4−6* factors (Extended Data Fig. 4g–j), which may be explained by the limited gene expression levels of strain 101 (Fig. 4b and Extended Data Fig. 4k). To assess the consequences of *ECE1* under conditions more closely representing the situation in vivo in which expression levels by strain 101 are only slightly lower than in SC5314 (Fig. 4b), we aimed at overexpressing *ECE1* in strain 101. Expression under control of the *TDH3* or *HWP1* promoters did not result in increased induction of cellular damage (Extended Data Fig. 4l)[20]. We then took advantage of strain 101$^{tetOff\_ECE1}$, which, in the absence of doxycycline, overexpressed *ECE1* in vitro (but not in vivo) owing to the insertion of a strong promoter in strain 101 to levels comparable to those in strain SC5314 (Extended Data Fig. 4m,n). Compared with the parental strain 101$^{WT}$, 101$^{tetOff\_ECE1}$ (-doxycycline) elicited enhanced

lactate dehydrogenase (LDH) release from IMOK cells (Fig. 4a) and did so in a contact-dependent manner (Fig. 4c), as expected. Strain 101 is thus indeed capable of inducing cellular damage in an *ECE1*-dependent manner despite the complex regulation of candidalysin-mediated cellular damage beyond *ECE1* expression[34].

We therefore returned to our in vivo model to assess the mechanism by which candidalysin facilitates the establishment of *C. albicans* colonization in the complex environment of the stratified oral epithelium. Because too few fungal elements were retrievable at 8 h to observe invasion on tissue sections (Fig. 1i,j,o), we instead removed tongues 2 h after fungal association—minimizing loss to saliva and mechanical forces—and allowed ex vivo completion of the colonization process. Loss of *ECE1* completely prevented *C. albicans* from invading the epithelium (Fig. 4d). Remarkably, fungal elements of both 101$^{ece1\Delta/\Delta}$ and SC5314$^{ece1\Delta/\Delta}$ strains remained peripheral despite extensive filamentation (Fig. 4d,e and Extended Data Fig. 4o). Extending the incubation period to 24 h made the critical role of Ece1 for invasion even more apparent (Fig. 4f). Ece1-dependent tissue invasion was not affected by pretreating mice with ampicillin (Extended Data Fig. 4p). The Ece1-mediated advantage for sustained colonization was also apparent at day 7, when the fungal burden of 101$^{ece1\Delta/\Delta}$ was low (Fig. 2g) and colonization foci were rare. In contrast to Ece1-competent

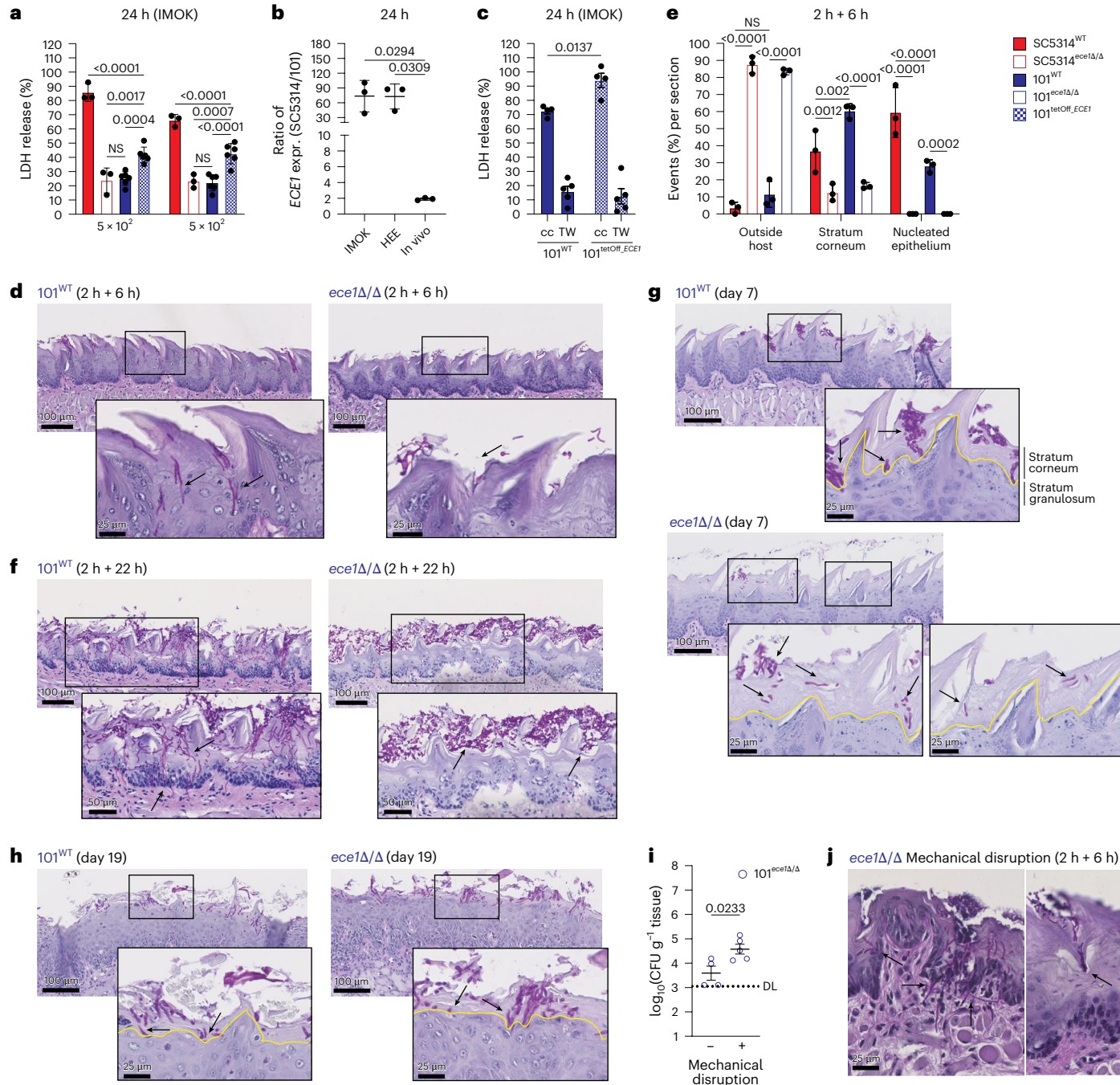

**Fig. 4 | Candidalysin promotes oral colonization by enabling *C. albicans* to overcome the barrier of the stratum corneum. a**, LDH release from IMOK cells in response to SC5314^WT, SC5314^ece1Δ/Δ 101^WT and 101^tetOff_ECE1 after 24 h of incubation (*n* = 3 or 6 per group). Each symbol represents an individual well with mean per group ± s.d. Data are from one representative of at least five independent experiments. **b**, Ratio of *ECE1* expression by SC5314 versus 101 after 24 h of exposure to IMOK keratinocytes, HEEs or murine tongue tissue (in vivo). Each symbol represents an independent experiment (mean of 1–3 replicates each) with mean ± s.d. per group. **c**, LDH release from IMOK cells in response to 101^WT or 101^tetOff_ECE1 (1 × 10^5 yeast cells per well) in direct contact (cc) or transwell (TW) for 24 h (*n* = 3 or 4 per group). Each symbol represents an individual well with the mean per group ± s.e.m. Data are pooled from 2 independent experiments. **d–f**, C57Bl/6 WT mice were associated with 101^WT or 101^ece1Δ/Δ for 2 h; tongues were collected and incubated for 6 h (**d,e**) or 22 h (**f**) in a humid chamber at 37 °C, 5% CO_2. Representative PAS-stained tongue tissue sections

(**d,f**) and quantification of fungal foci outside the host, in the stratum corneum or in nucleated epithelial layers (**e**). Each symbol represents the percentage of all events per category and section from an individual mouse (>28 events per section); *n* = 3 animals per group with mean ± s.d. per group. **g,h**, Representative PAS-stained tongue tissue sections from C57Bl/6 WT mice (**g**) or *Il17rc*^−/− mice (**h**) associated with 101^WT or 101^ece1Δ/Δ for 7 days (**g**) or 19 days (**h**). The yellow lines indicate the boundary between the stratum corneum and stratum granulosum. The black arrows indicate fungal elements. Data are pooled from at least 2 independent experiments. **i–j**, The tongue of C57Bl/6 mice was (+) or was not (−) mechanically disrupted before the association with 101^ece1Δ/Δ Tongue CFU on day 1 (**i**) and representative histology sections of tongues at 2 h plus additional incubation in a humid chamber at 37 °C, 5% CO_2, for 6 h (**j**); *n* = 4 pooled from 2 independent experiments with mean ± s.e.m. The statistical significance of differences between groups was determined by two-way ANOVA (**a,c,e**) or ordinary one-way ANOVA (**b**).

*C. albicans* cells, which frequently cross the stratum corneum, reaching and breaching into the stratum granulosum, *ECE1*-deficient fungal cells were preferentially observed in the outer part of the stratum corneum (Fig. 4g). In *Il17rc*[−/−] mice, however, ECE1 is redundant for *C. albicans* to find its niche in the stratified epithelium (Fig. 4h). To provide further evidence for the proposed mode of action of Ece1 during oral colonization, we experimentally disrupted the cornified epithelial layer before fungal association. This enabled strain 101[*ece1Δ/Δ*] to overcome its defect in colonization and to establish its niche (Fig. 4i,j). These data show that Ece1 enables *C. albicans* to access a safe niche that prevents it from being shed from the oral epithelium, which exhibits a rapid turnover[37].

### Tight regulation of *ECE1* expression ensures colonization by evading the inflammatory immune response

While the processes establishing *C. albicans* colonization of the oral mucosa are largely conserved between high- and low-virulent strains, the longer-term outcome of the fungus–host interaction differs greatly between the strains[19,20]. As such, strain SC5314 penetrates much deeper into the nucleated layers of the stratified epithelium than strain 101 (Figs. 1i and 4d,e, and Extended Data Fig. 3n). The in vivo phenotype of 101[WT] is reminiscent of SC5314[*eed1Δ/Δ*], a mutant lacking the hyphal extension gene *EED1* (ref. 38). As sustained colonization of the oral mucosa is supported by restrained filamentation and hyphae-associated gene expression[20], we examined the colonization capacity of SC5314[*eed1Δ/Δ*] compared with that of SC5314[WT] and 101[WT]. While filamentation is initially induced in SC5314[*eed1Δ/Δ*], and sufficient for the mutant to invade epithelial cells[38], hyphal growth is not maintained. Once inside the epithelial cell, *EED1* deficiency prevents the mutant from escaping and developing full virulence as it rapidly reverts to yeast cell growth[38,39]. Accordingly, *ECE1* expression by SC5314[*eed1Δ/Δ*] was transient in response to epithelial cell contact in vitro (Fig. 5a) reminiscent of what we observed with strain 101[WT] (Fig. 2c). In vivo, again similar to strain 101[WT], SC5314[*eed1Δ/Δ*] efficiently colonized the stratum corneum (Fig. 5b–d), reaching even higher fungal counts than SC5314[WT] on day 1, although it persists mainly in yeast morphology (Fig. 5c,d). Like 101[WT], and in contrast to SC5314[WT], SC5314[*eed1Δ/Δ*] did not elicit an inflammatory response (Fig. 5e,f and Extended Data Fig. 5a,b), emphasizing the relevance of restrained virulence for preventing immunopathology and hence promoting long-term colonization (Fig. 5g). Of note, *ECE1* was even more repressed in SC5314[*eed1Δ/Δ*] than in 101[WT] (Fig. 5h). Consistent with the requirement of continuous moderate expression of *ECE1* for persistence in the oral cavity, SC5314[*eed1Δ/Δ*] showed inferior capacity to sustain colonization than 101[WT] (Fig. 5g). This difference between 101[WT] and SC5314[*eed1Δ/Δ*] was even more pronounced in a competitive setting (Fig. 5i). Overexpression of *EED1* in strain 101 (101[OE_*EED1*]) enhanced filamentation and *ECE1* expression in vitro (Fig. 5j,k). Moreover, 101[OE_*EED1*] elicited the recruitment of inflammatory cells to the site of fungal colonization in the oral mucosa, which resulted in fungal clearance by day 7 (Fig. 5l–n), reminiscent of what happens in the case of strain SC5314 (Fig. 5g). Inflammation foci characterized by infiltration of neutrophils in close proximity to fungal elements were significantly increased in the tongue of mice colonized with 101[OE_*EED1*] compared with those with 101[WT]. The difference was even more pronounced when normalized to the fungal burden per tongue (Fig. 5n), given the low colonization efficiency of 101[OE_*EED1*] due to its aggregating behaviour. These data illustrate how tight regulation of *ECE1* expression via regulation of hyphal formation or maintenance curtails inflammation while promoting fungal fitness in the oral niche that facilitates fungal persistence in the mucosal tissue.

## Discussion

Best known for its role as a pathogen causing superficial and systemic, often fatal, infections, *C. albicans* is primarily a harmless colonizer in most healthy individuals[11,40]. The mechanisms enabling stable, homeostatic colonization remained elusive[13]. We show that candidalysin—originally identified as a virulence factor—is essential for *C. albicans* to establish and sustain its niche in the oral epithelium, while excessive expression is avoided to prevent tissue damage and inflammation. The stratum corneum provides a safe space protected from direct exposure to antimicrobial peptides and IgA in the saliva[41,42]. These findings complement recent data showing that candidalysin targets commensal gut bacteria to overcome colonization resistance[16] and highlight the pleiotropic functions of candidalysin in health and disease. Together, this reframes our understanding of *C. albicans* colonization in different tissue compartments and provides evidence for why fungal determinants have been conserved through evolution[43] despite their pathogenic potential bearing a constant threat to the fungus–host equilibrium. As a pore-forming peptide, candidalysin agglomerates and incorporates in the host cell membrane when accumulating in the invasion pocket that is formed by *C. albicans* hyphae[34,35], a process resulting in cell damage. Our data suggest that candidalysin acts via the same mechanism during homeostatic colonization of the oral mucosa to overcome the rigid barrier of the stratified epithelium, which consists of terminally differentiated transcriptionally inactive keratinocytes. Overexpression of *ECE1* in strain 101 allowed demonstrating that this strain, despite its low virulence, can confer detectable cell damage via candidalysin. The continuous dependence of colonization on candidalysin suggests that the fungus is constantly repeating the cycle of tissue invasion to renew its niche and avoid being shed from the epithelium due to desquamation of the stratified oral epithelium[44]. IL-17 can accelerate tissue turnover and consequent cell shedding from stratum corneum by enhancing the proliferation rate of epithelial cells[45]. This may explain the competitive advantage of *ECE1*-competent versus deficient *C. albicans* in immunocompetent mice, which was not observed in the absence of a functional IL-17 pathway.

Desquamation acts as a host defence mechanism against microbial colonization as shown for *Neisseria gonorrhoeae*, which evades these host defense activities by suppressing exfoliation of the epithelium[46]. *C. albicans* may inhibit epithelial cell shedding by inducing interferon type 1 (refs. 47,48). In turn, some commensal bacteria can enhance shedding of epithelial cells and thereby reduce host cell damage induced by pathogenic *C. albicans*[49].

Despite the fact that strains 101 and SC5314 greatly differ in their intrinsic host-damaging potential[19], the initial process of oral colonization was unexpectedly well conserved in contrast to the longer-term outcome of colonization. Although originally reported to show only very limited capacity to filament and to lack expression of virulence factors[19,20], we show that strain 101 does filament efficiently and robustly and expresses hyphae-associated genes at the host interface or when exposed to host-like conditions. The selective dependence on *ECE1* for oral colonization by strain 101 may explain its shallower penetration depth in the stratified epithelium—compared with strain SC5314—to prevent access to deeper epithelial layers where induction of cell damage results in the production of antimicrobial effectors, release of inflammatory cues and rapid recruitment of inflammatory leukocytes, which in turn eliminate the fungus[14,19,50,51]. Whether in strain 101 the biological activity of candidalysin is regulated by additional mechanisms beyond *ECE1* transcription remains to be determined[34,52,53].

Restricted *ECE1* expression is at least partially explained by high *NRG1* expression[20]. As such, we previously showed that suppression of *NRG1* in strain 101 enhances fungal virulence, translating in enhanced inflammation and more rapid fungal clearance in experimentally colonized mice[20]. In its role as a transcriptional repressor, Nrg1 also regulates filamentation by controlling *EED1* (ref. 39). We found Eed1 to maintain *ECE1* expression during hyphal extension validating the strong link between hyphae and hyphae-associated gene expression[25]. Consequently, *EED1* deletion recapitulated features of homeostatic colonization in strain SC5314; however, this was less pronounced than in strain 101 owing to a substantial decrease of *ECE1* expression and filamentation. Conversely, *EED1* overexpression in strain 101

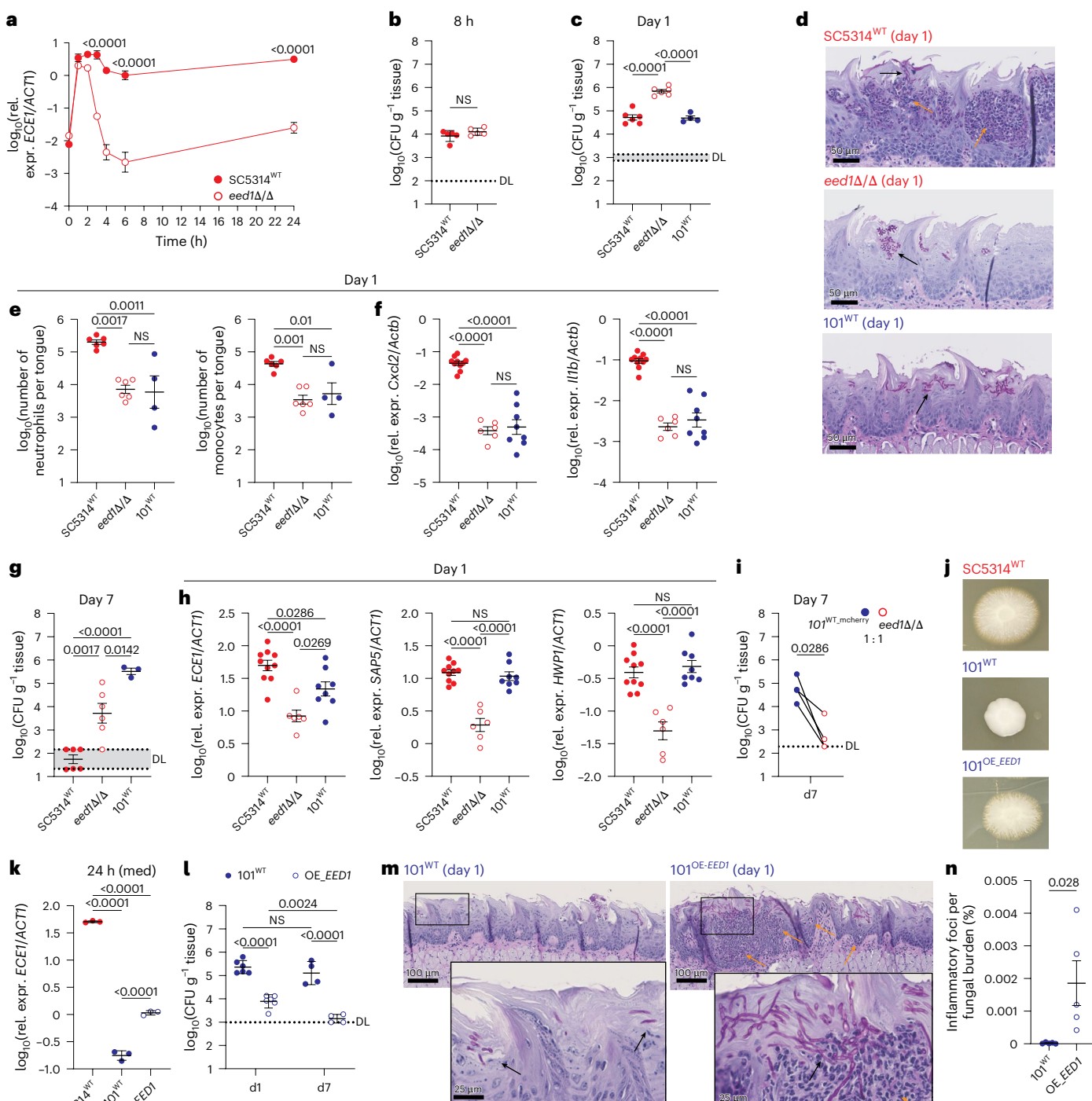

**Fig. 5 | Tight regulation of *ECE1* expression ensures long-term colonization by evading the inflammatory immune response. a**, *ECE1* expression by SC5314[WT] or SC5314[*eed1Δ/Δ*] exposed to TR146 keratinocytes for 0, 2, 4, 8 or 24 h. Each symbol represents the mean ± s.d. (*n* = 3 per group). **b–h**, C57Bl/6 WT mice associated with SC5314[WT], SC5314[*eed1Δ/Δ*] or 101[WT] for 8 h (**b**), 1 day (**c,e,f,h**) or 7 days (**g**). Tongue CFU (**b**, **c**, **g**, *n* = 4 or 6 per group). Representative PAS-stained histology sections (**d**). Quantification of tongue neutrophils (CD45+Ly6G+Ly6C−) and inflammatory monocytes (CD45+Ly6G−Ly6C+) by flow cytometry (**e**, *n* = 4 or 6 per group). *Cxcl2* and *Il1b* host transcripts (**f**) and *ECE1*, *SAP5* and *HWP1* fungal transcript (**h**) in the colonized tongue tissue (**f,h**, *n* = 6, 8 or 10 per group; **g**, *n* = 3 or 6 per group). Each symbol represents the mean per group ± s.d. (**b**) or mean per group ± s.e.m. (**c,e,h**). **i**, Competitive colonization of C57Bl/6 WT mice with 101[WT_mCherry] and SC5314[*eed1Δ/Δ*] (1:1 ratio) for 7 days; *n* = 4 per group. Connected symbols are 101[WT_mCherry] and SC5314[*eed1Δ/Δ*] CFU per individual animal.

**j,k**, SC5314[WT], 101[WT] and 101[OE_EED1] on spider agar at 37 °C, 5% $CO_2$, for 5 days (**j**) or in serum-containing medium (med) for 24 h before quantification of the *ECE1* transcript by RT-qPCR (**k**; *n* = 3 per group; mean ± s.d.). **l–n**, C57Bl/6 WT mice associated with 101[WT] or 101[OE_EED1]. Tongue CFU on day 1 and day 7 (**l**, *n* = 4 or 6 per group; mean ± s.d.). Representative PAS-stained histology sections on day 1 (**m**). Proportion of inflammatory foci (myeloid infiltrates) per fungal colonization site relative to the total number of CFU per tongue (**n**, *n* = 5 per group; mean ± s.e.m.). In **d** and **m**, the black arrows indicate fungal elements and the orange arrows indicate inflammatory infiltrates. Data are pooled from 2 to 3 independent experiments except for **b**, **i** and **k**. In **h**, data for SC5314[WT] and 101[WT] are the same as those shown in Fig. 2d. The statistical significance of differences between groups was determined by two-way ANOVA (**a,l**), ordinary one-way ANOVA (**c,e,f,g,h**), two-sided Mann–Whitney test (**b**), unpaired two-tailed *t*-test (**n**) or paired two-tailed *t*-test (**i**).

enhanced pathogenic traits. Together, this illustrates the importance of well-balanced virulence factor expression for evading inflammation while enabling sustained colonization. Our results further highlight the tissue-specific requirements for *C. albicans* colonization, as *EED1* deficiency provides a fitness advantage in systemic organs[54], which is not the case in the oral mucosa, where tonic virulence is a fitness benefit.

In summary, our findings attribute candidalysin additional functions beyond the previously known ones. In addition to serving as a classical virulence factor that causes host damage, and its function as an avirulence factor that stimulates a protective damage defense response in the host[12], we show here that candidalysin is essential for successful long-term oral colonization of *C. albicans* by directly interacting with the host to create and maintain its niche in the oral mucosa without triggering tissue damage or inflammation. This property has probably contributed to the conservation of this toxin throughout *C. albicans* evolution[55].

## Limitations of this study

We show that candidalysin enables *C. albicans* to establish its niche in the oral mucosa in both high- and low-virulent isolates. Future studies with additional isolates will clarify conservation across the species regarding the *ECE1*-dependent mechanism.

We cannot exclude additional mechanisms by which candidalysin may facilitate oral colonization. The epithelial niche may, for instance, provide the fungus access to nutrients and hence contribute to its metabolic adaptation to the host[56]. Environmental cues to which *C. albicans* gets exposed in the niche may also modulate the fungal properties. Furthermore, we cannot exclude additional molecules that may support candidalysin's function in facilitating oral colonization. We have not directly assessed these possibilities in this work.

Our study analysed the colonization process in the stratified epithelium of the tongue. It remains to be shown whether candidalysin is similarly involved in colonization of other stratified epithelia of the oral cavity and possibly the vaginal mucosa, while colonization of the gut is probably regulated differently given the distinct architecture of the epithelial lining in the gut.

## Methods

### Fungal strains

*C. albicans* strains used in this study are listed in Supplementary Table 1. All strains were maintained on YPD agar for short-term storage and in glycerol-supplemented medium at −80 °C for long-term storage. For fungal assays, *C. albicans* strains were first pre-cultured for 8 h in 5 ml YPD medium, then inoculated at $OD_{600} = 0.1$ in 10 ml of YPD medium and grown overnight, 15–18 h, at 30 °C and 180 rpm. At the end of the culture period, yeast cells were washed 3 times in PBS and their concentration was determined by spectrophotometry, whereby $1\ OD_{600} = 10^7$ yeast cells.

### Generation of *C. albicans* mutants

The deletion mutant strain in the *C. albicans* 101 background was generated using CRISPR–Cas9 technology[57]. We used a CRISPR plasmid constitutively expressing *C. albicans* codon-optimized Cas9, locus-specific single guide RNA and a *SAT1* resistance marker combined with a Flp-recombinase and FRT sites flanking the entire cassette together with a homology-directed repair template with 60 bp up- or downstream homology to the target locus and containing 20 bp overlap at their 3′ ends. Repair templates were generated by extension of DNA oligos using *Taq* polymerase (NEB). The guide RNAs for specific loci were identified using CRISPOR.org[58]. The corresponding DNA oligos were annealed and ligated into the guide RNA cloning site of the CRISPR plasmid sequence, and orientation was confirmed by Sanger sequencing. Then, 5 µg of linearized CRISPR plasmid and 3.5 µg of repair template were transformed into *C. albicans* using the polyethylene glycol (PEG3650)/lithium acetate (LiAc) method[59] and selected on

YPD containing 200 g ml[−1] nourseothricin. Positive locus deletion was verified by colony PCR. Removal of the CRISPR cassette was achieved by induction of Flp recombination. The pNIM1R plasmid[60] was used to generate a tet-repressible *ECE1*-expression mutant. The *ECE1* open reading frame of *C. albicans* 101 was cloned into pNIM1R at *Sal*I and *Bgl*II. The transformation cassette was excised with *Sac*II and *Kpn*I, and 5 µg was transformed into 101$^{ece1Δ/Δ}$ as described above. Correct integration into the *ADH1* locus was confirmed via colony PCR.

For generating 101$^{OE\_EED1}$, the plasmid for *EED1* overexpression was obtained by PCR amplifying *EED1* on SC5314 genomic DNA using primers VB40 and VB41 and cloning it into pCRII-TOPO for sequencing. The *EED1*-bearing BstZI-NotI fragment was then inserted into pKS-P$_{TDH3}$ cut with EcoRV and NotI. (pKS-P$_{TDH3}$ results from the insertion of a PCR fragment, amplified on SC5314 genomic DNA with primers SZ11 and SZ12, in XhoI and EcoRV-cut pBluescript-KS(+)). The resulting plasmid was then cut with NsiI and ApaI and the *EED1*-bearing fragment ligated in the same sites of CIpSAT-P$_{TDH3}$-GTW (CdE lab collection), yielding CIpSAT-P$_{TDH3}$-*EED1*. This plasmid was linearized with StuI before the 101 strain was transformed using the lithium acetate protocol[61].

A list of the oligonucleotide sequences for PCR and sequencing used in this study is provided in Supplementary Table 2.

### Keratinocyte cell culture

The human oral keratinocyte cell line TR146 (ref. [62]; Sigma, reference number 10032305, previously used in the LeibundGut-Lab) was grown in DMEM medium supplemented with 10% FCS, penicillin and streptomycin. The murine oral keratinocyte cell line IMOK[36] (obtained from L. Garrett-Sinha) was grown in CnT-07 Epithelial Proliferation Medium (CELLnTec). The human keratinocyte cell line N/TERT1 (ref. [63]; obtained from H.-D. Beer) was grown in K-SFM (Gibco) supplemented with 0.005% (w/v) Bovine Pituitary Extract (Gibco) and 0.02% (v/v) Epidermal Growth Factor (Gibco). All cell lines were maintained at 37 °C, 5% $CO_2$. For infection experiments, cells were seeded at $2 \times 10^5$ cells per well in 24-well tissue culture plates or at $2 \times 10^4$ cells per well in 96-well tissue culture plates, respectively, and grown to confluent monolayers for 2 days before infection as described below for the individual assays. In the case of TR146 cells, 1 day before the experiment, the DMEM medium was replaced with F12 medium (Hams's Nutrient Mixture F12 medium (Sigma) supplemented with L-glutamine, penicillin, streptomycin and 1% FCS). In the case of the IMOK cells, the medium was renewed on the day of the infection.

### HEEs

To generate HEEs[64], $2 \times 10^5$ N/TERT1 cells per well were seeded in cell culture inserts in a 24-well plate containing 1 ml of CnT prime medium (CELLnTec) in the bottom of the well and incubated at 37 °C, 5% $CO_2$ until confluent. The medium was replaced with 1 ml (bottom) and 100 µl (insert) of CnT Prime 3D Barrier medium (CELLnTec). After overnight incubation at 37 °C, 5% $CO_2$, the medium was carefully removed from the insert to start the air–liquid-interface culture. The medium in the bottom well was replaced every 2–3 days over 3 weeks until HEEs were fully differentiated. For infection, $8 \times 10^4$ yeast cells in 100 µl of PBS were applied to HEEs. The excess liquid was removed after 30 min, and HEEs were incubated at 37 °C, 5% $CO_2$, for 2, 8 or 24 h as indicated. To collect the 0 h timepoint, HEEs were centrifuged immediately after seeding *C. albicans* and excess liquid was removed. For histology, HEE samples were collected by detaching the tissue from the membrane and fixing in 4% PBS-buffered paraformaldehyde. For RNA isolation, HEEs were collected in TRI reagent (Sigma) and stored at −20 °C until further processing.

### Adhesion assay

Adhesion of *C. albicans* to TR146 keratinocytes was determined as described previously[34]. Briefly, *C. albicans* cells were added to confluent TR146 cells in 24-well plates on 12-mm-diameter glass coverslips

to a final concentration of $1 \times 10^5$ cells per well and incubated for 1 h at 37 °C, 5% $CO_2$. Non-adherent *C. albicans* cells were removed by washing with PBS. Samples were fixed with 4% PBS-buffered paraformaldehyde for 15 min at room temperature and rinsed with PBS. Adherent *C. albicans* were stained with Calcofluor white (10 μg ml$^{-1}$ in 100 mM TRIS·HCl pH 9.5) for 30 min at room temperature. Samples were rinsed with Milli-Q-water and mounted with ProLong Gold Antifade Mountant (Invitrogen). Images were acquired with a Zeiss Axio Observer Z1 microscope. Adherence was measured by counting the number of *C. albicans* cells in a defined area.

### Filamentation and cell invasion assay

To evaluate filamentation in serum-containing medium, $5 \times 10^4$ yeast cells were seeded in 0.5 ml F12 medium per well in a 24-well plate and incubated for 2 h at 37 °C, 5% $CO_2$, before fixation with 4% PBS-buffered paraformaldehyde. To measure *C. albicans* filamentation on and invasion of keratinocytes, monolayers of IMOK cells in 24-well plates prepared as described above were infected with $1 \times 10^5$ yeast cells per well and incubated for 3.5 h at 37 °C, 5% $CO_2$. Extracellular parts of fungal cells were then stained with 25 μl ml$^{-1}$ ConA for 20 min at room temperature. After fixation with 4.2% formaldehyde for 20 min at room temperature, intracellular and extracellular fungal parts were stained with an FITC-conjugated polyclonal anti-*C. albicans* antibody (Meridian BioScience) overnight at 4 °C. Images were acquired with an EVOS FL Auto Microscope and analyzed with ImageJ software[65]. The percentage of invasive hyphae was calculated relative to the total number of hyphae per imaged field. The proportion of hyphae inside keratinocytes was determined by calculating the ratio of the length of green-only hyphae to the entire invasive filament length. Only individual (non-aggregated) filaments were assessed.

### Cell damage assay

LDH release from TR146 and IMOK keratinocytes in response to *C. albicans* was performed as described[66]. Briefly, keratinocyte monolayers in 96-well plates prepared as described above were infected with the indicated number of yeast cells per well and incubated for 24 h at 37 °C, 5% $CO_2$, before assessing LDH release using the LDH cytotoxicity kit (Roche) according to the manufacturer's instructions. Control wells were incubated with medium only or with 1% Triton-X-100 to determine 0% and 100% damage, respectively.

### Animals

Female WT C57BL/6j mice were purchased from Janvier Elevage. The *Il17rc*$^{-/-}$ mice (C57BL/6j background) were obtained from Amgen and bred at the Institute of Laboratory Animals Science (University of Zurich). All mice were kept in individually ventilated cages under specific pathogen-free conditions at 21–24 °C, 40–60% humidity and a standard light cycle (12 h:12 h) and were provided with unrestricted access to water and food (irradiated vitamin-fortified maintenance extrudate, Kliba Nafag 3435). Animals were used at 6–14 weeks of age in sex- and age-matched groups. Animals were allowed to acclimatize for 1 week after arrival in the animal experimentation unit of Laboratory Animals Science before the experiments started. Only animals in good health were included in experiments. Experiments with WT mice used a randomized design; experiments with genetically modified female and male *Il7rc*$^{-/-}$ and *Il17rc*$^{+/-}$ littermates were not done fully randomized owing to variable distribution of the different genotypes in each litter. Colonized and uncolonized animals were kept separately to avoid cross-contamination. Sample size was chosen using Fermi's approximation and based on experience.

### Ethics statement

All mouse experiments in this study were conducted in strict accordance with the guidelines of the Swiss Animals Protection Law and were performed under the protocols approved by the Veterinary Office of the Canton Zurich, Switzerland (license numbers ZH167/2018, ZH141/2021 and ZH186/2024). All efforts were made to minimize suffering and ensure the highest ethical and humane standards according to the replace, reduce, refine) principles[67].

### Oral colonization of mice with *C. albicans*

Mice were infected sublingually as described[68], without immunosuppression or antibiotics unless stated explicitly, with $2.5 \times 10^6$ (normal infection dose) or with $5 \times 10^4$ (l.d.) *C. albicans* yeast cells. In competition experiments, *C. albicans* strains were mixed at a ratio of 1:1 (with $1.25 \times 10^6$ yeast cells of each strain) or 1:50 (with $5 \times 10^4$ 101$^{wt\_mCherry}$ and $2.5 \times 10^6$ yeast cells of the designated strain). In brief, mice were anaesthetized by injection of 100 mg kg$^{-1}$ ketamine and 20 mg kg$^{-1}$ xylazine in sterile saline intraperitoneally administered in two doses. *C. albicans* was administered by depositing a 2.5-mg cotton ball that was soaked in 100 μl yeast cell suspension under the tongue for 75–90 min. Mice were kept on a heating mat at 35–37 °C during the entire period of anaesthesia, administered with 10 ml kg$^{-1}$ sterile saline to stabilize the circulation and treated with vitamin A ointment to avoid desiccation of the eyes. For experiments with 101$^{tetOff\_ECE1}$, mice were treated with doxycycline (2 mg ml$^{-1}$) in 1% glucose drinking water for the indicated time. To induce dysbiosis, mice were treated with ampicillin (0.5 mg ml$^{-1}$) added to the drinking water that was also supplemented with 1% glucose starting 3–4 days before fungal association until the end of the experiment. The drinking water of the control group was supplemented with 1% glucose.

In some experiments, mice were perfused 2 h after fungal association, and tongues were removed and directly placed into a 6-well plate that we converted into a humid chamber with moist tissues and incubated at 37 °C, 5% $CO_2$, for 6–22 h, as indicated. Humidification was sufficient to maintain the integrity and moisture of the tissue. No ampicillin was added to the humid chamber.

In some experiments, the cornified layer of the stratified epithelium was experimentally disrupted by repeated shallow puncturing with a 26-G needle before fungal association.

### Determination of fungal burden

Tongues were collected from euthanized animals and homogenized in sterile water supplemented with 0.05% NP40 in $H_2O$ and a 5-mm steel ball for 3 min at 25 Hz using a Tissue Lyzer (Qiagen). Serial dilutions were plated on YPD agar containing 100 μg ml$^{-1}$ ampicillin, and colony-forming units (CFU) per gram tissue were enumerated. In the competition experiments, mCherry-positive colonies were identified using the IVIS Lumina III from Perkin Elmer.

### Histology

HEE and tongue tissue were fixed in 4% PBS-buffered paraformaldehyde overnight and embedded in paraffin. Sagittal sections (9 μm) were stained with periodic acid–Schiff (PAS) reagent and counterstained with haematoxylin and mounted with Pertex (Biosystem) according to standard protocols. Images were acquired with a digital slide scanner (NanoZoomer 2.0-HT, Hamamatsu) and analyzed with NDP.view2. To determine the localization of *C. albicans* in the tongue tissue, accumulations of fungal elements spanning a maximum diameter of 20 μm were considered a unit. Histology images were scored blinded.

### Quantification of tongue neutrophils and monocytes by flow cytometry

Tongues were collected from euthanized and perfused mice, cut into fine pieces and digested with DNase I (200 μg ml$^{-1}$, Roche) and Collagenase IV (4.8 mg ml$^{-1}$, Invitrogen) in PBS for 45 min at 37 °C. Single-cell suspensions were passed through a 70-μm strainer and stained in ice-cold PBS supplemented with 1% FCS, 5 mM EDTA and 0.02% $NaN_3$ with LIVE/DEAD Fixable Near-IR Stain (Life Technologies, 1/1,000 dilution) and fluorochrome-conjugated antibodies against

mouse CD45.2-AlexaFluor 700 (clone 104, Biolegend 109822, used at 1/200 dilution), CD11b-PE-Cy7 (clone M1/70, Biolegend 101216, used at 1/200 dilution), CD3-PE-Cy5 (clone 145-2C11, Biolegend 100310, used at 1/250 dilution), Ly6G-Pacific Blue (clone 1A8, Biolegend 127612, used at 1/250 dilution), Ly6C-FITC (clone AL-21, BD Biosciences 553104, used at 1/200 dilution), CD64-APC (clone X54-5/7.1, Biolegend 139306, used at 1/100 dilution) and F4/80-PE (clone BM8, Biolegend 123109, used at 1/100 dilution). Cells were fixed in 4.2% formaldehyde before acquisition on a SP6800 Spectral Analyzer (Sony). Data analysis was performed using FlowJo software v10 (FlowJo LLC). The gating of the flow cytometric data was performed according to the guidelines for the use of flow cytometry and cell sorting in immunological studies[69] including pre-gating on viable and single cells for analysis. A defined number of counting beads (BD Bioscience, Calibrite Beads), which were added to the samples before flow cytometric acquisition, were used to quantify absolute cell numbers.

### Isolation of fungal RNA from infected keratinocytes and HEEs

Monolayers of TR146 or IMOK cells in 24-well plates prepared as described above were infected with $1 \times 10^5$ yeast cells per well. After 0, 2, 4 and 24 h of incubation at 37 °C, 5% $CO_2$, the medium was removed and cells were frozen on liquid nitrogen and stored at −20 °C until further processing. *C. albicans* in serum-containing medium was manipulated the same way, except that no keratinocytes were seeded. Infected HEEs (see above) were homogenized in Tri Reagent with a 5-mm steel ball using a Tissue Lyzer (Qiagen) for 6 min at 25 Hz. RNA was then isolated with the RNeasy Mini Kit (Qiagen). Briefly, to break up fungal cell walls, samples were homogenized with 0.5-mm acid-washed glass beads (Sigma) using a Tissue Lyzer (Qiagen) for two cycles of 2 min at 30 Hz interrupted by 30-s cooling periods on ice. Cell debris was removed by centrifugation. One volume of 75% ethanol was admixed, and each sample was transferred to an RNeasy spin column. The loaded columns were washed using RW1 buffer and RPE buffer. RNA was eluted in 30 μl of RNAse-free water.

### RNA isolation from colonized tongue tissue

Total RNA (including fungal RNA) from colonized murine tongues was isolated with TRI Reagent. The tissue was homogenized in TRI Reagent with a 0.5-mm steel ball using a Tissue Lyzer for 6 min at 25 Hz. Fungal cell walls were broken up with acid-washed glass beads (0.5 mm) using a Tissue Lyzer for two cycles of 2 min at 30 Hz interrupted by a 30-s resting period on ice. RNA isolation was then pursued according to the manufacturer's instructions.

### RT-qPCR

cDNA was generated using RevertAid reverse transcriptase (Thermo Fisher). Quantitative PCR was performed using SYBR green (Roche) and an Applied Biosystems 7500 Fast instrument (Thermo Fisher Scientific). The primers used in this study to measure the expression of *C. albicans* and murine genes are listed in Supplementary Tables 3 and 4, respectively. All qPCRs were performed in duplicates. The relative expression (rel. expr.) of each *C. albicans* gene was determined after normalization to *ACT1* transcript levels while the rel. expr. of each murine gene was normalized to *ACTB* transcript levels. Data were analyzed using ABI7500 software v2.3.

### RNAseq data analysis

To determine differentially expressed genes in the colonized tongues of C57Bl/6 WT mice over time, we explored a published dataset available via National Center for Biotechnology Information (NCBI) BioProject accession number PRJNA491801 (ref. 31). To determine IL-17-dependent regulation of selected genes of interest, we explored an RNAseq dataset from *Il17rc*$^{-/-}$ and *Il17rc*$^{+/-}$ littermate control animals colonized with strain 101$^{WT}$ for 3 days (NCBI Gene Expression Omnibus (GEO) repository, accession number GSE280210) using Illumina bcl2fastq

Conversion and SUSHI app of the Functional Genomics Centre Zurich. Heat maps were generated using GraphPad prism v10.

### Statistical analysis

No statistical methods were used to predetermine sample sizes, but our sample sizes are similar to those reported in previous publications[19,20,30]. Data collection and analysis were not performed blind to the conditions of the experiments, unless specified. All statistical analyses were done with GraphPad Prism software v10. Normality was always calculated using the D'Agostino and Pearson test, Anderson–Darling test, Shapiro–Wilk test and Kolmogorov–Smirnov test. If all samples passed at least one normality test, they were considered to have normal distribution. The statistical test was applied depending on the normality result, variables and number of groups. Unpaired or paired two-tailed *t*-test or Mann–Whitney test was used for parametric or non-parametric datasets, respectively, when two groups were compared. One-way ANOVA with Sidák multiple comparisons was used when comparing more than two groups. Two-way ANOVA with Sidák or Dunnet's multiple-comparison or Fisher's LSD test was used when comparing more than two sets of variables. Correction for repeated measures was applied for time course experiments. Comparison of matched values was conducted for competition experiments. Data presented on a logarithmic scale underwent log transformation before statistical evaluation. For datasets containing suspected outliers, an outlier analysis was performed.

### Reporting summary

Further information on research design is available in the Nature Portfolio Reporting Summary linked to this Article.

## Data availability

The data that support the findings of this study are publicly available via Zenodo at https://doi.org/10.5281/zenodo.16734135 (ref. 70). RNAseq datasets used in this study are available at NCBI (BioProject PRJNA491801 and GEO repository GSE280210).

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

## Acknowledgements

We thank H.-D. Beer for support with establishing HEE cultures, the staff of the Laboratory Animal Service Center of University of Zürich for support with animal husbandry, the staff of the Laboratory for Animal Model Pathology of University of Zürich for support with histology studies, the Center for Clinical Studies (ZKS) for access to equipment, and members of the LeibundGut lab, Jakob Sprague and Lydia Kaspar for helpful advice and discussions. This study was supported by the Swiss National Science Foundation (grant number CRSII5_173863 to S.L.L., D.S. and C.d'E.), the European Union's Horizon 2020 research and innovation programme under the Marie Sklodowska-Curie action, Innovative Training Network: FunHoMic (grant number 812969 to S.L.L., B.H. and C.d'E.), the Novartis Foundation for Medical-Biological Research (grant number 22C224 to S.L.L.), the Uniscientia Foundation (grant number 206-2023 to S.L.L. and J.L.) and a UZH Candoc grant (to R.F.-M.). B.H. and T.B.S. were supported by the German Research Foundation (Deutsche Forschungsgemeinschaft (DFG)) within the Cluster of Excellence 'Balance of the Microverse', under Germany's Excellence Strategy, EXC 2051, project ID 390713860; B.H. was further supported by the Collaborative Research Centre/Transregio 124 'FungiNet' (DFG project number 210879364, projects C1 and C2) of the German Research Foundation. Work in the laboratory of C.d'E. is supported by the Agence Nationale de Recherche (ANR-10-LABX-62-IBEID).

## Author contributions

R.F.-M. and S.L.-L. designed the study and wrote the paper. R.F.-M. performed most of the experiments and analyzed the data. J.L., K.M.d.S.V., S. Mertens and T.B.S. performed selected experiments. M.S. and I.K. helped with the experiments. T.B.S., O.E., S. Mogavero and B.H. generated *C. albicans* mutants, provided advice and edited the paper. N.S., S.B.-B. and C.d'E. constructed the 101^OE_EED1 mutant. D.S. generated fluorescent variants of *C. albicans* mutants and *ECE1* overexpression mutants. S.L.-L. oversaw the study design and data analysis and acquired funding. All authors discussed the results and commented on the paper.

## Competing interests

The authors declare no competing interests.

## Additional information

**Extended data** is available for this paper at https://doi.org/10.1038/s41564-025-02122-4.

**Correspondence and requests for materials** should be addressed to Bernhard Hube or Salomé LeibundGut-Landmann.

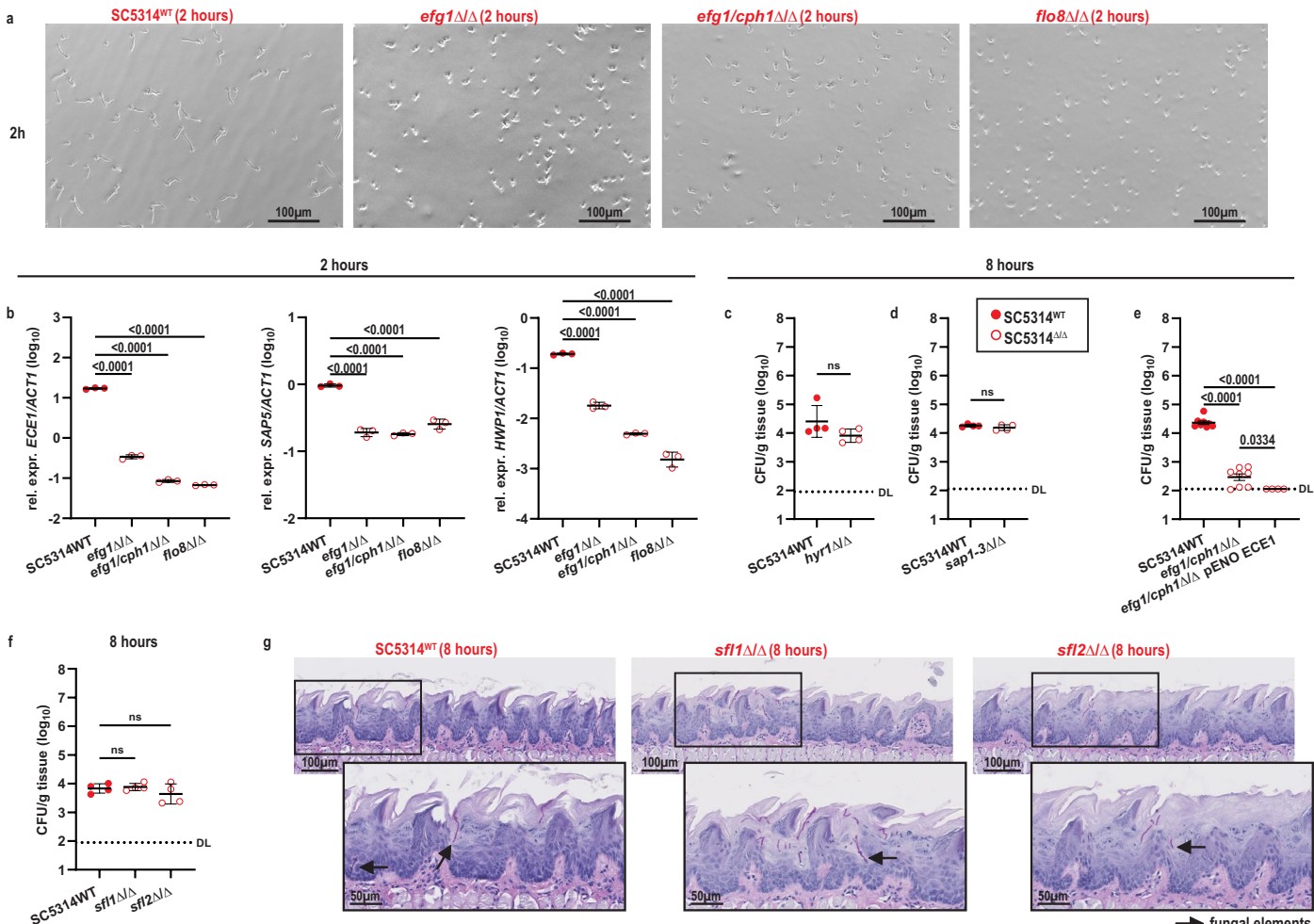

**Extended Data Fig. 1 | Filamentation and hyphae-associated genes are required for *C. albicans* oral colonization. a**–**b**. SC51314*efg1Δ/Δ*, SC51314*efg1/cph1Δ/Δ*, SC51314*flo8Δ/Δ* and the parental SC5314*WT* strain were incubated for 2 hours in serum containing medium and assessed for filamentation (**a**, scale bar, 100 μm) and expression of *ECE1*, *HWP1* and *SAP5* transcripts (**b**); n = 3 / group; mean ± SD. **c**–**g**. C57BL/6 WT mice were associated with *C. albicans* strain SC51314*hyr1Δ/Δ* (**c**), SC51314*sap1-3Δ/Δ* (**d**), SC51314 *efg1/cph1Δ/Δ* and SC51314 *efg1/cph1Δ/Δ_pENO_ECE1* (**e**), or

SC51314*sfl1Δ/Δ*, and SC51314*sfl2Δ/Δ* (**f**–**g**) in comparison to their parental SC5314*WT* strain for 8 hours and analyzed for tongue CFU (**c**–**f**) and histology (**g**, PAS-stained tongues sections, scale bar, 100 μm; black arrows indicate fungal elements); n = 4 / group. DL, detection limit. Each data point represents an independent biological sample (**a**, **b**) or an individual mouse (**c**–**f**) with mean/group ± SD per group. The statistical significance of differences between groups was determined by unpaired two-tailed *t*-test (**c**, **d**) or ordinary One-way ANOVA (**b**, **e** and **f**).

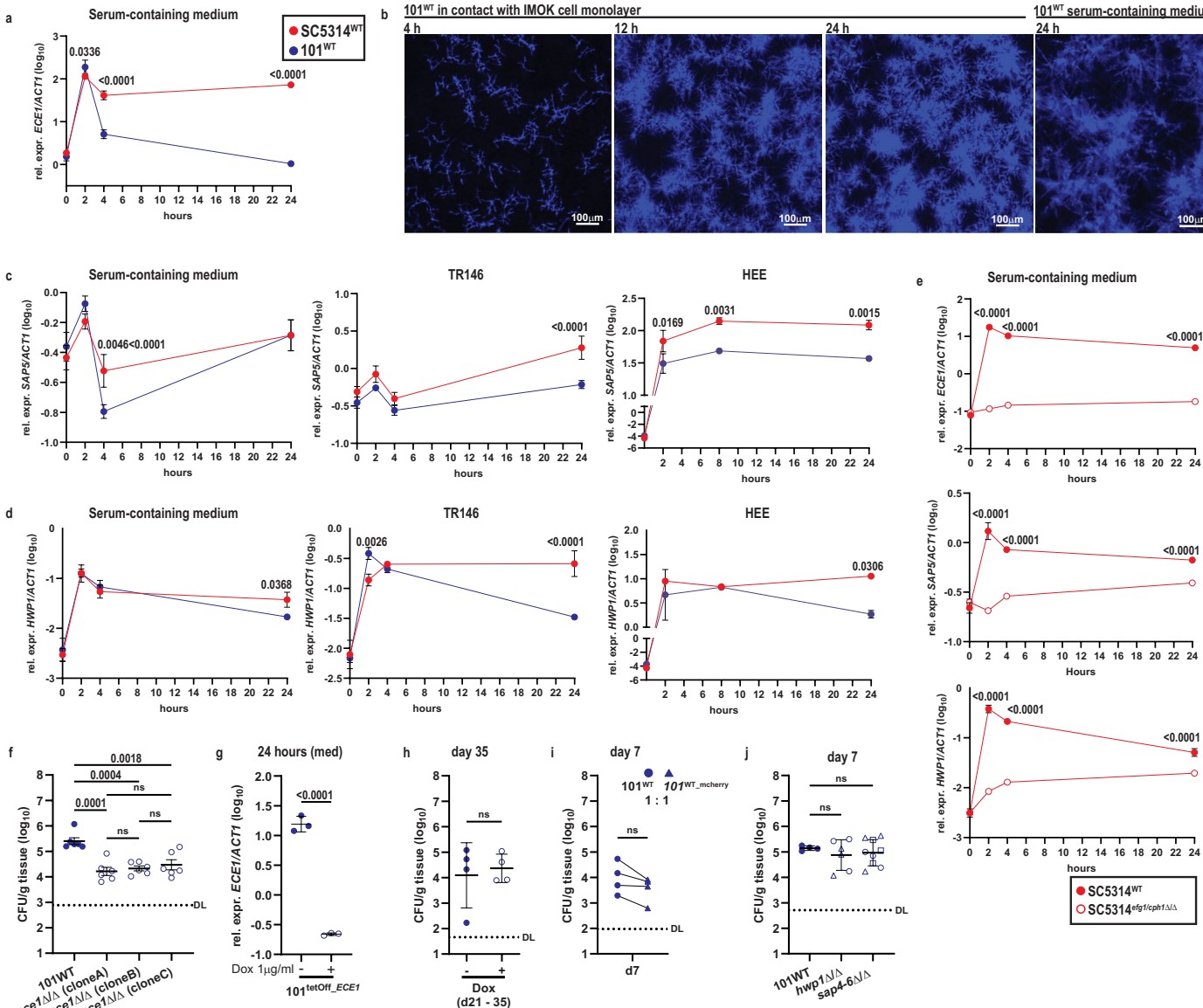

**Extended Data Fig. 2 | Low virulent *C. albicans* acquires hallmarks of virulence for the establishment of oral colonization. a.** *ECE1* expression by SC5314[WT] and 101[WT] incubated in serum containing medium for 0, 2, 4 and 24 h. **b.** CFW staining of *C. albicans* strain 101 grown on IMOK cells or in serum-containing medium for 4 h, 8 h or 24 h. Scale bars, 100 μm. **c–d** *SAP5* (**c**), and *HWP1* expression (**d**) by SC5314[WT] and 101[WT] incubated in serum containing medium (left panels in **c–d**), on TR146 keratinocytes (middle panels in **c–d**) or on human epidermal equivalents (HEE, right panels in **c-d**) for 0, 2, 4 and 24 hours as indicated. **e.** *ECE1* (top)*, SAP5* (middle), and *HWP1* expression (bottom) by SC5314[WT] and SC5314[efg1/cph1Δ/Δ] incubated in serum-containing medium for 0, 2, 4 and 24 hours. Each symbol from **a**–**e** represents the mean ± SD of triplicate wells. Data are representative of at least 2 independent experiments. **f.** Tongue CFU of C57Bl/6 WT mice associated with 3 different clones of 101[ece1Δ/Δ] (n = 6 / group ± SEM) at 24 hours. **g.** Loss of *ECE1* expression by 101[tetOff_ECE1] upon doxycycline treatment

(1 μg/ml) for 24 hours in serum containing medium; n = 3 / group with mean ± SD. **h**. Tongue CFU of C57Bl/6 WT mice that were associated with 101[WT] for 35 days and treated or not with doxycycline from day 21 to the end of the experiment; n = 4 / group with mean ± SD. **i.** Competitive colonization of 101[WT] and 101[WT_mCherry] at a 1:1 ratio for 7 days before enumerating fluorescent and non-fluorescent tongue CFU; n = 4. Connected symbols are 101[WT] and 101[WT_mCherry] CFUs per individual animal. **j.** C57BL/6 WT mice were associated with strain 101[hwp1Δ/Δ] (n = 6, 2 independent clones indicated by circles and triangles) or 101[sap4-6Δ/Δ] (n = 9, 3 independent clones indicated by circles, triangles and squares) in comparison to the parental strain 101 (n = 4) and tongue CFU (mean ± SD) were enumerated on day 7. The statistical significance of differences between groups was determined by Two-way ANOVA (**a**, **c**–**e**), One-way ANOVA (**f**, **j**), unpaired two-tailed *t*-test (**g**, **h**), paired two-tailed *t*-test (**i**).

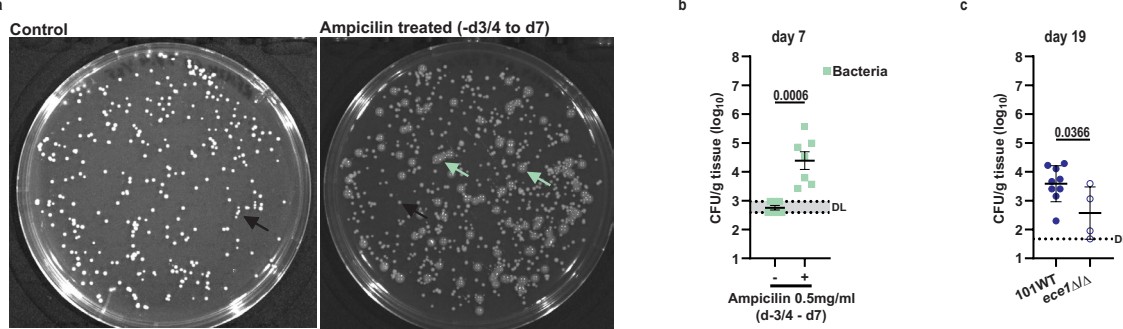

**Extended Data Fig. 3 | Candidalysin confers resistance to host antifungal responses. a–b.** C57Bl/6 mice were or were not treated with ampicillin from day -4/-3 prior to fungal association. Tongue bacteria growing on YPD agar was assessed at day 7; n = 7 pooled from 2 independent experiments with mean ± SEM. **c.** *Il17rc*[+/−] mice were associated with strain 101[WT] or strain 101[*ece1Δ/Δ*] and tongue CFU were enumerated on day 19. n = 4 or 9 with mean ± SD. The statistical significance of differences between groups was determined by paired two-tailed *t*-test (**b**) or unpaired two-tailed *t*-test (**c**).

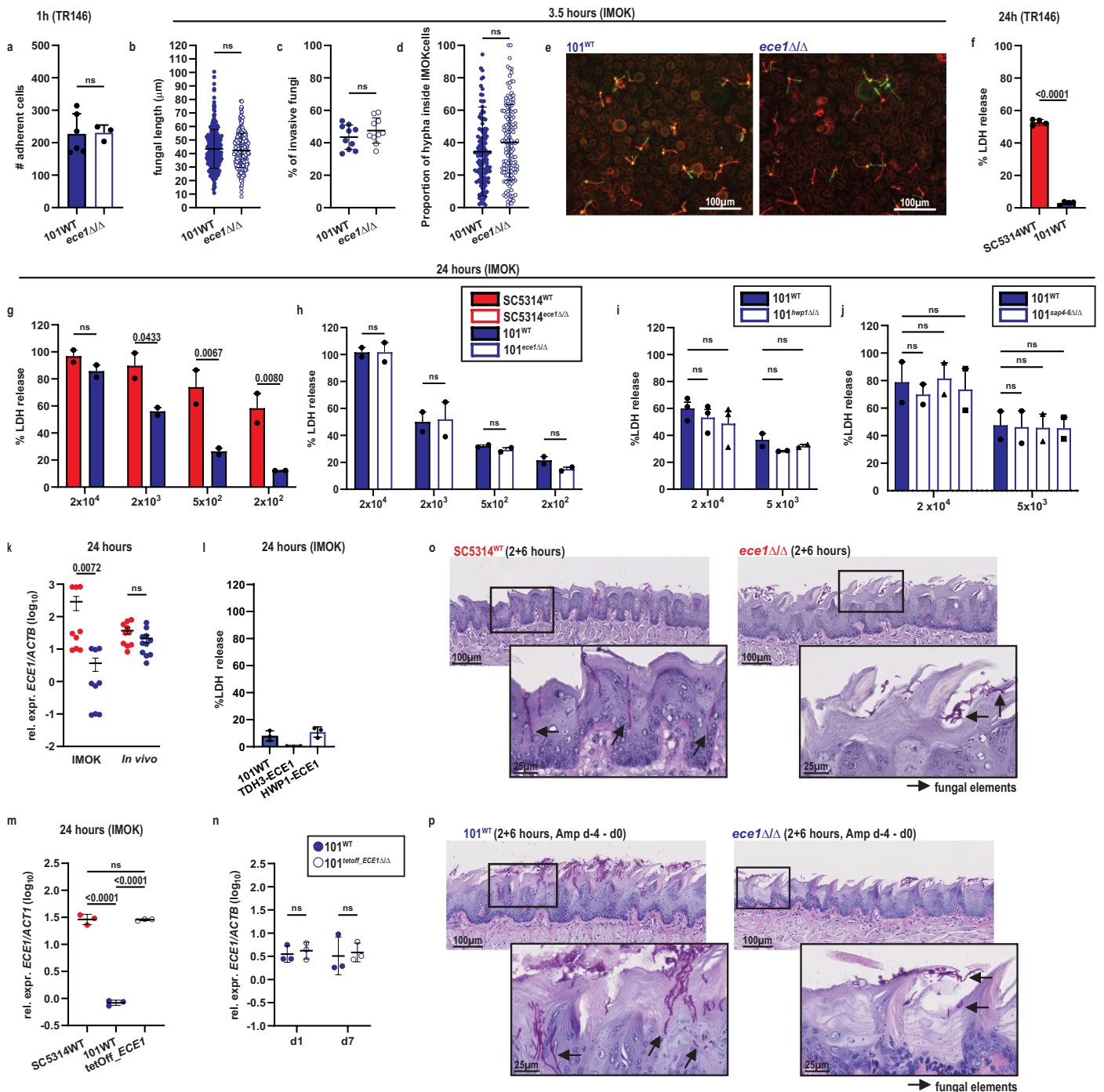

**Extended Data Fig. 4 | See next page for caption.**

**Extended Data Fig. 4 | Candidalysin promotes oral colonization by enabling *C. albicans* to overcome the barrier of the stratum corneum. a.** Adhesion of *C. albicans* 101$^{WT}$ and 101$^{ece1\Delta/\Delta}$ to TR146 keratinocytes at 1 hour post infection. **b**–**e.** *C. albicans* 101$^{WT}$ and 101$^{ece1\Delta/\Delta}$ were incubated on IMOK cells for 3.5 hours and stained with concanavalin A (extracellular parts of *C. albicans*, red) and with FITC-labelled anti-*C. albicans* (intra- and extracellular parts, green). Total filament length (**b**), % of invading fungi (**c**, % of fungal elements that are at least partially stained green in **e**) and % of fungal elements inside IMOK cells (**d**, proportion of the stained green part of fungal element within all, red and green stain, of the respective invasive filament) were quantified. Representative images are shown in (**e**). Scale bars, 100 µm. In A, each symbol represents the number of adherent fungal cells in a defined area with mean ± SD per group. In **b** and **d**, each symbol represents the measurement of an individual filament with mean / group ± SD. In **c**, each symbol is the percentage of all invading filaments per image with mean / group ± SD. Data in **b**-**e** are representative of two independent experiments. **f.** LDH release from TR146 keratinocytes in response to SC5314$^{WT}$ or 101$^{WT}$ (2 x 10$^4$ yeast cells / well) at 24 hours (n = 4 / group ± SD). Each symbol represents an individual well. **g**–**J.** LDH release from IMOK cells in response to SC5314$^{WT}$ and 101$^{WT}$ (**g**), 101$^{WT}$ and 101$^{ece1\Delta/\Delta}$ (**h**) or 101$^{WT}$ and 101$^{hwp1\Delta/\Delta}$ (2 different clones indicated by circles and triangles) (**i**), or 101$^{WT}$ and 101$^{sap4-6\Delta/\Delta}$ (3 different clones indicated by circles, triangles and squares) (**j**) at the indicated yeast cell numbers per well for 24 hours with mean ± SD. In **g**–**j**

each symbol represent the mean (3-6 technical replicates). **g**-**j** are data pooled from 2 independent experiments. **k.** Relative *ECE1* expression by SC5314 and 101 after 24 hours of exposure to IMOK keratinocytes or murine tongue tissue (in vivo) underlying the data shown in Fig. 3b, the mean ± SEM is indicated. Each symbol represents an individual well (IMOK) or an individual mouse (in vivo). **l.** LDH release from IMOK cells in response to 101$^{WT}$, 101$^{TDH3-ECE1}$ and 101$^{HWP1-ECE1}$ at 24 hours (n = 3 / group). Each symbol represents an individual well, the mean / group ± SD is indicated. Each symbol represents an individual well. **m**–**n.** Relative *ECE1* expression by SC5314 $^{WT}$, 101$^{WT}$ and 101$^{ECE1-OE}$ exposed to IMOK cells for 24 hours (**m**, n = 3 /group ± SD) or in the tongue tissue after 1 day or 7 days after colonization (**n**, n = 3 /group ± SD). While in M each symbol represents an individual well, in N each symbol represents an individual mouse. **o.** Representative PAS-stained tongue tissue sections of C57Bl/6 WT mice that were associated with SC5314$^{WT}$ or SC5314$^{ece1\Delta/\Delta}$ for 2 hours followed by 6 hours *ex vivo* incubation of the tissue as in Fig. 3. **p.** Representative PAS-stained tongue tissue sections of ampicillin-pretreated C57Bl/6 WT mice that were associated with 101$^{WT}$ and 101$^{ece1\Delta/\Delta}$ for 2 hours followed by 6 hours *ex vivo* incubation of the tissue as in Fig. 3. Scale bars in **o** and **p**, 100µm (inserts: 25µm). Black arrows indicate fungal elements. The statistical significance of differences between groups was determined by unpaired two-tailed *t*-test (**a**, **c**, **f**), two-sided Mann-Whitney test (**b** and **d**) or Two-way ANOVA (**g**–**k**) or ordinary One-way ANOVA (**m**).

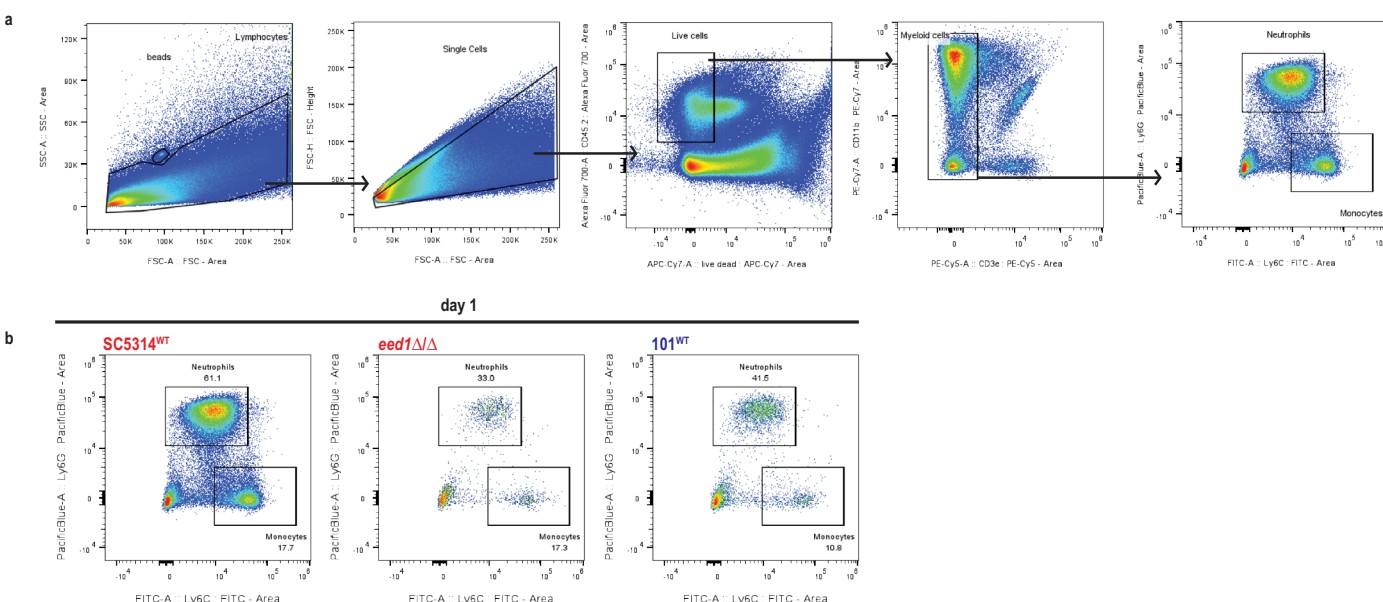

**Extended Data Fig. 5 | Tight regulation of *ECE1* expression ensures commensalism by evading the inflammatory immune response. a–b**. Gating strategy used to quantify neutrophils and monocytes in the colonized tongue (**a**) and representative plots for each fungal strain (**b**) shown in Fig. 4e.

# Reporting Summary

## Statistics

For all statistical analyses, confirm that the following items are present in the figure legend, table legend, main text, or Methods section.

| n/a | Confirmed | |
|---|---|---|
| ☐ | ☒ | The exact sample size (*n*) for each experimental group/condition, given as a discrete number and unit of measurement |
| ☐ | ☒ | A statement on whether measurements were taken from distinct samples or whether the same sample was measured repeatedly |
| ☐ | ☒ | The statistical test(s) used AND whether they are one- or two-sided<br>*Only common tests should be described solely by name; describe more complex techniques in the Methods section.* |
| ☒ | ☐ | A description of all covariates tested |
| ☐ | ☒ | A description of any assumptions or corrections, such as tests of normality and adjustment for multiple comparisons |
| ☐ | ☒ | A full description of the statistical parameters including central tendency (e.g. means) or other basic estimates (e.g. regression coefficient) AND variation (e.g. standard deviation) or associated estimates of uncertainty (e.g. confidence intervals) |
| ☐ | ☒ | For null hypothesis testing, the test statistic (e.g. $F$, $t$, $r$) with confidence intervals, effect sizes, degrees of freedom and $P$ value noted<br>*Give P values as exact values whenever suitable.* |
| ☒ | ☐ | For Bayesian analysis, information on the choice of priors and Markov chain Monte Carlo settings |
| ☒ | ☐ | For hierarchical and complex designs, identification of the appropriate level for tests and full reporting of outcomes |
| ☒ | ☐ | Estimates of effect sizes (e.g. Cohen's *d*, Pearson's *r*), indicating how they were calculated |

*Our web collection on statistics for biologists contains articles on many of the points above.*

## Software and code

Policy information about availability of computer code

| | |
|---|---|
| Data collection | RNAsequencing: Illumina NovaSeq 6000; RTqPCR: ABI7500Fast; microscopy (in vitro assays): EVOS FL Auto Microscope; scanning of histology slides: NanoZoomer 2.0-HTdigital scanner; flow cytometry: Sony Spectral Analyzer SP6800 |
| Data analysis | GraphPad Prism v10 was used to create graphs and performed statistical analyses; FlowJo v10 was used to analyse flow cytometry data; ImageJ was used to analyse tmicroscopy images of in vitro assays; ABI7500 software v2.3 was used to analyse gene expression data; NDP. view2 was used to analyse histology images; Illumina bcl2fastq Conversion and SUSHI app of the Functional Genomics Centre Zurich were used to analysed RNAsequecing data. |

For manuscripts utilizing custom algorithms or software that are central to the research but not yet described in published literature, software must be made available to editors and reviewers. We strongly encourage code deposition in a community repository (e.g. GitHub). See the Nature Portfolio guidelines for submitting code & software for further information.

## Data

Policy information about [availability of data](availability of data)

All manuscripts must include a [data availability statement](data availability statement). This statement should provide the following information, where applicable:

- Accession codes, unique identifiers, or web links for publicly available datasets
- A description of any restrictions on data availability
- For clinical datasets or third party data, please ensure that the statement adheres to our [policy](policy)

> The data that support the findings of this study are publicly available at zenodo.org (https://doi.org/10.5281/zenodo.16734135). RNAseq datasets used in this study are available at NCBI (BioProject PRJNA491801 and GEO repository GSE280210).

## Research involving human participants, their data, or biological material

Policy information about studies with [human participants or human data](human participants or human data). See also policy information about [sex, gender (identity/presentation), and sexual orientation](sex, gender (identity/presentation), and sexual orientation) and [race, ethnicity and racism](race, ethnicity and racism).

| | |
|---|---|
| Reporting on sex and gender | n/a |
| Reporting on race, ethnicity, or other socially relevant groupings | n/a |
| Population characteristics | n/a |
| Recruitment | n/a |
| Ethics oversight | n/a |

Note that full information on the approval of the study protocol must also be provided in the manuscript.

# Field-specific reporting

Please select the one below that is the best fit for your research. If you are not sure, read the appropriate sections before making your selection.

☒ Life sciences ☐ Behavioural & social sciences ☐ Ecological, evolutionary & environmental sciences

For a reference copy of the document with all sections, see [nature.com/documents/nr-reporting-summary-flat.pdf](nature.com/documents/nr-reporting-summary-flat.pdf)

# Life sciences study design

All studies must disclose on these points even when the disclosure is negative.

| | |
|---|---|
| Sample size | No statistical methods were used to pre-determine sample sizes but our sample sizes are similar to those reported in previous publications (ref 21, 22, 33). Sample size was chosen by the Fermi's approximation and based on experience. We generally used 3-4 mice per group and experiment. Experiments were generally repeated at least 2 times |
| Data exclusions | Datapoints determined outliers based on an outlier analysis performed with GraphPad Prism were excluded. |
| Replication | Experiments were generally repeated at least 2 times. |
| Randomization | Experiments with C57BL/6JRj WT mice used a randomized design. Experiments with genetically modified animals were not done fully randomized due to variable distribution of the different genotypes in each litter. Upon weaning, mice were caged by sex, but not genotype. Experimental groups were kept separate during the experiment to prevent fungal transmission between groups. |
| Blinding | Data collection and analysis were not performed blind to the conditions of the experiments (except for histology analyses). Mouse infections and treatments could not be carried out fully blinded. Turbidity of the fungal inoculum inadvertently informed the experimenter about the identity of the infection vs. the control inoculum. Sample collection during the experiment and ex vivo analyses such as counting of fungal colonies and analysis of histology sections were executed in a blinded way wherever possible. Ex vivo flow cytometry analyses were done in an unbiased way through acquisition of a fixed number of cells. A fixed gating strategy was applied to all samples of the experiment. |

# Reporting for specific materials, systems and methods

We require information from authors about some types of materials, experimental systems and methods used in many studies. Here, indicate whether each material, system or method listed is relevant to your study. If you are not sure if a list item applies to your research, read the appropriate section before selecting a response.

## Materials & experimental systems

| n/a | Involved in the study |
|-----|----------------------|
| ☐ | ☒ Antibodies |
| ☐ | ☒ Eukaryotic cell lines |
| ☒ | ☐ Palaeontology and archaeology |
| ☐ | ☒ Animals and other organisms |
| ☒ | ☐ Clinical data |
| ☒ | ☐ Dual use research of concern |
| ☒ | ☐ Plants |

## Methods

| n/a | Involved in the study |
|-----|----------------------|
| ☒ | ☐ ChIP-seq |
| ☐ | ☒ Flow cytometry |
| ☒ | ☐ MRI-based neuroimaging |

## Antibodies

| | |
|---|---|
| Antibodies used | CD45.2-AlexaFluor 700 (clone 104, Biolegend #109822, used at 1/200 dilution), CD11b-PE-Cy7 (clone M1/70, Biolegend #101216, used at 1/200 dilution), CD3-PE-Cy5 (clone 145-2C11, Biolegend #100310, used at 1/250 dilution), Ly6G-Pacific Blue (clone 1A8, Biolegend #127612, used at 1/250 dilution), Ly6C-FITC (clone AL-21, BD Biosciences #553104, used at 1/200 dilution), CD64-APC (clone X54-5/7.1, Biolegend #139306, used at 1/100 dilution) and F4/80-PE (clone BM8, Biolegend #123109, used at 1/100 dilution) |
| Validation | We relied on the manufacturer's specifications. |

## Eukaryotic cell lines

Policy information about cell lines and Sex and Gender in Research

| | |
|---|---|
| Cell line source(s) | TR146 were purchased from Sigma (#10032305); IMOK were provided by Ann Garett-Sinha; N/TERT1 cells were provided by Hans-Dietmar Beer. |
| Authentication | None of the cell lines were authenticated after having been found to behave as previously described in the literature. |
| Mycoplasma contamination | The cell lines were not tested for mycoplasma contamination. |
| Commonly misidentified lines (See ICLAC register) | n/a |

## Animals and other research organisms

Policy information about studies involving animals; ARRIVE guidelines recommended for reporting animal research, and Sex and Gender in Research

| | |
|---|---|
| Laboratory animals | WT C57BL/6j mice were purchased by Janvier Elevage. Il17rc−/− mice (C57BL/6j background) were obtained from Amgen (Thousand Oaks, CA) and bred at the Institute of Laboratory Animals Science (LASC, University of Zurich). All mice were kept in individually ventilated cages under specific pathogen-free conditions at 21 - 24°C, 40-60% humidity, and a standard light cycle (12h:12h) and were provided with unrestricted access to water and food (irradiated vitamin-fortified maintenance extrudate, Kliba Nafag #3435). Animals were used at 6-14 weeks of age in sex- and age-matched groups. Animals were allowed to acclimatize for 1 week after arrival in the BSL2 animal experimentation unit of LASC before starting experiments. Only animals in good health were included in experiments. Colonized and uncolonized animals were kept separately to avoid cross-contamination. |
| Wild animals | n/a |
| Reporting on sex | Experiments with WT C57BL/6Jj mice were conducted with females. Female and male Il17rc deficient and heterozygous littermate control mice were used for experiments. |
| Field-collected samples | The study didn't involve field-collected samples |
| Ethics oversight | All mouse experiments in this study were conducted in strict accordance with the guidelines of the Swiss Animals Protection Law and were performed under the protocols approved by the Veterinary office of the Canton Zurich, Switzerland (license number ZH167/2018, ZH141/2021, ZH186/2024)). All efforts were made to minimize suffering and ensure the highest ethical and humane standards according to the 3R principles |

Note that full information on the approval of the study protocol must also be provided in the manuscript.

# Plants

| | |
|---|---|
| Seed stocks | n/a |
| Novel plant genotypes | n/a |
| Authentication | n/a |

# Flow Cytometry

## Plots

Confirm that:

☒ The axis labels state the marker and fluorochrome used (e.g. CD4-FITC).

☒ The axis scales are clearly visible. Include numbers along axes only for bottom left plot of group (a 'group' is an analysis of identical markers).

☒ All plots are contour plots with outliers or pseudocolor plots.

☒ A numerical value for number of cells or percentage (with statistics) is provided.

## Methodology

| | |
|---|---|
| Sample preparation | Tongues were collected from euthanized and perfused mice, cut into fine pieces, and digested with DNase I (200 μg/ml, Roche) and Collagenase IV (4.8 mg/ml, Invitrogen) in PBS for 45 min at 37° C. Single cell suspensions were passed through a 70 μm strainer and stained in ice-cold PBS supplemented with 1 % FCS, 5 mM EDTA, and 0.02 % NaN3 with LIVE/DEAD Fixable Near-IR Stain (Life Technologies) and fluorochrome-conjugated antibodies. |
| Instrument | Sony SP6800 Spectral Analyzer |
| Software | FlowJo v10 software (FlowJo LLC) |
| Cell population abundance | The abundance of the cell populations of interest is provided in Extended Data Figure 5A-B. Absolute cell numbers per tongue are provided in Figure 5E. |
| Gating strategy | SSC-Area/FSC-Area to select for all tissue cells, FSC-Heigt/FSC-Area to select single cells, CD45.2-alexa Fluor 700-Area/ live dead APC-cy7-Area to selected live immune cells, CD11b-PE-Cy7-Area/CD3e-PE-Cy5-Area to select myeloid cells, and Ly6G-Pacific Blue-Area/Ly6C-FITC-Area to select neutrophils and monocytes. |

☒ Tick this box to confirm that a figure exemplifying the gating strategy is provided in the Supplementary Information.

