## [Peer Review File · Nature Microbiology]

Dynamic expression of candidalysin facilitates oral colonization of *Candida albicans* in mice

Corresponding Author: Professor Salomé LeibundGut-Landmann

Version 0:

Reviewer comments:

Reviewer #1

(Remarks to the Author)

In this work by Frois-Martins et al, the authors describe functions of the fungal peptide toxin - candidalysin in promoting oral commensalism by *Candida albicans*. Unlike commensalism at the intestinal mucosa, the fungal regulators and the mechanistic determinants of oral commensalism regulators are not defined, and the current work efficiently address this critical gap in our knowledge.

The authors have used clever fungal genetic approaches, two distinct *C. albicans* strains (systemic fungal isolate - SC5314, and commensal isolate - 101), and in vivo/in vitro models of fungal colonization to define candidalysin as one of the determinants that promotes fungal commensalism at the oral mucosa. Furthermore, with the use of IL17 signaling deficient mice, the authors have touched upon the roles of candidalysin in countering the host pressure.

The manuscript is well-written and the data are logically presented. Many of the conclusions are well-supported by the data. Overall, the manuscript presents a significant advance in our understanding of how *C. albicans* establishes a long-term commensalism. The authors' use of two distinct strains is commendable, as it allowed them to assess how a commensal isolate can distinctly employ the candidalysin to enable colonization. While their data and the conclusions are solid, I have only a comments, as highlighted below, which if addressed further strengthen this excellent work:

Major Points:

1. The authors posit that continuous ECE1 expression provides an intrinsic benefit for sustained colonization, while minimizing damage and excessive inflammation. On the other hand, high ECE1 expression is avoided in 101 to limit the damage, and potentially clearance. This conclusion is based on comparative analysis of SC5314 and 101, plus temporal expression analysis of ECE1 and the use of 101tetOff_ECE1 strain to knockout ECE1 in a doxycycline-dependent manner. While the data are convincing, a direct demonstration of enhanced ECE1 expression (under the 101 background) would be convincing. For example, does constitutive ECE1 expressing 101 strain, which the authors have shown to damage the cell line, exhibit enhanced clearance when used to infect the mice, similar to the SC5314 strain? Does this infection lead to excessive inflammation? Similarly, would the authors expect the 101 strain to have a competitive disadvantage relative to the parent strain when ECE1 is expressed under the constitutive promoter (e.g., their 101tetOff_ECE1 strain)? These data will increase the robustness of their findings is the authors can directly test this.
2. Related to the above point, the authors discuss that Nrg1 can potentially regulate ECE1 transcription and thus may act as one of the regulators controlling ECE1 and hence the long term commensal colonization. Would the authors expect Nrg1 KO 101 strain to get cleared rapidly?
3. The authors described Eed1 to maintain ECE1 expression and SC5314 EED1 KO to phenocopy the commensal phenotype of 101-WT. Conversely, EED1-overexpressing strain was said to recapture the pathogenic traits as judged using in vitro filamentation and ECE1 expression. Would the 101-EED1OE strain phenocopy the SC5314-WT in oral colonization experiment?
4. The authors show that Hwp1 and Sap4-6 are not required for the early tissue persistence by 101, while they are required by SC5314. Is the non-requirement of these proteins by 101 true at later timepoints as well? Do the 101 sap4-6 KO and hwp1 KO show similar fungal burden to the parent 101 at later timepoints too?
5. Related to the above point, it is possible that the Hwp1, Sap4-6, Hyr1 are not required for fungal persistence, but they may promote damage and inflammation. Does Sap4-6 or Hwp1 or Hyr1 overexpression under 101 parent or 101-ece1KO background promote higher inflammation? If yes, it is likely that their expression downregulation provides an additional mechanism to ensure long-term, non-inflammatory persistence.

Minor points:

1. It would be helpful if arrow-heads can be placed to indicate fungal elements in histological micrographs; especially for the histological micrographs for 8h timepoints, where there is a sparse amount of fungal presence.
2. For the method describing ex vivo incubation of tongues under "Oral colonization of mice with *C. albicans*", additional specific details would be helpful. For example, it would be helpful to know whether the tongues submerged in any medium or just humidification was sufficient? Were the tongues extracted and washed before placing in the plate for incubation or were directly placed?

Reviewer #2

(Remarks to the Author)

The scientific premise of this work lies in a novel overarching hypothesis that tight regulation of certain virulence genes, and in this case the candidalysin gene of *C. albicans*, play a role in commensalism, in addition to driving pathogenicity. This paper is a follow up of work in a gut model by investigators at Brown published this year in Nature. The authors use an established immunocompetent mouse oral infection model of the group led by Dr LeibundGut-Landmann at the University of Zurich. The oral model which has been consistently published by the Zurich group as a disease or oropharyngeal candidiasis (OPC) infection model, is now being used to study commensalism or colonization. However, this is done without any discernible changes in the fungal inoculation methods or the host background. The biggest concern about this work is that low-grade infection caused by strain 101, an almost avirulent strain and the model "commensal" strain used in this work, is used as a model of oral commensalism or colonization. This strain resides in the oral keratin layer of immunocompetent mice until a well-controlled but limited inflammatory response drives its clearance several days-weeks post-inoculation. This fact has been well established in two outstanding prior publications by the group (references 31 and 32), but is almost ignored in this work. Also disregarded by the group is the fact that in humans, a biopsy that shows keratin invasion by *C. albicans* in the oral mucosa is a diagnostic sign of infection, not commensalism. This is basic oral pathology taught in dental school. In the oral environment commensalism is more likely associated with mucosal *Candida* adhesion and not epithelial invasion, thus a role of candidalysin, an epithelial cell toxin, is not only surprising, and but also not convincing to this reviewer based on the data shown in this work.

More specifically:

Introduction

Lines 95-96: This sentence suggests that the commensal state in the oral cavity is consistent with "residing inside the epithelium". A reminder to the authors that *Candida* is not an obligate intracellular pathogen, so invasion of the epithelial layer is not required for survival or a commensal lifestyle, especially in the nutrient-rich oral cavity. The premise that candidalysin breaches the keratin layer to allow commensalism to take place is counter to the central role of this toxin in cell damage and induction of inflammation that is associated with disease. Throughout this paper the authors conflate commensalism with low grade infection of avirulent strains.

Results

Figures 1AB, S1CD are redundant, they show the reduced virulence (fungal burdens) of filamentation mutants in the SC5314 strain background, which involve genes with an established role in OPC pathogenesis. Also in describing these data, the authors use the term "colonization" inappropriately, since the SC5314 WT strain mounts a substantial inflammatory response in this model early in the infection process. In prior publications with this model and this strain the group refer to this as infection, not colonization.

Figure 2: The statements that there are "similar degrees of filamentation" between strains SC5314 and 101, and that strain 101 "reverted back to the yeast morphology" are not supported by the histology images or any other data. In histology images, there is an isolated hyphal cell of strain 101 and a considerable number of hyphae in strain SC5314, and at 24h strain 101 seems to have mostly a yeast or short pseudohyphal morphology, consistent with its filamentation defect reported in previous work by this group (ref 31 and 32).

Lines 143-145: "together these data establish a filamentation prerequisite for oral colonization" in fact the data are weak and run contrary to the well-established defect in filamentation of strain 101 in vitro and in vivo.

The authors show that deletion of the candidalysin gene in the 101 strain background reduces oral fungal burdens in mice. These data support the hypothesis that invasion of the superficial epithelial layers is required for the increase in fungal burdens with this strain, but does not necessarily support a role in commensalism, this is a stretch. The acute and transient expression of *ece1* also explains the reduced burdens in the *ece1* KO strain in the 101 background. However, in previous work (ref#32) the Zurich group showed that overexpression of *ece1* in this strain background does not change its phenotype in this mouse model. How can the authors reconcile this mouse work with the previously published?

Lines 211, 224-226: authors conclude that the role of *ece1* is "independent of antibiotic induced dysbiosis" and independent of the "oral microbiota". The evidence to support this statement is at best grossly inadequate. The investigators have not shown that treatment with ampicillin disrupts the oral mucosal bacterial communities in any quantifiable or qualitative manner. Typically, as in the Nature gut paper, a gnotobiotic model would be used to substantiate these claims, but at the very least bacterial CFUs should be presented in this work.

Figures 2N-O are hard to follow and are not well described in the text. It is also unclear why the authors focused on these host genes and not on other functional group clusters (eg inflammation).

Figure 3: The source of the epithelial cell lines and confirmation of their phenotype, especially the mouse cell line should be described. The OEC:Fungal cell ratios are not mentioned in the LDH assays and the justification for the use of the IMOK cell line is confusing. The investigators state that they are using the IMOK cell line because it can be damaged more easily but this rationale seems to weaken and not strengthen their conclusions with the 101 strain. Also the doxycycline-inducible strain is used in the in vitro assay (with no added doxycycline), why not use the *ece1* overexpressing strain that the group has used in the past?

Figure 3 and Lines 260-1: The authors state they "allowed the colonization process to complete ex vivo". This is an inappropriate use of a model as a "colonization" model. The authors use an ex vivo tongue model to allow the strains to complete the "colonization" that started in vivo. The ex vivo model disregards the oral defenses that modify a strain's ability to colonize the oral mucosa, including saliva, mastication, neutrophils etc. In this model again the authors claim no role of the oral microbiota, without testing the effects of ampicillin ex-vivo.

Figure 4 and the conclusions drawn by comparing the *eed1* KO strain in the SC5314 background to the WT 101 strain are very hard to follow and confusing. It is also unclear where the conclusion that *ece1* helps strain 101 evade the inflammatory response is based, when the work shown in this figure involves the *eed1* overexpressing and not the *ece1* overexpressing strain in this strain background.

If indeed *ece1* helps strain 101 establish residence in the cornified layer as the investigators posit, it would be appropriate to test whether candidalysin (among its many other functions) has keratinolytic activity. This strain is mainly localized in the acellular keratin layer and not deep in the stratified layers of the oral epithelium, thus epithelial cell damage may not be as important for "colonization" as this paper suggests.

Respectfully,
Anna Dongari-Bagtzoglou

Reviewer #3

(Remarks to the Author)

This study by Frois-Martins et al. investigates the mechanisms that promote oral colonization by the commensal and opportunistic fungal pathogen *Candida albicans*. These mechanisms are poorly understood and Frois-Martin et al. demonstrate that during commensal oral colonization, expression of factors that are usually thought of as virulence factors promotes colonization. The findings that virulence factors play a role in establishing a commensal niche extends our understanding of the role of these factors in host-*C. albicans* interaction. The authors advance the model that virulence factors evolved to promote sustained colonization of the host rather than solely to promote pathogenic interactions. These observations are extremely interesting because the *C. albicans* genome encodes large gene families of virulence factors with complex and different regulation. These gene families may now be understood as adaptations for colonization of diverse niches within the host rather than only as adaptations for virulence.

Comments for improvement of the manuscript:

1. None of the mutations are complemented. This is an essential control to demonstrate that the differences being observed are related to the gene that has been deleted, especially for new strains that the authors constructed. The authors should include a complemented strain along with the mutant strain, at least for key experiments.
2. The histology images need some kind of quantification so the reader knows how frequently the observations were seen. Scoring of fungal localization and extent of filamentation should be conducted by blinded observers and provided. For example, quantification would support the very interesting conclusion stated in line 264 that *ece1* mutants underwent "extensive filamentation" but "remained peripheral".
3. It would be helpful for non-mycologist readers if the authors added symbols (e.g. arrows/arrowheads) to point out fungal hyphae and yeast cells.
4. Growth in tissue culture or cell-free serum. What was the morphology of the cells? Is ECE1 expression lower because substantial numbers of cells have converted to yeast morphology in one strain but not the other? Cellular morphology should be quantified.
5. Before making conclusions about the role of bacterial dysbiosis (e.g. lines 12, 225) the authors need to show that their antibiotic treatment was sufficient to establish dysbiosis. Alternatively, the authors could change the language to state conclusions about the effect of antibiotic treatment.
6. Fig. 21 should include a control experiment with strain 101 without the Dox-regulated ECE1 gene to show that Dox treatment alone does not produce low colonization.

7. The competition experiments need a control experiment (competition of tagged WT versus untagged WT) to show that the tagging does not change the colonization of the strain.
8. A nice way to analyze competition data is to calculate the Competitive Index (CI) which is the ratio of strain 1 in the output divided by strain 2 in the output divided by the ratio of the strains in the input. CI values can then be compared statistically between the control experiment (tagged WT vs untagged WT) and the experiment of interest. (Auerbuch et al, *Infect Immun*;69(9):5953–5957. doi: 10.1128/IAI.69.9.5953-5957.2001, PMID 11500481.) Without this analysis, it is unclear how the statistics were calculated. Were all CFU measurements for one strain compared to all CFU measurements for the other without taking into account which values came from the same mouse?
9. A key experiment would be to introduce the Dox-regulated ECE1 construct into the eed mutant and ask whether restoring expression of ECE1 would increase long term colonization and affect the localization of the organisms.
10. Fig. 2P shows colonization results at day 19. Il17rc^{-/-} mice at day 19 should be shown for comparison.
11. Why is the ratio of ECE1 expression in SC5314/101 so low in vivo (Fig. 3B)? Is expression in SC5314 very low in vivo compared to IMOK and expression in 101 is similar in the 2 conditions? Or is expression in SC5314 the same in both conditions and expression in 101 is very high in vivo but not in IMOK? What features of the IMOK vs in vivo environment are important for these differences in expression?
12. The Ece1-independent damage to IMOK cells is very interesting. Are HWP1 or SAP4-6 needed for this damage or does this damage represent an activity that is independent of standard virulence factors?
13. Fig. 4 E,F,G what time points are shown, is Fig 4G really 24 hours? Why is SC5314 CFU so low? The legend says that the data are the same as Fig 2D, but Fig 2D shows ECE1 gene expression and Fig 4G shows CFU. The legend needs to be corrected.
14. If SC5314 CFU is very low (Fig 4G), how was it possible to measure ECE1 gene expression from these cells at day 7 (panel 4H)?

Minor comments:

1. The use of antibiotics needs to be described more clearly. The methods should state that antibiotics were not used in any of the experiments except for results shown in Fig. 2L and S3K. For S3K, were antibiotics present during the ex vivo incubation?
2. The legend for figure 2 should be corrected to explain the antibiotic experiment more fully. It sounds like all mice in Fig. 2L were treated with antibiotics.
3. Fig. 1I is difficult to see. An inset as used in later figures would be helpful.
4. Throughout the figures, many different days are being compared. It would be helpful to label the various panels with the day of colonization being shown (as in Fig. 1A-H) so that the reader can quickly understand the differences between the panels.
5. Are the statistics in Fig. 2B corrected for repeated measures? This should be mentioned in the methods.
6. How was the ex vivo incubation of tongue tissue conducted? Were the tissues incubated without any medium, submerged under medium or with an air-liquid interface? If there was medium, what medium was used and were antibiotics included?
7. Conclusions about morphology are complicated by the fact that Efg1, Cph1 and Flo8 are all transcription factors that regulate expression of genes. It is difficult to know how much a defect reflects the actual physical form of the fungus (a yeast versus a hypha) rather than the absence of proteins that decorate its surface. It would be best to modify some statements, e.g. that "filamentation is a prerequisite for oral colonization" (line 144), to take into account the possible contributions of proteins expressed by hyphae.
8. Line 144, filamentation is a prerequisite independent of fungal strain. Where was this shown for 101?
9. Line 152 candidalysin is essential... for colonization. There are a lot of colonies in Fig 1B so there is substantial candidalysin-independent colonization.
10. The terms "mucosal tissue" (line 174) and "oral mucosal tissue" (line 182) are vague since the authors have discussed mice, cell cultures and HEE. Authors should state what sample is being used. If it is mouse tongue, authors should describe whether a dissection was used to analyze just the mucosal tissue or whether they are analyzing the entire tongue.
11. Line 191, add comment that 50x less is labeled "ld" in the figure. Also define the "ld" term in the figure legend.
12. Line 194, should say inducible repression of ECE1 gene expression, not deletion.
13. Line 219, "IL-17-deficient mice". What mice are being used for the experiment? The text seems to indicate Il17rc^{-/-} mice, which are knocked out for the receptor, not for IL-17.
14. Fig. 2N,O needs more explanation. It should be stated that the results are from an RNASeq experiment.

Decision Letter:

13th January 2025

Dear Professor LeibundGut-Landmann,

Thank you for your patience while your manuscript "Dynamic expression of the fungal toxin candidalysin governs commensal oral colonization" was under peer-review at Nature Microbiology. It has now been seen by 3 referees, whose expertise and comments you will find at the end of this email. Although they find your work of some potential interest, they have raised a number of concerns that will need to be addressed before we can consider publication of the work in Nature Microbiology.

In particular, you will see that the referees have expressed concerns regarding the discrepancy over the phenotype of *ece1* expression, commensalism vs low-grade inflammation, keratinolytic activity of candidalysin, and some controls in the experiments. However, for us to consider a revised version of this manuscript, all the concerns, not just the suggested ones, will need to be addressed. We'd also require a point-by-point response to all the comments made by the referees.

Should further experimental data allow you to address these criticisms, we would be happy to look at a revised manuscript.

Please include a data availability statement as a separate section after Methods but before references, under the heading "Data Availability". This section should inform readers about the availability of the data used to support the conclusions of your study. This information includes accession codes to public repositories (data banks for protein, DNA or RNA sequences, microarray, proteomics data etc...), references to source data published alongside the paper, unique identifiers such as URLs to data repository entries, or data set DOIs, and any other statement about data availability. At a minimum, you should include the following statement: "The data that support the findings of this study are available from the corresponding author upon request", mentioning any restrictions on availability. If DOIs are provided, we also strongly encourage including these in the Reference list (authors, title, publisher (repository name), identifier, year). For more guidance on how to write this section please see: <http://www.nature.com/authors/policies/data/data-availability-statements-data-citations.pdf>

* If you have not done so already we suggest that you begin to revise your manuscript so that it conforms to our Article format instructions at <http://www.nature.com/nmicrobiol/info/final-submission>. Refer also to any guidelines provided in this letter.

When submitting the revised version of your manuscript, please pay close attention to our [href="https://www.nature.com/nature-portfolio/editorial-policies/image-integrity">Digital Image Integrity Guidelines](https://www.nature.com/nature-portfolio/editorial-policies/image-integrity) and to the following points below:

EXTENDED DATA FIGURES

Finally, please ensure that you retain unprocessed data and metadata files after publication, ideally archiving data in perpetuity,

as these may be requested during the peer review and production process or after publication if any issues arise.

Link Redacted

Note: This url links to your confidential homepage and associated information about manuscripts you may have submitted or be reviewing for us. If you wish to forward this e-mail to co-authors, please delete this link to your homepage first.

Nature Microbiology is committed to improving transparency in authorship. As part of our efforts in this direction, we are now requesting that all authors identified as 'corresponding author' on published papers create and link their Open Researcher and Contributor Identifier (ORCID) with their account on the Manuscript Tracking System (MTS), prior to acceptance. This applies to primary research papers only. ORCID helps the scientific community achieve unambiguous attribution of all scholarly contributions. You can create and link your ORCID from the home page of the MTS by clicking on 'Modify my Springer Nature account'. For more information please visit www.springernature.com/orcid.

If you wish to submit a suitably revised manuscript we would hope to receive it within 6 months. If you cannot send it within this time, please let us know. We will be happy to consider your revision, even if a similar study has been accepted for publication at Nature Microbiology or published elsewhere (up to a maximum of 6 months).

Yours sincerely,

Reviewer Expertise:

Referee #1: Commensal fungi, immunology

Referee #2: Oral candidiasis

Referee #3: Commensal fungi

Reviewer Comments:

Reviewer #1 (Remarks to the Author):

In this work by Frois-Martins et al, the authors describe functions of the fungal peptide toxin - candidalysin in promoting oral commensalism by *Candida albicans*. Unlike commensalism at the intestinal mucosa, the fungal regulators and the mechanistic determinants of oral commensalism regulators are not defined, and the current work efficiently address this critical gap in our knowledge.

The authors have used clever fungal genetic approaches, two distinct *C. albicans* strains (systemic fungal isolate - SC5314, and commensal isolate - 101), and in vivo/in vitro models of fungal colonization to define candidalysin as one of the determinants that promotes fungal commensalism at the oral mucosa. Furthermore, with the use of IL17 signaling deficient mice, the authors have touched upon the roles of candidalysin in countering the host pressure.

The manuscript is well-written and the data are logically presented. Many of the conclusions are well-supported by the data. Overall, the manuscript presents a significant advance in our understanding of how *C. albicans* establishes a long-term commensalism. The authors' use of two distinct strains is commendable, as it allowed them to assess how a commensal isolate can distinctly employ the candidalysin to enable colonization. While their data and the conclusions are solid, I have only a comments, as highlighted below, which if addressed further strengthen this excellent work:

Major Points:

1. The authors posit that continuous ECE1 expression provides an intrinsic benefit for sustained colonization, while minimizing damage and excessive inflammation. On the other hand, high ECE1 expression is avoided in 101 to limit the damage, and potentially clearance. This conclusion is based on comparative analysis of SC5314 and 101, plus temporal expression analysis of ECE1 and the use of 101tetOff_ECE1 strain to knockout ECE1 in a doxycycline-dependent manner. While the data are convincing, a direct demonstration of enhanced ECE1 expression (under the 101 background) would be convincing. For example, does constitutive ECE1 expressing 101 strain, which the authors have shown to damage the cell line, exhibit enhanced clearance when used to infect the mice, similar to the SC5314 strain? Does this infection lead to excessive inflammation? Similarly, would the authors expect the 101 strain to have a competitive disadvantage relative to the parent strain when ECE1 is expressed under the constitutive promoter (e.g., their 101tetOff_ECE1 strain)? These data will increase the robustness of their findings is the authors can directly test this.

2. Related to the above point, the authors discuss that Nrg1 can potentially regulate ECE1 transcription and thus may act as one of the regulators controlling ECE1 and hence the long term commensal colonization. Would the authors expect Nrg1 KO 101 strain to get cleared rapidly?

3. The authors described Eed1 to maintain ECE1 expression and SC5314 EED1 KO to phenocopy the commensal phenotype of 101-WT. Conversely, EED1-overexpressing strain was said to recapture the pathogenic traits as judged using in vitro filamentation and ECE1 expression. Would the 101-EED1OE strain phenocopy the SC5314-WT in oral colonization experiment?

4. The authors show that Hwp1 and Sap4-6 are not required for the early tissue persistence by 101, while they are required by SC5314. Is the non-requirement of these proteins by 101 true at later timepoints as well? Do the 101 sap4-6 KO and hwp1 KO show similar fungal burden to the parent 101 at later timepoints too?

5. Related to the above point, it is possible that the Hwp1, Sap4-6, Hyr1 are not required for fungal persistence, but they may promote damage and inflammation. Does Sap4-6 or Hwp1 or Hyr1 overexpression under 101 parent or 101-ece1KO background promote higher inflammation? If yes, it is likely that their expression downregulation provides an additional mechanism to ensure long-term, non-inflammatory persistence.

Minor points:

1. It would be helpful if arrow-heads can be placed to indicate fungal elements in histological micrographs; especially for the histological micrographs for 8h timepoints, where there is a sparse amount of fungal presence.

2. For the method describing ex vivo incubation of tongues under "Oral colonization of mice with *C. albicans*", additional specific details would be helpful. For example, it would be helpful to know whether the tongues submerged in any medium or just humidification was sufficient? Were the tongues extracted and washed before placing in the plate for incubation or were directly placed?

Reviewer #2 (Remarks to the Author):

The scientific premise of this work lies in a novel overarching hypothesis that tight regulation of certain virulence genes, and in this case the candidalysin gene of *C. albicans*, play a role in commensalism, in addition to driving pathogenicity. This paper is a follow up of work in a gut model by investigators at Brown published this year in Nature. The authors use an established immunocompetent mouse oral infection model of the group led by Dr LeibundGut-Landmann at the University of Zurich. The oral model which has been consistently published by the Zurich group as a disease or oropharyngeal candidiasis (OPC) infection model, is now being used to study commensalism or colonization. However, this is done without any discernible changes in the fungal inoculation methods or the host background. The biggest concern about this work is that low-grade infection caused by strain 101, an almost avirulent strain and the model "commensal" strain used in this work, is used as a model of oral commensalism or colonization. This strain resides in the oral keratin layer of immunocompetent mice until a well-controlled but limited inflammatory response drives its clearance several days-weeks post-inoculation. This fact has been well established in two outstanding prior publications by the group (references 31 and 32), but is almost ignored in this work. Also disregarded by the group is the fact that in humans, a biopsy that shows keratin invasion by *C. albicans* in the oral mucosa is a diagnostic sign of infection, not commensalism. This is basic oral pathology taught in dental school. In the oral environment commensalism is more likely associated with mucosal *Candida* adhesion and not epithelial invasion, thus a role of candidalysin, an epithelial cell toxin, is not only surprising, and but also not convincing to this reviewer based on the data shown in this work.

More specifically:

Introduction

Lines 95-96: This sentence suggests that the commensal state in the oral cavity is consistent with "residing inside the epithelium". A reminder to the authors that *Candida* is not an obligate intracellular pathogen, so invasion of the epithelial layer is not required for survival or a commensal lifestyle, especially in the nutrient-rich oral cavity. The premise that candidalysin breaches the keratin layer to allow commensalism to take place is counter to the central role of this toxin in cell damage and induction of inflammation that is associated with disease. Throughout this paper the authors conflate commensalism with low grade infection of avirulent strains.

Results

Figures 1AB, S1CD are redundant, they show the reduced virulence (fungal burdens) of filamentation mutants in the SC5314 strain background, which involve genes with an established role in OPC pathogenesis. Also in describing these data, the authors use the term "colonization" inappropriately, since the SC5314 WT strain mounts a substantial inflammatory response in this model early in the infection process. In prior publications with this model and this strain the group refer to this as infection, not colonization.

Figure 2: The statements that there are "similar degrees of filamentation" between strains SC5314 and 101, and that strain 101 "reverted back to the yeast morphology" are not supported by the histology images or any other data. In histology images, there is an isolated hyphal cell of strain 101 and a considerable number of hyphae in strain SC5314, and at 24h strain 101 seems to have mostly a yeast or short pseudohyphal morphology, consistent with its filamentation defect reported in previous work by this group (ref 31 and 32).

Lines 143-145: "together these data establish a filamentation prerequisite for oral colonization" in fact the data are weak and run contrary to the well-established defect in filamentation of strain 101 in vitro and in vivo.

The authors show that deletion of the candidalysin gene in the 101 strain background reduces oral fungal burdens in mice.

These data support the hypothesis that invasion of the superficial epithelial layers is required for the increase in fungal burdens with this strain, but does not necessarily support a role in commensalism, this is a stretch. The acute and transient expression of *ece1* also explains the reduced burdens in the *ece1* KO strain in the 101 background. However, in previous work (ref#32) the Zurich group showed that overexpression of *ece1* in this strain background does not change its phenotype in this mouse model. How can the authors reconcile this mouse work with the previously published?

Lines 211, 224-226: authors conclude that the role of *ece1* is "independent of antibiotically induced dysbiosis" and independent of the "oral microbiota". The evidence to support this statement is at best grossly inadequate. The investigators have not shown that treatment with ampicillin disrupts the oral mucosal bacterial communities in any quantifiable or qualitative manner. Typically, as in the Nature gut paper, a gnotobiotic model would be used to substantiate these claims, but at the very least bacterial CFUs should be presented in this work.

Figures 2N-O are hard to follow and are not well described in the text. It is also unclear why the authors focused on these host genes and not on other functional group clusters (eg inflammation).

Figure 3: The source of the epithelial cell lines and confirmation of their phenotype, especially the mouse cell line should be described. The OEC:Fungal cell ratios are not mentioned in the LDH assays and the justification for the use of the IMOK cell line is confusing. The investigators state that they are using the IMOK cell line because it can be damaged more easily but this rationale seems to weaken and not strengthen their conclusions with the 101 strain. Also the doxycycline-inducible strain is used in the in vitro assay (with no added doxycycline), why not use the *ece1* overexpressing strain that the group has used in the past?

Figure 3 and Lines 260-1: The authors state they "allowed the colonization process to complete ex vivo". This is an inappropriate use of a model as a "colonization" model. The authors use an ex vivo tongue model to allow the strains to complete the "colonization" that started in vivo. The ex vivo model disregards the oral defenses that modify a strain's ability to colonize the oral mucosa, including saliva, mastication, neutrophils etc. In this model again the authors claim no role of the oral microbiota, without testing the effects of ampicillin ex-vivo.

Figure 4 and the conclusions drawn by comparing the *eed1* KO strain in the SC5314 background to the WT 101 strain are very hard to follow and confusing. It is also unclear where the conclusion that *ece1* helps strain 101 evade the inflammatory response is based, when the work shown in this figure involves the *eed1* overexpressing and not the *ece1* overexpressing strain in this strain background.

If indeed *ece1* helps strain 101 establish residence in the cornified layer as the investigators posit, it would be appropriate to test whether candidalysin (among its many other functions) has keratinolytic activity. This strain is mainly localized in the acellular keratin layer and not deep in the stratified layers of the oral epithelium, thus epithelial cell damage may not be as important for "colonization" as this paper suggests.

Respectfully,
Anna Dongari-Bagtzoglou

Reviewer #3 (Remarks to the Author):

This study by Frois-Martins et al. investigates the mechanisms that promote oral colonization by the commensal and opportunistic fungal pathogen *Candida albicans*. These mechanisms are poorly understood and Frois-Martin et al. demonstrate that during commensal oral colonization, expression of factors that are usually thought of as virulence factors promotes colonization. The findings that virulence factors play a role in establishing a commensal niche extends our understanding of the role of these factors in host-*C. albicans* interaction. The authors advance the model that virulence factors evolved to promote sustained colonization of the host rather than solely to promote pathogenic interactions. These observations are extremely interesting because the *C. albicans* genome encodes large gene families of virulence factors with complex and different regulation. These gene families may now be understood as adaptations for colonization of diverse niches within the host rather than only as adaptations for virulence.

Comments for improvement of the manuscript:

1. None of the mutations are complemented. This is an essential control to demonstrate that the differences being observed are related to the gene that has been deleted, especially for new strains that the authors constructed. The authors should include a complemented strain along with the mutant strain, at least for key experiments.

2. The histology images need some kind of quantification so the reader knows how frequently the observations were seen. Scoring of fungal localization and extent of filamentation should be conducted by blinded observers and provided. For example, quantification would support the very interesting conclusion stated in line 264 that *ece1* mutants underwent "extensive filamentation" but "remained peripheral".

3. It would be helpful for non-mycologist readers if the authors added symbols (e.g. arrows/arrowheads) to point out fungal hyphae and yeast cells.
4. Growth in tissue culture or cell-free serum. What was the morphology of the cells? Is ECE1 expression lower because substantial numbers of cells have converted to yeast morphology in one strain but not the other? Cellular morphology should be quantified.
5. Before making conclusions about the role of bacterial dysbiosis (e.g. lines 12, 225) the authors need to show that their antibiotic treatment was sufficient to establish dysbiosis. Alternatively, the authors could change the language to state conclusions about the effect of antibiotic treatment.
6. Fig. 2I should include a control experiment with strain 101 without the Dox-regulated ECE1 gene to show that Dox treatment alone does not produce low colonization.
7. The competition experiments need a control experiment (competition of tagged WT versus untagged WT) to show that the tagging does not change the colonization of the strain.
8. A nice way to analyze competition data is to calculate the Competitive Index (CI) which is the ratio of strain 1 in the output divided by strain 2 in the output divided by the ratio of the strains in the input. CI values can then be compared statistically between the control experiment (tagged WT vs untagged WT) and the experiment of interest. (Auerbuch et al, *Infect Immun*;69(9):5953–5957. doi: 10.1128/IAI.69.9.5953-5957.2001, PMID 11500481.) Without this analysis, it is unclear how the statistics were calculated. Were all CFU measurements for one strain compared to all CFU measurements for the other without taking into account which values came from the same mouse?
9. A key experiment would be to introduce the Dox-regulated ECE1 construct into the eed mutant and ask whether restoring expression of ECE1 would increase long term colonization and affect the localization of the organisms.
10. Fig. 2P shows colonization results at day 19. Il17rc^{-/-} mice at day 19 should be shown for comparison.
11. Why is the ratio of ECE1 expression in SC5314/101 so low in vivo (Fig. 3B)? Is expression in SC5314 very low in vivo compared to IMOK and expression in 101 is similar in the 2 conditions? Or is expression in SC5314 the same in both conditions and expression in 101 is very high in vivo but not in IMOK? What features of the IMOK vs in vivo environment are important for these differences in expression?
12. The Ece1-independent damage to IMOK cells is very interesting. Are HWP1 or SAP4-6 needed for this damage or does this damage represent an activity that is independent of standard virulence factors?
13. Fig. 4 E,F,G what time points are shown, is Fig 4G really 24 hours? Why is SC5314 CFU so low? The legend says that the data are the same as Fig 2D, but Fig 2D shows ECE1 gene expression and Fig 4G shows CFU. The legend needs to be corrected.
14. If SC5314 CFU is very low (Fig 4G), how was it possible to measure ECE1 gene expression from these cells at day 7 (panel 4H)?

Minor comments:

1. The use of antibiotics needs to be described more clearly. The methods should state that antibiotics were not used in any of the experiments except for results shown in Fig. 2L and S3K. For S3K, were antibiotics present during the ex vivo incubation?
2. The legend for figure 2 should be corrected to explain the antibiotic experiment more fully. It sounds like all mice in Fig. 2L were treated with antibiotics.
3. Fig. 1I is difficult to see. An inset as used in later figures would be helpful.
4. Throughout the figures, many different days are being compared. It would be helpful to label the various panels with the day of colonization being shown (as in Fig. 1A-H) so that the reader can quickly understand the differences between the panels.
5. Are the statistics in Fig. 2B corrected for repeated measures? This should be mentioned in the methods.
6. How was the ex vivo incubation of tongue tissue conducted? Were the tissues incubated without any medium, submerged under medium or with an air-liquid interface? If there was medium, what medium was used and were antibiotics included?
7. Conclusions about morphology are complicated by the fact that Efg1, Cph1 and Flo8 are all transcription factors that regulate expression of genes. It is difficult to know how much a defect reflects the actual physical form of the fungus (a yeast versus a hypha) rather than the absence of proteins that decorate its surface. It would be best to modify some statements, e.g. that "filamentation is a prerequisite for oral colonization" (line 144), to take into account the possible contributions of proteins expressed by hyphae.
8. Line 144, filamentation is a prerequisite independent of fungal strain. Where was this shown for 101?
9. Line 152 candidalysin is essential... for colonization. There are a lot of colonies in Fig 1B so there is substantial candidalysin-

independent colonization.

10. The terms “mucosal tissue” (line 174) and “oral mucosal tissue” (line 182) are vague since the authors have discussed mice, cell cultures and HEE. Authors should state what sample is being used. If it is mouse tongue, authors should describe whether a dissection was used to analyze just the mucosal tissue or whether they are analyzing the entire tongue.

11. Line 191, add comment that 50x less is labeled” Id” in the figure. Also define the “Id” term in the figure legend.

12. Line 194, should say inducible repression of ECE1 gene expression, not deletion.

13. Line 219, “IL-17-deficient mice”. What mice are being used for the experiment? The text seems to indicate Il17rc^{-/-} mice, which are knocked out for the receptor, not for IL-17.

14. Fig. 2N,O needs more explanation. It should be stated that the results are from an RNASeq experiment.

Version 1:

Reviewer comments:

Reviewer #1

(Remarks to the Author)

In the revised manuscript, the authors have provided further experimental evidence for the requirement of the regulated expression of the fungal toxin – candidalysin – in establishing commensal colonization. Specifically, in this revised version, the authors have now provided the evidence for the functional requirements of Nrg1, Eed1 and their links to Ece1 during oral colonization. Similarly, the non-requirement of the additional hyphae-associated genes – HWP1, SAP4-6 – is also now experimentally demonstrated. These new data are robust and they effectively address the comments 2 - 5 in my original critique.

Regarding my original comment 1 in the initial critique, the 101 strain overexpressing ECE1 would have been ideal to further the author's claim that higher ECE1 expression (hence candidalysin production) is damaging, inflammatory and leads to decreased colonization. The authors have provided additional data that ECE1 under constitutive “tet-off” promoter has higher ECE1 expression (at a level equivalent to the SC5314 strain) in vitro, but somehow in vivo ECE1 is not induced during colonization of the oral mucosa, which is rather intriguing. Furthermore, with the inability of strong promoters such as pTDH3 in enhancing candidalysin-induced damage, this reviewer appreciates the technical challenge of candidalysin-induction under 101. Does other oral commensal isolates (e.g., 529L) behave similarly (i.e., do they also not allow ECE1 over-expression and candidalysin-enhanced cell damage)? Some brief discussion would be ideal for the readers. While such an approach would have further solidified the authors' claim, the experimental evidence from multiple different angles, as present, has considerable strength.

Overall, I believe that the revised version is a much stronger version and supports the role of a regulated candidalysin expression in establishing oral colonization.

Reviewer #2

(Remarks to the Author)

The authors have gone to extensive lengths to resolve what they perceive as “misunderstandings” and have done additional work which has improved the manuscript. However, the paper's central premise and authors' final conclusion that *C. albicans* depends on active penetration of the oral mucosa to establish commensal colonization, which is based on the phenotype of a single hypovirulent strain and its genetic derivatives, is conceptually flawed. *C. albicans* has several adhesins that could allow stable mucosal colonization without the need for invasion, at least in other strains. Mucosal invasion is seldom seen in healthy mucosa of humans and the references cited by the authors do not contradict this long held basic knowledge in oral pathology. It would be counterproductive for this organism to expend energy to invade tissues when it can form tight associations using adhesins such as hwp1. The work with strain 101 is extensive and scientifically sound. However, the overall conclusions regarding the commensal lifestyle of *C. albicans* should be toned down significantly, unless the authors can show that ece1 plays a similar role in other commensal strains, preferably clinical isolates.

Reviewer #3

(Remarks to the Author)

The authors have modified the paper in response to reviews and produced a stronger manuscript. Most of my comments have been appropriately addressed. A few clarifying comments added to the paper would be helpful.

1. While it is nice that the authors provided figures for the reviewers, the purpose of reviewers' comments is to improve the paper for future readers. The authors should add a comment to Methods that multiple, independently isolated clones of mutant strains were characterized and multiple clones are shown in Fig. 1 M-N. Otherwise the “2 clones/3 clones” comments in the legend will

be confusing for most readers.

2. In Methods, the authors should add the comment that histology images were scored by blinded investigators. This statement enhances the rigor of the analysis.

3. Mention/show in supplemental the morphology data for strain 101 (Figure Rev 3.2). It is very helpful to know that this strain retains filamentous morphology for up to 24 hrs.

4. Figure Rev 3.4 is also very interesting and should be added to the supplemental data to bolster the observation that damage of IMOK cells is independent of standard virulence factors.

Decision Letter:

6th June 2025

Dear Salomé,

Thank you for your patience while your manuscript "Dynamic expression of the fungal toxin candidalysin governs commensal oral colonization" was under peer-review at Nature Microbiology. It has now been seen by 3 referees. We are very interested in the possibility of publishing your study in Nature Microbiology, but would like to consider your response to these concerns in the form of a revised manuscript before we make a final decision on publication.

In particular, we would like you to tone down the claims of commensal life and mention the appropriate caveats as pointed out by R1 and R2. We don't think any experimental work is required at this stage (ECE1 activity in other commensals) which you should mention as a future line of work in the discussion. We would also recommend adding all the new figures, mentioned in the rebuttal, in the revised manuscript. Apart from that we request you to convert Supplementary Figures into Extended Data figures, so they are checked by our image proofreading service.

I think this should be a quick revision and we do not intend to go back to the referees for another round of review if the revised version appropriately addresses the comments made by the referees not just the editorially highlighted ones.

If you have not done so already please begin to revise your manuscript so that it conforms to our Article format instructions at <http://www.nature.com/nmicrobiol/info/final-submission/>

The usual length limit for a Nature Microbiology Article is six display items (figures or tables) and 3,000 words. We have some flexibility, and can allow a revised manuscript at 3,500 words, but please consider this a firm upper limit. There is a trade-off of ~250 words per display item, so if you need more space, you could move a Figure or Table to Supplementary Information.

Some reduction could be achieved by focusing any introductory material and moving it to the start of your opening 'bold' paragraph, whose function is to outline the background to your work, describe in a sentence your new observations, and explain your main conclusions. The discussion should also be limited. Methods should be described in a separate section following the discussion, we do not place a word limit on Methods.

Nature Microbiology titles should give a sense of the main new findings of a manuscript, and should not contain punctuation. Please keep in mind that we strongly discourage active verbs in titles, and that they should ideally fit within 90 characters each (including spaces).

Please include a data availability statement as a separate section after Methods but before references, under the heading "Data Availability". This section should inform readers about the availability of the data used to support the conclusions of your study. This information includes accession codes to public repositories (data banks for protein, DNA or RNA sequences, microarray, proteomics data etc...), references to source data published alongside the paper, unique identifiers such as URLs to data repository entries, or data set DOIs, and any other statement about data availability. At a minimum, you should include the following statement: "The data that support the findings of this study are available from the corresponding author upon request", mentioning any restrictions on availability. If DOIs are provided, we also strongly encourage including these in the Reference list (authors, title, publisher (repository name), identifier, year). For more guidance on how to write this section please see: <http://www.nature.com/authors/policies/data/data-availability-statements-data-citations.pdf>

To improve the accessibility of your paper to readers from other research areas, please pay particular attention to the wording of the paper's opening bold paragraph, which serves both as an introduction and as a brief, non-technical summary in about 150

words. If, however, you require one or two extra sentences to explain your work clearly, please include them even if the paragraph is over-length as a result. The opening paragraph should not contain references. Because scientists from other sub-disciplines will be interested in your results and their implications, it is important to explain essential but specialised terms concisely. We suggest you show your summary paragraph to colleagues in other fields to uncover any problematic concepts.

If your paper is accepted for publication, we will edit your display items electronically so they conform to our house style and will reproduce clearly in print. If necessary, we will re-size figures to fit single or double column width. If your figures contain several parts, the parts should form a neat rectangle when assembled. Choosing the right electronic format at this stage will speed up the processing of your paper and give the best possible results in print. We would like the figures to be supplied as vector files - EPS, PDF, AI or postscript (PS) file formats (not raster or bitmap files), preferably generated with vector-graphics software (Adobe Illustrator for example). Please try to ensure that all figures are non-flattened and fully editable. All images should be at least 300 dpi resolution (when figures are scaled to approximately the size that they are to be printed at) and in RGB colour format. Please do not submit Jpeg or flattened TIFF files. Please see also 'Guidelines for Electronic Submission of Figures' at the end of this letter for further detail.

Figure legends must provide a brief description of the figure and the symbols used, within 350 words, including definitions of any error bars employed in the figures.

When submitting the revised version of your manuscript, please pay close attention to our [href="https://www.nature.com/nature-research/editorial-policies/image-integrity">Digital Image Integrity Guidelines.](https://www.nature.com/nature-research/editorial-policies/image-integrity) and to the following points below:

EXTENDED DATA FIGURES

Please include a statement before the acknowledgements naming the author to whom correspondence and requests for materials should be addressed.

Finally, we require authors to include a statement of their individual contributions to the paper -- such as experimental work, project planning, data analysis, etc. -- immediately after the acknowledgements. The statement should be short, and refer to authors by their initials. For details please see the Authorship section of our joint Editorial policies at http://www.nature.com/authors/editorial_policies/authorship.html

* include a point-by-point response to any editorial suggestions and to our referees. Please include your response to the editorial suggestions in your cover letter, and please upload your response to the referees as a separate document.

* ensure it complies with our format requirements for Letters as set out in our guide to authors at www.nature.com/nmicrobiol/info/gta/

* state in a cover note the length of the text, methods and legends; the number of references; number and estimated final size of figures and tables

* resubmit electronically if possible using the link below to access your home page:

Link Redacted

*This url links to your confidential homepage and associated information about manuscripts you may have submitted or be reviewing for us. If you wish to forward this e-mail to co-authors, please delete this link to your homepage first.

Please ensure that all correspondence is marked with your Nature Microbiology reference number in the subject line.

Nature Microbiology is committed to improving transparency in authorship. As part of our efforts in this direction, we are now requesting that all authors identified as 'corresponding author' on published papers create and link their Open Researcher and Contributor Identifier (ORCID) with their account on the Manuscript Tracking System (MTS), prior to acceptance. This applies to primary research papers only. ORCID helps the scientific community achieve unambiguous attribution of all scholarly contributions. You can create and link your ORCID from the home page of the MTS by clicking on 'Modify my Springer Nature

account'. For more information please visit www.springernature.com/orcid.

We hope to receive your revised paper within three weeks. If you cannot send it within this time, please let us know.

Have a great weekend!

Reviewers Comments:

Reviewer #1 (Remarks to the Author):

In the revised manuscript, the authors have provided further experimental evidence for the requirement of the regulated expression of the fungal toxin – candidalysin – in establishing commensal colonization. Specifically, in this revised version, the authors have now provided the evidence for the functional requirements of Nrg1, Eed1 and their links to Ece1 during oral colonization. Similarly, the non-requirement of the additional hyphae-associated genes – HWP1, SAP4-6 – is also now experimentally demonstrated. These new data are robust and they effectively address the comments 2 - 5 in my original critique.

Regarding my original comment 1 in the initial critique, the 101 strain overexpressing ECE1 would have been ideal to further the author's claim that higher ECE1 expression (hence candidalysin production) is damaging, inflammatory and leads to decreased colonization. The authors have provided additional data that ECE1 under constitutive "tet-off" promoter has higher ECE1 expression (at a level equivalent to the SC5314 strain) in vitro, but somehow in vivo ECE1 is not induced during colonization of the oral mucosa, which is rather intriguing. Furthermore, with the inability of strong promoters such as pTDH3 in enhancing candidalysin-induced damage, this reviewer appreciates the technical challenge of candidalysin-induction under 101. Does other oral commensal isolates (e.g., 529L) behave similarly (i.e., do they also not allow ECE1 over-expression and candidalysin-enhanced cell damage)? Some brief discussion would be ideal for the readers. While such an approach would have further solidified the authors' claim, the experimental evidence from multiple different angles, as present, has considerable strength.

Overall, I believe that the revised version is a much stronger version and supports the role of a regulated candidalysin expression in establishing oral colonization.

Reviewer #2 (Remarks to the Author):

The authors have gone to extensive lengths to resolve what they perceive as "misunderstandings" and have done additional work which has improved the manuscript. However, the paper's central premise and authors' final conclusion that *C. albicans* depends on active penetration of the oral mucosa to establish commensal colonization, which is based on the phenotype of a single hypovirulent strain and its genetic derivatives, is conceptually flawed. *C. albicans* has several adhesins that could allow stable mucosal colonization without the need for invasion, at least in other strains. Mucosal invasion is seldom seen in healthy mucosa of humans and the references cited by the authors do not contradict this long held basic knowledge in oral pathology. It would be counterproductive for this organism to expend energy to invade tissues when it can form tight associations using adhesins such as hwp1. The work with strain 101 is extensive and scientifically sound. However, the overall conclusions regarding the commensal lifestyle of *C. albicans* should be toned down significantly, unless the authors can show that ece1 plays a similar role in other commensal strains, preferably clinical isolates.

Reviewer #3 (Remarks to the Author):

The authors have modified the paper in response to reviews and produced a stronger manuscript. Most of my comments have been appropriately addressed. A few clarifying comments added to the paper would be helpful.

1. While it is nice that the authors provided figures for the reviewers, the purpose of reviewers' comments is to improve the paper for future readers. The authors should add a comment to Methods that multiple, independently isolated clones of mutant strains were characterized and multiple clones are shown in Fig. 1 M-N. Otherwise the "2 clones/3 clones" comments in the legend will be confusing for most readers.

2. In Methods, the authors should add the comment that histology images were scored by blinded investigators. This statement enhances the rigor of the analysis.

3. Mention/show in supplemental the morphology data for strain 101 (Figure Rev 3.2). It is very helpful to know that this strain retains filamentous morphology for up to 24 hrs.

4. Figure Rev 3.4 is also very interesting and should be added to the supplemental data to bolster the observation that damage of IMOK cells is independent of standard virulence factors.

Version 2:

Decision Letter:

Our ref: NMICROBIOL-24113500B

7th July 2025

Dear Salomé,

Thank you for submitting your revised manuscript "Dynamic expression of the fungal toxin candidalysin governs homeostatic oral colonization" (NMICROBIOL-24113500B). I am glad to inform you that we'll publish it in principle in Nature Microbiology, pending minor revisions to comply with our editorial and formatting guidelines.

Thank you again for your interest in Nature Microbiology Please do not hesitate to contact me if you have any questions.

Congratulations and best wishes,

Version 3:

Decision Letter:

14th August 2025

Dear Salomé,

I am pleased to accept your Article "Dynamic expression of candidalysin facilitates oral colonization of *Candida albicans* in mice" for publication in Nature Microbiology. Thank you for having chosen to submit your work to us and many congratulations.

Authors may need to take specific actions to achieve compliance with funder and institutional open access mandates. If your research is supported by a funder that requires immediate open access (e.g. according to [Plan S principles](https://www.springernature.com/gp/open-science/plan-s-compliance) or the [NIH public access policy](https://www.springernature.com/gp/open-science/us-federal-agency-compliance)) then you should select the gold OA route, and we will direct you to the compliant route where possible. Because authors warrant under our subscription licensing terms that they haven't committed to licensing any version of their article under a licence inconsistent with the terms of our agreement – including the applicable embargo period – publication under the subscription model isn't suitable for authors whose funders require no embargo.

Enjoy the summer!

Best wishes,

P.S. Click on the following link if you would like to recommend Nature Microbiology to your librarian <http://www.nature.com/subscriptions/recommend.html#forms>

** Visit the Springer Nature Editorial and Publishing website at http://editorial-jobs.springernature.com?utm_source=ejP_NMicro_email&utm_medium=ejP_NMicro_email&utm_campaign=ejp_NMicro for more information about our career opportunities. If you have any questions please click [here](mailto:editorial.publishing.jobs@springernature.com).

Point-to-point reply

Reviewer #1

We would like to thank the reviewer for his/her positive feedback of our work and for highlighting that it addresses a critical gap in knowledge, presents a significant advance in the field and, that conclusions are solid. We would also like to thank him/her for the helpful suggestions to further improve the manuscript.

Major point 1: The authors posit that continuous ECE1 expression provides an intrinsic benefit for sustained colonization, while minimizing damage and excessive inflammation. On the other hand, high ECE1 expression is avoided in 101 to limit the damage, and potentially clearance. This conclusion is based on comparative analysis of SC5314 and 101, plus temporal expression analysis of ECE1 and the use of 101tetOff_ECE1 strain to knockout ECE1 in a doxycycline-dependent manner. While the data are convincing, a direct demonstration of enhanced ECE1 expression (under the 101 background) would be convincing. For example, does constitutive ECE1 expressing 101 strain, which the authors have shown to damage the cell line, exhibit enhanced clearance when used to infect the mice, similar to the SC5314 strain? Does this infection lead to excessive inflammation? Similarly, would the authors expect the 101 strain to have a competitive disadvantage relative to the parent strain when ECE1 is expressed under the constitutive promoter (e.g., their 101tetOff_ECE1 strain)? These data will increase the robustness of their findings if the authors can directly test this.

Author reply: As shown in Suppl. Fig. S3I, 101^{tetOff_ECE1} expresses elevated levels of ECE1, reaching levels comparable to those of strain SC5314, when exposed to keratinocytes in culture (Supplementary Figure S3J), which in turn resulted in increased cellular damage in IMOK cells, as shown in Fig. 3C. *In vivo* however, ECE1 expression by 101^{tetOff_ECE1} was not elevated compared to the parental strain 101 (**Figure Rev 1.1**), presumably due to different promoter regulation under the different experimental conditions. We also attempted a different strategy to overexpress ECE1 in strain 101, using promoters of the genes ADH1 and TDH3. However, despite choosing the strong TDH3 promoter, we could not achieve ECE1 expression levels that were high enough to elicit detectable levels of secreted peptide toxin or to increase damage induction in TR146 keratinocytes (see Lemberg et al., PMID: 35404986, Figure S5A), presumably due to the complex regulation of ECE1 expression, Ece1 folding and processing and candidalysin secretion and the requirements that must be met at the host interaction interface for candidalysin to exert its cytolytic activity (PMID: 29362237, PMID: 38388771, PMID: 34245079). We have also tested damage induction in IMOK cells, which are more sensitive than TR146 cells, but did again not observe cytolysis (LDH release) that was higher than for the parental 101 strain (**Figure Rev 1.2**). We did therefore not test any of these strains *in vivo* nor in a competitive setting.

In PMID: 35404986 we had pursued another strategy to test the consequences of elevated fungal virulence in the oral cavity of mice *in vivo*: Reducing the expression of NRG1 in strain 101 resulted in elevated ECE1 expression, which in combination with elevated filamentation and de-repression of other virulence factor was sufficient to drive inflammation (day 1, day 7) and enhanced clearance (day 7) in experimentally colonized mice *in vivo* (Figures 6 and 7 in PMID: 35404986). See also our reply to major point 2.

Figure Rev 1.1: *ECE1* expression in the tongue of C57BL/6 WT mice colonized with 101^{WT} or 101^{tetOff_ECE1} for 1 day or 7 days, as indicated.

Figure Rev 1.2: Damage induction in IMOK keratinocytes by 101 mutants (over)expressing *ECE1* under the *ADH1* or *TDH3* promoter.

Major point 2: Related to the above point, the authors discuss that Nrg1 can potentially regulate ECE1 transcription and thus may act as one of the regulators controlling ECE1 and hence the long-term commensal colonization. Would the authors expect Nrg1 KO 101 strain to get cleared rapidly?

Author reply: Our previous work (Lemberg et al., PMID: 35404986) showed that depletion of just one *NRG1* allele in strain 101 was sufficient to raise *ECE1* expression levels *in vitro* when in contact to keratinocytes. *In vivo*, the mutant drove an inflammatory response in the experimentally colonized tongue, which in turn resulted in a reduced fungal burden by day 7 (see Figure 6 in PMID: 35404986). Similar results were obtained when suppressing *NRG1* expression via a TET-off strategy (see Figure 7 in PMID: 35404986). Deletion of both *NRG1* alleles in strain 101 (101^{NRG1Δ/Δ}) led to strong aggregate formation, which precluded any *in vivo* colonization experiments (see Figure 6 in PMID: 35404986). We expanded the discussion to emphasize our previous work on *NRG1* and to better explain the regulation of *ECE1* expression via *Nrg1* (line 407).

Major point 3: The authors described Eed1 to maintain ECE1 expression and SC5314 EED1 KO to phenocopy the commensal phenotype of 101-WT. Conversely, EED1-overexpressing strain was said to recapture the pathogenic traits as judged using in vitro filamentation and ECE1 expression. Would the 101-EED1OE strain phenocopy the SC5314-WT in oral colonization experiment?

Author reply: We have conducted *in vivo* experiments with an *EED1*-overexpressing 101 mutant strain (101^{OE_EED1}) as suggested by the reviewer. Strain 101^{OE_EED1} did not colonize the oral mucosa equally well as the parental strain 101, presumably due to its high propensity to form aggregates, a property we previously observed for the *NRG1* deletion mutant (101^{nrg1Δ/Δ}, see PMID: 35404986). Despite the lower fungal load recovered from the tongue on day 1 post colonization with 101^{OE_EED1} compared to the parental strain 101, we observed that 101^{OE_EED1} formed longer filaments in the tongue tissue and elicited a local inflammatory response with

larger numbers of neutrophils recruited to the site of fungal colonization. Although the numbers of fungal colonization sites detected across the analyzed histology sections were not very high, these colonization sites were more inflammatory as compared to colonization sites in strain 101-colonized tongues. As a consequence of the inflammatory response, strain 101^{OE_EED1} was cleared from the tongue by day 7. We have included these data in a new Figure 4 L-N, and moved the data, which were originally shown in Supplementary Fig. S4C-D to the main Figures (new Figure 4J-K). We have also modified the text in the results section and legends accordingly (line 327 and line 1103).

Major point 4: The authors show that Hwp1 and Sap4-6 are not required for the early tissue persistence by 101, while they are required by SC5314. Is the non-requirement of these proteins by 101 true at later timepoints as well? Do the 101 sap4-6 KO and hwp1 KO show similar fungal burden to the parent 101 at later timepoints too?

Author reply: In response to the suggestion of the reviewer we have assessed the fate of 101^{hwp1Δ/Δ} and 101^{sap4-6Δ/Δ} in the oral mucosa over longer periods of time. As late as day 7 after fungal association, the fungal load was unchanged for both mutants compared to their parental strain 101. We have included these data in a new Supplementary Figure S2H and adapted the text in the results section and legends accordingly (line 207 and line 1143).

Major point 5: Related to the above point, it is possible that the Hwp1, Sap4-6, Hyr1 are not required for fungal persistence, but they may promote damage and inflammation. Does Sap4-6 or Hwp1 or Hyr1 overexpression under 101 parent or 101-ece1KO background promote higher inflammation? If yes, it is likely that their expression downregulation provides an additional mechanism to ensure long-term, non-inflammatory persistence.

Author reply: Although this is an interesting question, we consider it as very unlikely that *SAP5* or *HWP1* overexpression in strain 101 would be sufficient to elicit inflammation in the colonized epithelium. Strain 101 expresses *SAP5* and *HWP1* to comparable levels as strain SC5314 in the oral mucosa (see Figure 4H in the original version of the manuscript). We would argue that experimental overexpression of individual hyphae-associated genes in strain 101 would lead to an artificial scenario where these distinct, naturally hyphae-associated genes are expressed in the yeast morphology.

Strain 101, at least in parts switches back from hyphae to the yeast morphology in the oral mucosa – potentially as a mechanism to avoid inflammation – in agreement with this reviewer's comment. Based on our negative experience with our attempts to overexpress *ECE1* in strain 101 (see our response to this reviewer's first point), we consider it unlikely that the proposed overexpression experiment would lead to a noticeable inflammation-associated phenotype, as inflammation *in vivo* likely needs a combination of hyphal formation and expression of hyphae associated genes. This is further supported by the new data with 101^{OE_EED1} included in new Figure 4L-N of the revised version of the manuscript.

Minor point 1: It would be helpful if arrow-heads can be placed to indicate fungal elements in histological micrographs; especially for the histological micrographs for 8h timepoints, where there is a sparse amount of fungal presence.

Author reply: We thank the reviewer for this suggestion. We have added black arrows for easier identification of fungal elements on histological micrographs in Figures 1I, 2A, 3D, 3F, 3G, 4D,

S1F, S3J, S3K and in the new Figure 4M. In Figures 4D and 4M we have additionally added orange arrows to indicate inflammatory foci. The legends have been adjusted accordingly. See also our response to Reviewer #3, major point 3.

Minor point 2: For the method describing ex vivo incubation of tongues under "Oral colonization of mice with C. albicans", additional specific details would be helpful. For example, it would be helpful to know whether the tongues submerged in any medium or just humidification was sufficient? Were the tongues extracted and washed before placing in the plate for incubation or were directly placed?

Author reply: We have revised the description of the method in line 574.

Reviewer #2

This reviewer raises several points, some of which may be based on a misunderstanding. The first misunderstanding concerns the experimental model:

The murine oropharyngeal candidiasis (OPC) model consists of sublingually administering the high-virulent *C. albicans* strain SC5314 to mice that are commonly immunocompromised by corticosteroids (PMID: 22402633). An acute inflammation is elicited in response to the highly tissue damaging fungus, which is characterized by high levels of cytokines and chemokines and recruitment of neutrophils and inflammatory monocytes to the tongue. In turn, the fungus is rapidly cleared and usually not detectable beyond 4-5 days.

As we published, this model is strongly dependent on the fungal strain (PMID: 28176789). In contrast to strain SC5314, strain 101 establishes a long-lasting non-inflammatory association with the host in the oral cavity. Since our original publication, we extensively characterized this latter model, demonstrating that the fungus persists over months in the oral cavity of immunocompetent mice without causing tissue damage nor inflammation (PMID: 30873177, 32719409, 35404986). Therefore, the current study is a natural follow-up of our previous work, building on previous findings, rather than ignoring them as the reviewer reproaches.

We would like to clarify that, in contrast to what the reviewer claims in her introductory statement, strain 101 does not induce an inflammatory response in the oral mucosa (PMID: 30873177) and it does not get cleared several days-weeks post-inoculation (PMID: 28176789, 32719409). The long-lasting persistence of strain 101 that does not cause any harm to the colonized mice resembles *C. albicans* commensalism in humans. The model mimics *C. albicans* commensalism in humans also with respect to the homeostatic antifungal type 17 response that is induced with tissue-resident *C. albicans*-specific Th17 cells (PMID: 32719409), reminiscent of *C. albicans*-specific Th17 found in blood and tissue of naturally colonized individuals (PMID: 30799037, 29128674).

The second misunderstanding concerns the presence of *C. albicans* filamentous elements in the epithelial tissue:

Most studies visualizing *C. albicans* in the human tissue, and in the human oral cavity in particular, have analysed biopsies from mucocutaneous candidiasis patients. However, in contrast to this reviewer's view, *C. albicans* filaments have been described in the tissue of healthy individuals (PMID: 7823301, 36290044), emphasizing that hyphae are not exclusively a pathogenic morphotype. Firm association with the oral mucosa, including controlled invasion to the superficial layers of the stratified epithelium, represent a means by which *C. albicans* prevents being shed and swallowed. Our study proposes a mechanism, by which *C. albicans*

can establish its niche in the stratified epithelium to remain protected from saliva, mechanical forces and epithelial cell shedding.

We would further like to clarify that our study is not a follow-up of the publication to which the reviewer refers in her introductory statement (“work in a gut model published by investigators at Brown University this year in Nature”), which was co-authored by some of us (including the corresponding author Bernhard Hube as well as Tim Schille, Osama Elsafee, Selene Mogavero) and which proposed that *C. albicans* hyphae secreting candidalysin promotes intestinal fungal commensalism by overcoming colonization resistance mediated by gut bacteria, at least in part by directly acting against bacteria. Our present study is complementary and distinct. It demonstrates that in the oral cavity, continuous expression of the candidalysin-encoding gene *ECE1* provides *C. albicans* perpetual access to an epithelial niche that shields it from shear forces and from the antifungal activity of the immune system, that this occurs via direct candidalysin-host interactions, and that in contrast to the situation in the gut, this function of *ECE1* is independent of dysbiosis.

Specific point 1: Lines 95-96: This sentence suggests that the commensal state in the oral cavity is consistent with "residing inside the epithelium". A reminder to the authors that Candida is not an obligate intracellular pathogen, so invasion of the epithelial layer is not required for survival or a commensal lifestyle, especially in the nutrient-rich oral cavity. The premise that candidalysin breaches the keratin layer to allow commensalism to take place is counter to the central role of this toxin in cell damage and induction of inflammation that is associated with disease. Throughout this paper the authors conflate commensalism with low grade infection of avirulent strains.

Author reply: We adjusted the wording in line 96 to clarify that we are referring to the tissue localization of *C. albicans* strain 101 in the stratified epithelium of mice that were experimentally associated with the fungus. We are well aware that *C. albicans* is not an obligate intracellular microbe and we are not claiming that the fungus invades the intracellular compartment of epithelial cells.

Candidalysin has originally been characterized as a toxin that promotes host cell damage and induction of inflammation. In this study however, we provide several lines of evidence that the candidalysin-encoding gene *ECE1* has a role beyond causing tissue damage and inflammation. As such, we provide experimental evidence that *ECE1* enables *C. albicans* to establish a stable interaction with the host without driving noticeable inflammation. The outermost layer of the stratified epithelium of the murine tongue is composed of terminally differentiated anucleated keratinocytes, i.e. an accumulation of cell membranes, and pore formation in the membrane of dead cells does not elicit release of alarmins that trigger an inflammatory response.

Furthermore, our previous data have shown that repair mechanisms of oral epithelial cell can prevent and counteract candidalysin-mediated damage and can maintain epithelial integrity during both commensal growth and infection by *C. albicans* (PMID: 34986345).

That *ECE1*/candidalysin may have a role beyond causing host cell damage and inflammation has long been proposed (PMID: 19577508) and the first experimental evidence was demonstrated in 2024 (PMID: 38448595).

Specific point 2: Figures 1AB, S1CD are redundant, they show the reduced virulence (fungal burdens) of filamentation mutants in the SC5314 strain background, which involve genes with an established role in OPC pathogenesis. Also in describing these data, the authors use the term "colonization" inappropriately, since the SC5314 WT strain mounts a substantial

inflammatory response in this model early in the infection process. In prior publications with this model and this strain the group refer to this as infection, not colonization.

Author reply: This must be a misunderstanding. The mutant strains used in Figures 1A-B (SC5314^{efg1Δ/Δ}, SC5314^{efg1/cph11Δ/Δ}, SC5314^{flo8Δ/Δ}, SC5314^{ece11Δ/Δ}, SC5314^{ece1p3Δ/Δ}) and Supplementary Figures S1C-D (SC5314^{hyr1Δ/Δ} and SC5314^{sap1-3Δ/Δ}) have not been systematically studied in the oral cavity of immunocompetent mice (NB: most investigators pre-treat the animals with corticosteroids prior sublingual administration of the fungus, PMID: 22402633). Furthermore, we did not assess virulence (which in the oral model is evaluated as induction of inflammation) of these strains, but instead assessed the tongue fungal burden on day 1 to determine the capacity of the mutants to establish an interaction with the host. We adjusted the wording when referring to results with strain SC5314 and its derivatives in line 113 – 146.

Specific point 3: Figure 2: The statements that there are "similar degrees of filamentation" between strains SC5314 and 101, and that strain 101 "reverted back to the yeast morphology" are not supported by the histology images or any other data. In histology images, there is an isolated hyphal cell of strain 101 and a considerable number of hyphae in strain SC5314, and at 24h strain 101 seems to have mostly a yeast or short pseudohyphal morphology, consistent with its filamentation defect reported in previous work by this group (ref 31 and 32).

Author reply: We believe that the reviewer refers to Figures 1I-J, rather than Figure 2. We modified Figure 1I, now including additional histology sections to better document that strain 101 indeed filaments in the murine tongue at 8 hours. These data are consistent with our first published report of strain 101 (PMID: 28176789, see Figure 4). Subsequent studies comparing filamentation of strain 101 with strain SC5314 were conducted under *in vitro* conditions where it was found that strain 101 filaments with slower kinetics when put in serum-containing medium or in contact with keratinocytes (PMID: 35404986, PMID: 34212720). However, we never proposed that strain 101 is defective in filamentation.

Specific point 4: Lines 143-145: "together these data establish a filamentation prerequisite for oral colonization" in fact the data are weak and run contrary to the well-established defect in filamentation of strain 101 in vitro and in vivo.

Author reply: We would like to re-emphasize that strain 101 does filament. Please see our response to the specific point 3 above for more details. The statement in lines 143-145 (lines 145-146 in the revised version of the manuscript) refers primarily to the data in Figure 1A and 1E showing that yeast-locked mutants (derived from strain SC5314 as a parental) are defective in establishing a firm association with the tongue as tongue fungal burden was strongly diminished by 8 hours and 24 hours.

Specific point 5: The authors show that deletion of the candidalysin gene in the 101 strain background reduces oral fungal burdens in mice. These data support the hypothesis that invasion of the superficial epithelial layers is required for the increase in fungal burdens with this strain, but does not necessarily support a role in commensalism, this is a stretch. The acute and transient expression of ece1 also explains the reduced burdens in the ece1 KO strain in the 101 background. However, in previous work (ref#32) the Zurich group showed that overexpression of ece1 in this strain background does not change its phenotype in this mouse model. How can the authors reconcile this mouse work with the previously published?

Author reply: The reviewer claims that in our previous work (Lemberg, et al., PMID: 35404986), we “showed that overexpression of *ece1* in this strain background does not change its phenotype in this mouse model.”. This must be a misunderstanding. There is no discrepancy between this and our previous work. While we attempted to overexpress *ECE1* in strain 101 in the study by Lemberg et al., using promoters of gene *ADH1* and *TDH3*, we did not achieve *ECE1* expression levels that were high enough to elicit detectable levels of secreted peptide toxin or to increase damage induction in keratinocytes. We did, therefore, not use these mutant strains *in vivo*. Also see our response to Reviewer #1, major comment 1, and Reviewer #3, major comment 9.

*Specific point 6: Lines 211, 224-226: authors conclude that the role of *ece1* is "independent of antibioticly induced dysbiosis" and independent of the "oral microbiota". The evidence to support this statement is at best grossly inadequate. The investigators have not shown that treatment with ampicillin disrupts the oral mucosal bacterial communities in any quantifiable or qualitative manner. Typically, as in the Nature gut paper, a gnotobiotic model would be used to substantiate these claims, but at the very least bacterial CFUs should be presented in this work.*

Author reply: We followed the approach used in the Nature paper by Richard Bennett and colleagues (PMID: 38448595) to induce dysbiosis by treating mice with ampicillin. To demonstrate the effect of ampicillin, which was reported to reduce the bacterial faecal content by 100-1000 times (PMID: 28494240), we now include a new Supplementary Figure S2I-J, and we adjusted the text in line 218, line 279, and line 1146 in the revised manuscript. Also see our response to Reviewer #3, major point 5.

Specific point 7: Figures 2N-O are hard to follow and are not well described in the text. It is also unclear why the authors focused on these host genes and not on other functional group clusters (eg inflammation).

Author reply: We have adjusted the text in line 224 to better explain the data shown in Figures 2N-O. The legends were also updated (line 1042 and line 1044). As opposed to SC5314, acute inflammatory genes are barely induced by strain 101 in the tongue, as we demonstrated previously (PMID: 30873177), and are therefore not shown.

*Specific point 8: Figure 3: The source of the epithelial cell lines and confirmation of their phenotype, especially the mouse cell line should be described. The OEC:Fungal cell ratios are not mentioned in the LDH assays and the justification for the use of the IMOK cell line is confusing. The investigators state that they are using the IMOK cell line because it can be damaged more easily but this rationale seems to weaken and not strengthen their conclusions with the 101 strain. Also the doxycycline-inducible strain is used in the in vitro assay (with no added doxycycline), why not use the *ece1* overexpressing strain that the group has used in the past?*

Author reply: We included in the revised manuscript the source of the epithelial cell lines that were used in this study (line 479). The spontaneously immortalized IMOK cell line was characterized as described in detail in reference 52. IMOK cells have a cobblestone morphology typical of proliferative epithelial cells and can be induced to terminally differentiate. The reasons for using this cell line are stated in line 247.

As described in the methods section (line 485), cells were seeded at 2×10^5 cells/well in 24-well tissue culture plates or at 2×10^4 cells/well in 96-well tissue culture plates, respectively, and grown to confluent monolayers for 2 days prior to infection with the number of fungal cells indicated in the figure panels/legends.

The doxycycline-inducible strain 101^{tetOff_ECE1} expresses elevated levels of *ECE1* transcripts, reaching levels comparable to those of strain SC5314, when exposed to keratinocytes in culture (Supplementary Figures S3J), which in turn resulted in increased cellular damage, as shown in Fig. 3C. *In vivo* however, *ECE1* expression by 101^{tetOff_ECE1} was not elevated compared to the parental strain 101 (**Figure Rev 1.1**), which precluded using the strain for *ECE1* overexpression *in vivo*. Also see our response to Reviewer #1, major point 1. As explained above in our response to this reviewer's specific point 5, the previously generated 101 mutant strains (mentioned in Lemberg et al., PMID: 35404986) did not express *ECE1* levels that were high enough to elicit detectable levels of secreted peptide or to increase damage induction in keratinocytes.

Specific point 9: Figure 3 and Lines 260-1: The authors state they "allowed the colonization process to complete ex vivo". This is an inappropriate use of a model as a "colonization" model. The authors use an ex vivo tongue model to allow the strains to complete the "colonization" that started in vivo. The ex vivo model disregards the oral defenses that modify a strain's ability to colonize the oral mucosa, including saliva, mastication, neutrophils etc. In this model again the authors claim no role of the oral microbiota, without testing the effects of ampicillin ex-vivo.

Author reply: We rephrased the statement in line 271 and line 279 in response to the reviewer's concerns.

Specific point 10: Figure 4 and the conclusions drawn by comparing the eed1 KO strain in the SC5314 background to the WT 101 strain are very hard to follow and confusing. It is also unclear where the conclusion that ece1 helps strain 101 evade the inflammatory response is based, when the work shown in this figure involves the eed1 overexpressing and not the ece1 overexpressing strain in this strain background.

Author reply: We revised the text in lines 303 – 327 for more clarity.

Specific point 11: If indeed ece1 helps strain 101 establish residence in the cornified layer as the investigators posit, it would be appropriate to test whether candidalysin (among its many other functions) has keratinolytic activity. This strain is mainly localized in the acellular keratin layer and not deep in the stratified layers of the oral epithelium, thus epithelial cell damage may not be as important for "colonization" as this paper suggests.

Author reply: In response to the reviewer's suggestion, we performed a new experiment to elucidate the mechanism by which Ece1 may enable *C. albicans* to establish its niche in the cornified layer of the stratified epithelium. For this, we experimentally disrupted the outermost barrier of the stratum corneum, which allowed strain 101^{ece1Δ/Δ} to invade into the stratum corneum a controlled manner as illustrated by increased CFU counts in comparison to control mice in which the barrier was not experimentally disrupted, and by histology (new Figures 3I-J). The text was adjusted accordingly in the results section and in the legends (line 286, line 1080).

Further evidence for Ece1 playing a role in enabling *C. albicans* to overcome the keratinized barrier of the stratified epithelium is provided by *Il17rc*^{-/-} mice. *Il17rc*^{-/-} mice exhibit an

impaired barrier (as emphasized by the RNAseq data shown in Figure 2O), which facilitates *C. albicans* to gain access the stratum corneum, independently of Ece1. As such, we observed comparable colonization of the tongue of *Il17rc*^{-/-} mice by both, 101^{*ece1*Δ/Δ} and 101^{WT}. These data have been included in a new Figure 3H. The text was adjusted accordingly in the results section and in the legends (line 285, line 1077).

Reviewer #3

We like to thank this reviewer for his/her positive evaluation and helpful suggestions to further improve the manuscript. We appreciate his/her attention to detail and suggestions for improving linguistic accuracy.

Major point 1: None of the mutations are complemented. This is an essential control to demonstrate that the differences being observed are related to the gene that has been deleted, especially for new strains that the authors constructed. The authors should include a complemented strain along with the mutant strain, at least for key experiments.

Author reply: The mutants generated in the course of this study were made using CRISPR-Cas9 technology, a targeted gene editing method with very limited off-target effects. In case of 101^{*ece1*Δ/Δ}, 101^{*sap4,5,6*Δ/Δ}, and 101^{*hwp1*Δ/Δ} we tested several isogenic clones, which all exhibited comparable phenotypes (**Figure Rev 3.1**). For 101^{*ece1*Δ/Δ}, we therefore chose a single clone. For 101^{*sap4,5,6*Δ/Δ} and 101^{*hwp1*Δ/Δ}, we pooled data from all isogenic clones (Figure 1M-N, Supplementary Figure S2H).

Figure Rev 3.1: C57Bl/6 WT mice were associated sublingually with individual clones of the indicated mutants in comparison to the parental 101 strain and tongue CFU were enumerated after 8 hours (101^{*hwp1*Δ/Δ}, 101^{*hwp1*Δ/Δ}) or 1 day (101^{*ece1*Δ/Δ}).

*Major point 2: The histology images need some kind of quantification so the reader knows how frequently the observations were seen. Scoring of fungal localization and extent of filamentation should be conducted by blinded observers and provided. For example, quantification would support the very interesting conclusion stated in line 264 that *ece1* mutants underwent “extensive filamentation” but “remained peripheral”.*

Author reply: We did conduct blinded quantification of fungal localization in the stratified epithelium in Figure 3F in the original version of the manuscript. We conducted the scoring

fungal elements as 'outside', 'in the stratum corneum' or 'in the nucleated epithelium' and observed clear differences between *ECE1*-sufficient and -deficient *C. albicans*. We modified the legends to explain in more detail how the quantification was done (line 1074). Quantitative assessment of the length of individual filaments is however not possible on tissue sections, as the dimension of the filaments depends largely on the direction of cutting.

Major point 3: It would be helpful for non-mycologist readers if the authors added symbols (e.g. arrows/arrowheads) to point out fungal hyphae and yeast cells.

Author reply: Thank you for your suggestion. In response to the suggestion of this reviewer and reviewer #1 (minor point 1), we have added black arrows for easier identification of fungal elements on histological micrographs in Figures 1I, 2A, 3D, 3F, 3G, 4D, S1F, S3J, S3K and in the new Figure 4M. In Figures 4D and 4M we have additionally added orange arrows to indicate inflammatory foci. The legends have been adjusted accordingly.

Major point 4: Growth in tissue culture or cell-free serum. What was the morphology of the cells? Is ECE1 expression lower because substantial numbers of cells have converted to yeast morphology in one strain but not the other? Cellular morphology should be quantified.

Author reply: We assume that the reviewer is asking for morphological differences between strains SC5314 and 101. We have documented this previously in Lemberg et al. PMID: 35404986 and in Pultar et al. PMID: 34212720. Despite slightly delayed initial filamentation by strain 101 in comparison to strain SC5314, the induction of *ECE1* expression was largely comparable between the two strains at 2 hours of incubation in cell-free serum or upon exposure to keratinocytes (Figure 2B, 2C, Supplementary Figure S2A). Strain 101 retains filamentous morphology for up to 24 hours, both in cell-free medium and in contact with keratinocytes (**Figure Rev 3.2**). Therefore, the sharp drop in *ECE1* expression by strain 101 at 4 hours (Figure 2B-C, S2A) is unlikely a consequence of a morphotype transition.

Figure Rev 3.2: Filamentation of strain 101 on IMOK keratinocytes and in cell-free medium.

Major point 5: Before making conclusions about the role of bacterial dysbiosis (e.g. lines 12, 225) the authors need to show that their antibiotic treatment was sufficient to establish dysbiosis. Alternatively, the authors could change the language to state conclusions about the effect of antibiotic treatment.

Author reply: In the experiment shown in Figure 2L, we followed the approach previously described by Richard Bennett and colleagues (PMID: 38448595), which was reported to reduce the bacterial faecal content by 100-1000 times (PMID: 28494240). To demonstrate dysbiosis in response to ampicillin treatment in our experiment, we included a new Supplementary Figure

S2I-J showing dysbiosis upon ampicillin treatment. We also adjusted the text to moderate our statement (line 218, line 279, and line 1149).

Major point 6: Fig. 2I should include a control experiment with strain 101 without the Dox-regulated ECE1 gene to show that Dox treatment alone does not produce low colonization.

Author reply: We have followed the reviewer's suggestion and included a new Supplementary Figure S2F and adjusted the text accordingly in the results section and legends (line 200, line 1142).

Major point 7: The competition experiments need a control experiment (competition of tagged WT versus untagged WT) to show that the tagging does not change the colonization of the strain.

Author reply: We have performed this control experiment and included the results in a new Supplementary Figure S2G. The text was adjusted accordingly in the results section and legends (line 202, line 1144). Strain 101^{WT_mCherry} colonizes C57BL/6 mice equally well as the parental strain 101^{WT}, with a trend towards reduced colonization. The latter does not interfere with our conclusion that 101^{ece1Δ/Δ} exhibits a colonization failure when competing 101^{ece1Δ/Δ} with 101^{WT_mCherry}.

Major point 8: A nice way to analyze competition data is to calculate the Competitive Index (CI) which is the ratio of strain 1 in the output divided by strain 2 in the output divided by the ratio of the strains in the input. CI values can then be compared statistically between the control experiment (tagged WT vs untagged WT) and the experiment of interest. (Auerbuch et al, Infect Immun;69(9):5953–5957. doi: 10.1128/IAI.69.9.5953-5957.2001, PMID: 11500481.) Without this analysis, it is unclear how the statistics were calculated. Were all CFU measurements for one strain compared to all CFU measurements for the other without taking into account which values came from the same mouse?

Author reply: In all competition experiments, GFP-tagged and untagged *C. albicans* strains were compared for each animal individually and consequently we used statistical tests for paired datasets (i.e. paired two-tailed t-test in Figure 2J, 4I, S2G; Matched values Two-way ANOVA in Figures 2K, L and 2Q). We adjusted the methods section (line 657) and the legends accordingly for more clarity. Following the reviewer's suggestion, we calculated the fold difference in CFU between the two strain per animal for all competition experiments, in which we compared several experimental conditions (e.g. different time points in modified Fig. 2K, with/without antibiotic treatment in modified Fig. 2L, or different mouse genotypes in modified Fig. 2Q). We kept also our initial graphs to provide full transparency. We feel that for our data, showing fold differences in CFU is more appropriate and intuitive than the Competitive Index, given the 50x difference in input between 101^{WT} and 101^{ece1Δ/Δ}.

Major point 9: A key experiment would be to introduce the Dox-regulated ECE1 construct into the eed mutant and ask whether restoring expression of ECE1 would increase long term colonization and affect the localization of the organisms.

Author reply: We believe that reviewer is asking to overexpress *ECE1* in strain SC5314^{eed1Δ/Δ} by means of introducing the doxycycline construct that we found to overexpress *ECE1* in absence of doxycycline (Supplementary Figure S3I, Figure 3C). We would like to emphasize that the

construct does not drive *ECE1* overexpression *in vivo* (see our response to Reviewer #1's major point 1). We have attempted to overexpresses *ECE1* using various alternative approaches, both in strain 101 (see our response to Reviewer #1's major point 1), and in the yeast-locked mutant SC5314^{efg1/cph1Δ/Δ}, without achieving a change in the colonization behavior of the strain (**Figure Rev 3.3**). This is explained by the complex regulation of *ECE1* expression, Ece1 folding and processing and candidalysin secretion and the requirements that must be met at the host interaction interface for candidalysin to exert its cytolytic activity (PMID: 29362237, PMID: 38388771, PMID: 34245079).

Figure Rev 3.3: C57Bl/6 WT mice were associated with SC5314^{WT}, SC5314^{efg1/cph1Δ/Δ}, or SC5314^{efg1/cph1Δ/Δ} transfected with a construct in which the *ECE1* gene was placed under the control of the *ENO1* promoter. Tongue CFU were enumerated after 8 hours.

Major point 10: Fig. 2P shows colonization results at day 19. *Il17rc*^{+/-} mice at day 19 should be shown for comparison.

Author reply: In reply to the reviewer's suggestion, we include a new Supplementary Figures S2K showing tongue fungal load for 101^{WT} and 101^{ece1Δ/Δ} in *Il17rc*^{+/-} mice, 19 days after association. The results section and the legends were adjusted accordingly (line 230, line 1151).

Major point 11: Why is the ratio of *ECE1* expression in SC5314/101 so low *in vivo* (Fig. 3B)? Is expression in SC5314 very low *in vivo* compared to IMOK and expression in 101 is similar in the 2 conditions? Or is expression in SC5314 the same in both conditions and expression in 101 is very high *in vivo* but not in IMOK? What features of the IMOK vs *in vivo* environment are important for these differences in expression?

Author reply: *ECE1* expression by SC5314 is similar on IMOK cells and *in vivo*, while *ECE1* expression by 101 largely differs between the two conditions. We have added a new Supplementary Figure S3I showing the *ECE1* expression levels (relative to the housekeeping gene *ACTB*) by both strains in each condition. The results section and the legends were adjusted accordingly (line 254, line 1171). The fungal interaction interface provided by rapidly proliferating undifferentiated IMOK cells grown in monolayers under laboratory conditions vs. the stratified epithelium of the tongue tissue differ largely. While the exact host factors or conditions modulating *ECE1* expression under both conditions remain to be determined, we postulate that *C. albicans* depends more prominently on *ECE1* expression *in vivo* where it is continuously challenged to maintain its niche in the most differentiated and constantly renewing layer of keratinocytes to remain protected from adverse effects of saliva, physical forces and the immune system.

Major point 12: The *Ece1*-independent damage to IMOK cells is very interesting. Are *HWP1* or *SAP4-6* needed for this damage or does this damage represent an activity that is independent of standard virulence factors?

Author reply: Cytolytic damage of IMOK cells (quantified by monitoring LDH release) is not only independent of *ECE1*, but also independent of *HWP1* and *SAP4-6* (**Figure Rev 3.4**). We therefore conclude that the capacity of strain 101 to damage IMOK cells is independent of standard virulence factors.

Figure Rev 3.4: LDH release by IMOK cells in response to 101^{WT}, 101^{*hwp1*Δ/Δ} (2 isogenic clones), and 101^{*sap4-6*Δ/Δ} (3 isogenic clones).

Major point 13: Fig. 4 E,F,G what time points are shown, is Fig 4G really 24 hours? Why is SC5314 CFU so low? The legend says that the data are the same as Fig 2D, but Fig 2D shows *ECE1* gene expression and Fig 4G shows CFU. The legend needs to be corrected.

Author reply: We apologize for this error. CFU data shown in panel 4G are at 7 days. We have corrected the legends (line 1093).

Major point 14: If SC5314 CFU is very low (Fig 4G), how was it possible to measure *ECE1* gene expression from these cells at day 7 (panel 4H)?

Author reply: We apologize once again for the error in the legend to Figure 4. *ECE1* expression (shown in panel 4H) was measured at 24 hours. The corresponding CFU data are shown in panel 4C.

Minor point 1: The use of antibiotics needs to be described more clearly. The methods should state that antibiotics were not used in any of the experiments except for results shown in Fig. 2L and S3K. For S3K, were antibiotics present during the ex vivo incubation?

Author reply: The description of the methods has been adjusted to provide more details regarding the use of antibiotics in *in vivo* and *ex vivo* experiments (line 568).

Minor point 2: The legend for figure 2 should be corrected to explain the antibiotic experiment more fully. It sounds like all mice in Fig. 2L were treated with antibiotics.

Author reply: The legend to Figure 2L has been adjusted (line 1035)

Minor point 3: Fig. 1I is difficult to see. An inset as used in later figures would be helpful.

Author reply: Insets were added for each of the histology images in modified Figures 1I, as suggested. Arrows pointing to the fungal elements were also added.

Minor point 4: Throughout the figures, many different days are being compared. It would be helpful to label the various panels with the day of colonization being shown (as in Fig. 1A-H) so that the reader can quickly understand the differences between the panels.

Author reply: Time points of analysis have been added to all figure panels, as suggested by the reviewer.

Minor point 5: Are the statistics in Fig. 2B corrected for repeated measures? This should be mentioned in the methods.

Author reply: We adjusted the description of the statistical analyses in the methods section (line 657).

Minor point 6: How was the ex vivo incubation of tongue tissue conducted? Were the tissues incubated without any medium, submerged under medium or with an air-liquid interface? If there was medium, what medium was used and were antibiotics included?

Author reply: The methods were updated to provide more details on the ex vivo incubation of tongue tissue (line 574).

Minor point 7: Conclusions about morphology are complicated by the fact that Efg1, Cph1 and Flo8 are all transcription factors that regulate expression of genes. It is difficult to know how much a defect reflects the actual physical form of the fungus (a yeast versus a hypha) rather than the absence of proteins that decorate its surface. It would be best to modify some statements, e.g. that "filamentation is a prerequisite for oral colonization" (line 144), to take into account the possible contributions of proteins expressed by hyphae.

Author reply: We agree that mutants lacking *EFG1*, *CPH1*, or *FLO8* do not allow to discriminate whether *C. albicans* filamentation is required due to the physical form of hyphae or due to the expression of hyphae-associated genes, given the intricate connection between the morphological changes and the regulation of the hyphae-associated genes. Exactly because of this intricate relationship between the two processes, we believe that the conclusion 'filamentation is a prerequisite...' is justified as it does not preclude the implication of hyphae-associated genes. We therefore retained the 'filamentation' in line 144 (line 145 in the revised manuscript). Instead, we adjusted the wording in line 114.

Minor point 8: Line 144, filamentation is a prerequisite independent of fungal strain. Where was this shown for 101?

Author reply: The statement in lines 144 (lines 145-146 in the revised version of the manuscript) refers primarily to the data in Figure 1A and 1E showing that yeast-locked mutants (derived from strain SC5314 as a parental) are defective in establishing a firm association with the tongue as tongue fungal burden was strongly diminished by 8 hours and 24 hours. As we do not directly show the implication of morphotype switching by strain 101 we adjusted the statement in line 146.

Minor point 9: Line 152 candidalysin is essential... for colonization. There are a lot of colonies in Fig 1B so there is substantial candidalysin-independent colonization.

Author reply: We adjusted the wording in line 151 and line 154 to "... candidalysin is important ... for colonization."

Minor point 10: The terms "mucosal tissue" (line 174) and "oral mucosal tissue" (line 182) are vague since the authors have discussed mice, cell cultures and HEE. Authors should state what sample is being used. If it is mouse tongue, authors should describe whether a dissection was used to analyze just the mucosal tissue or whether they are analyzing the entire tongue.

Author reply: We have adjusted the wording in line 176 and line 184 as suggested by the reviewer to be more precise. The statement in line 176 refers to Figure 2D, which we made clear by citing the figure panel at the end of the sentence. In the original version of the manuscript, it was cited only in the subsequent linked sentence.

Minor point 11: Line 191, add comment that 50x less is labeled "ld" in the figure. Also define the "ld" term in the figure legend.

Author reply: We made the changes as suggested (line 193, line 1029)

Minor point 12: Line 194, should say inducible repression of ECE1 gene expression, not deletion.

Author reply: We corrected the wording as suggested (line 197).

Minor point 13: Line 219, "IL-17-deficient mice". What mice are being used for the experiment? The text seems to indicate Il17rc^{-/-} mice, which are knocked out for the receptor, not for IL-17.

Author reply: We apologize for this error and adjusted the wording (line 229).

Minor point 14: Fig. 2N, O needs more explanation. It should be stated that the results are from an RNASeq experiment.

Author reply: We adjusted the text in the results section and in the legends to provide more details as suggested by the reviewer (line 226, 1042, 1044).

Point-to-point reply (NMICROBIOL-24113500A, 2nd revision)

Reviewer #1 (Remarks to the Author):

In the revised manuscript, the authors have provided further experimental evidence for the requirement of the regulated expression of the fungal toxin – candidalysin – in establishing commensal colonization. Specifically, in this revised version, the authors have now provided the evidence for the functional requirements of Nrg1, Eed1 and their links to Ece1 during oral colonization. Similarly, the non-requirement of the additional hyphae-associated genes – HWP1, SAP4-6 – is also now experimentally demonstrated. These new data are robust, and they effectively address the comments 2 - 5 in my original critique.

Regarding my original comment 1 in the initial critique, the 101 strain overexpressing ECE1 would have been ideal to further the author's claim that higher ECE1 expression (hence candidalysin production) is damaging, inflammatory and leads to decreased colonization. The authors have provided additional data that ECE1 under constitutive "tet-off" promoter has higher ECE1 expression (at a level equivalent to the SC5314 strain) in vitro, but somehow in vivo ECE1 is not induced during colonization of the oral mucosa, which is rather intriguing. Furthermore, with the inability of strong promoters such as pTDH3 in enhancing candidalysin-induced damage, this reviewer appreciates the technical challenge of candidalysin-induction under 101. Do other oral commensal isolates (e.g., 529L) behave similarly (i.e., do they also not allow ECE1 over-expression and candidalysin-enhanced cell damage)? Some brief discussion would be ideal for the readers. While such an approach would have further solidified the authors' claim, the experimental evidence from multiple different angles, as present, has considerable strength.

Overall, I believe that the revised version is a much stronger version and supports the role of a regulated candidalysin expression in establishing oral colonization.

Author reply: We thank the reviewer for his/her positive feedback. We have now incorporated the figures, which we previously provided only to the reviewer, in the Extended Data Figures and adjusted the text in the manuscript accordingly.

We do not have access to other oral commensal isolates that are engineered to over-express *ECE1*. Generating additional mutants than the ones already included would go beyond the scope of this study. We have however added a statement in the discussion to acknowledge the limitation that this study is restricted to only two strain backgrounds.

Reviewer #2 (Remarks to the Author):

The authors have gone to extensive lengths to resolve what they perceive as "misunderstandings" and have done additional work which has improved the manuscript. However, the paper's central premise and authors' final conclusion that *C. albicans* depends on active penetration of the oral mucosa to establish commensal colonization, which is based on the phenotype of a single hypovirulent strain and its genetic derivatives, is conceptually flawed. *C. albicans* has several adhesins that could allow stable mucosal colonization without the need for invasion, at least in other strains. Mucosal invasion is seldom seen in healthy mucosa of humans and the references cited by the authors do not contradict this long held basic knowledge in oral pathology. It would be counterproductive for this organism to expend energy to invade tissues when it can form tight associations using adhesins such as hwp1. The work with strain 101 is extensive and scientifically sound. However, the overall conclusions regarding the commensal lifestyle of *C. albicans* should be toned down significantly, unless the authors can show that *ece1* plays a similar role in other commensal strains, preferably clinical isolates.

Author reply: We thank the reviewer for acknowledging that the work we performed during the revision has improved the manuscript and for commending the soundness of our work with strain 101 and its derivatives. We have added a statement in the discussion to acknowledge the limitation of this study to a single low-virulent strain. Furthermore, we have edited the entire manuscript to tone down the conclusions about the commensal lifestyle. As such, we have in many instances replaced 'commensalism' by 'colonization' and 'commensal strain' by 'low-virulent strain'.

Reviewer #3 (Remarks to the Author):

The authors have modified the paper in response to reviews and produced a stronger manuscript. Most of my comments have been appropriately addressed. A few clarifying comments added to the paper would be helpful.

Author reply: We thank this reviewer for his/her positive feedback on the revised version of our manuscript.

1. While it is nice that the authors provided figures for the reviewers, the purpose of reviewers' comments is to improve the paper for future readers. The authors should add a comment to Methods that multiple, independently isolated clones of mutant strains

were characterized, and multiple clones are shown in Fig. 1 M-N. Otherwise the “2 clones/3 clones” comments in the legend will be confusing for most readers.

Author reply: We have modified Fig. 1M-N to indicate the different clones by different figure symbols and adjust the legend accordingly. The individual clones are now also listed separately in Extended Data Table 1. We have furthermore incorporated all figures that we previously provided to the reviewers (point-to-point letter to the first round of comments) in the Extended Data Figures with corresponding adjustments to the text in the results section and legends.

2. In Methods, the authors should add the comment that histology images were scored by blinded investigators. This statement enhances the rigor of the analysis.

Author reply: We added the information in the Methods section (Line 599)

3. Mention/show in supplemental the morphology data for strain 101 (Figure Rev 3.2). It is very helpful to know that this strain retains filamentous morphology for up to 24 hrs.

Author reply: We included Figure Rev 3.2 in the Extended Data Figure 2 (new panel 2B) and adjusted the text accordingly.

4. Figure Rev 3.4 is also very interesting and should be added to the supplemental data to bolster the observation that damage of IMOK cells is independent of standard virulence factors.

Author reply: We included Figure rev 3.4 in the Extended Data Figure 3 (new panels 3I and 3J) and adjusted the text accordingly.